# LLaVA-FA: Learning Fourier Approximation for Compressing Large Multimodal Models

**Pengcheng Zheng[1], Chaoning Zhang[1]\*, Jiarong Mo[1], GuoHui Li[1], Jiaquan Zhang[1], Jiahao Zhang[2], Sihan Cao[1], Sheng Zheng[1], Caiyan Qin[3], Guoqing Wang[1], Yang Yang[1]**
[1]University of Electronic Science and Technology of China
[2]Chengdu University of Information Technology
[3]Harbin Institute of Technology, Shenzhen
zpc777@std.uestc.edu.cn, chaoningzhang1990@gmail.com

## Abstract

Large multimodal models (LMMs) have achieved impressive performance on various vision-language tasks, but their substantial computational and memory costs hinder their practical deployment. Existing compression methods often decouple low-rank decomposition and quantization, leading to compounded reconstruction errors, especially in multimodal architectures with cross-modal redundancy. To address this issue, we propose LLaVA-FA, a novel efficient LMM that performs joint low-rank plus quantization approximation in the frequency domain. By leveraging the de-correlation and conjugate symmetry properties of Fourier transform, LLaVA-FA achieves more compact and accurate weight representations. Furthermore, we introduce PolarQuant, a polar-coordinate quantization method tailored for complex matrices, and an optional diagonal calibration (ODC) scheme that eliminates the need for large-scale calibration data. Extensive experimental results demonstrate that our proposed LLaVA-FA outperforms existing efficient multimodal models across multiple benchmarks while maintaining minimal activated parameters and low computational costs, validating its effectiveness as a powerful solution for compressing LMMs.

## 1 Introduction

In the past few years, large multimodal models (LMMs) have shown its extraordinary performance on multiple down-stream tasks, such as zero-shot classification (Ge et al., 2023; Udandarao et al., 2024), image-text retrival (Long et al., 2023; Liu et al., 2025) and multimodal dialogue generation (Liu et al., 2024a; Koh et al., 2023). Despite their substantial applications, LMMs are computationally expensive, which limits their broader usage. For instance, the training of LLaVA 70B models necessitates a total of over 800 GPU hours, as calculated based on NVIDIA A100 GPUs (Li et al., 2024a). The substantial computational costs of LMMs present ongoing challenges. Aside from training, the inference of LMMs requires significant electricity consumption, which raises concerns about environmentally friendly AI. In light of this, developing efficient LMMs to reduce the memory and computational requirements is becoming crucial to ensure their broad accessibility.

Considering the correlated nature of multimodal semantics during training, the weight matrices of LMMs often exhibit redundancy, which manifests as a low-rank structure (Saha et al., 2024). To this end, one promising framework for memory-efficient LMMs is through parameter-efficient fine-tuning methods, *e.g.,* low-rank adaptation (LoRA) (Hu et al., 2022), where the pretrained models weight matrix $\mathbf{W}$ is reparameterized as $\mathbf{W} + \mathbf{L}_1\mathbf{L}_2$. Recent studies such as LoRD (Kaushal et al., 2023), ASVD (Yuan et al., 2023), FWSVD (Hsu et al.), LASER (Sharma et al., 2023), LQER (Zhang et al., 2024), and ZeroQuant-V2 (Yao et al., 2024b) have explored the efficacy of low-rank structures in large language models (LLMs) weights, utilizing LoRA technique to compress LLMs. However, these approaches treat low-rank factorization and quantization independently, which leaves the rank-selection stage blind to the impending quantization noise and inevitably compounds the final reconstruction error (Iacovides et al., 2024). Therefore, Saha et al. (Saha et al., 2024). approach

---

*Corresponding Author

this by uniquely formulating a joint optimization problem for generic matrices, providing additional flexibility for compression.

Due to its simplicity and astounding accuracy, the low-rank with quantization paradigm dominates in the field of efficient LLMs recently. However, we argue that this paradigm still faces storage challenges when handling extensive customization approximation for large vision-language models. For example, unlike pure-text LLMs, large vision-language models carry an extra tower of image encoders whose cross-modal adapter ranks balloon with every new visual domain, so the same low-rank with quantization recipe that tamed LLMs still leaves their multimodal stack uncomfortably obese. To this end, we naturally ask the question: *How can we aggressively compress learnable parameters even further for compressing large vision-language models?*

Previous works have demonstrated the powerful expressiveness of Fourier transform in data compression, where extremely sparse spectral information can be used to recover high fidelity 1D signal vectors (Rudelson & Vershynin, 2008; Duarte & Baraniuk, 2013; Zwartjes & Gisolf, 2007; Luo et al., 2026) and 2D image matrices (Vlaardingerbroek & Boer, 2013; Song et al., 2021). More importantly, when dealing with more general matrices, *e.g.,* weight matrices of neural networks that lack strong spatial semantics and are not frequency-spare, Fourier transform can still handle approximation effectively (Chen & Chi, 2013; Yang & Xie, 2016). Due to the inherent de-correlation capabilities of frequency domain, we observe that the weight matrices of LMMs in the frequency domain have a more compact spread of singular values as compare to spatial domain. Furthermore, we provide both theoretical and analytical evidence demonstrating that the accumulated error for matrix factorization in the frequency domain, utilizing a low-rank approximation, is smaller than that in the spatial domain at the same rank. This effectively enhances generalization performance. Additionally, benefiting from the conjugate symmetry of Fourier transform, approximating weight matrix in the frequency domain can save nearly half learnable parameters compared to the spatial domain. Motivated by these analyses [1], we investigate the potential for compressing the weight matrix of LMMs with matrix approximation in the frequency domain. We therefore ask a natural question: *Can we harness the de-correlation, conjugate symmetry, and energy-compaction merits of Fourier space to perform LoRA-style low-rank plus quantization compression in a single shot?*

To answer this question, we design a novel efficient and high-quality LMM with Fourier approximation, coined as LLaVA-FA, which approximately decomposes weight matrices into low-rank plus quantized weight in the frequency domain. We devise a novel PolarQuant codec that separately discretizes amplitude and phase in polar coordinates. The conjugate symmetry of Fourier transform allows us to store only half of the coefficients, while the energy concentration guarantees smaller Frobenius error for the same rank compared with spatial-domain truncation. Moreover, we derive an optional diagonal calibration scheme that approximates the full Hessian with row/column means, eliminating the need for large-scale calibration data [2]. Extensive experiments demonstrate that LLaVA-FA surpasses existing works across various benchmarks while maintaining minimal activated parameters and low computational costs. In conclusion, the main contributions of this paper are summarized as follows.

- We introduce LLaVA-FA, a novel efficient LMM that decomposes the weight matrices into a low-rank plus quantized weight via Fourier approximation.

- We design PolarQuant, an amplitude-and-phase polar codec that tailored for quantizing complex matrix. It preserves complex structure and stabilizes the low-bit reconstruction.

- We introduce an optional diagonal calibration (ODC) scheme, which can approximate the full Hessian matrix with row/column means, which enables more robust compression without large-scale calibration sets.

- Extensive experimental results demonstrate that LLaVA-FA surpasses existing works across various tasks, including comprehension-oriented and hallucination-oriented benchmarks, while maintaining minimal activated parameters and low computational costs.

---

[1] We give a detailed proof of these analyses in Appendix B.1.

[2] This approximation leverages the empirical observation that Hessian matrices in deep networks often exhibit diagonal dominance or low-rank structure, enabling accurate low-rank approximation via row/column means (LeCun et al., 1989; Botev et al., 2017).

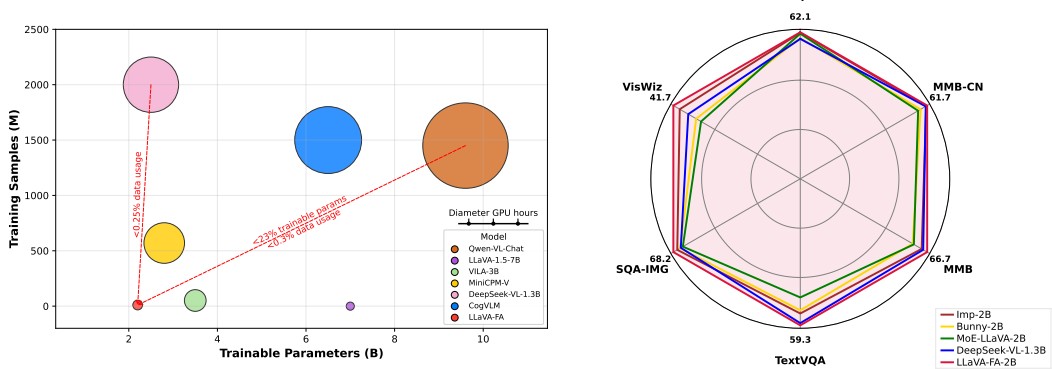

Figure 1: Comparisons of training cost and performance. LLaVA-FA achieves comparable performance with advanced LMMs using lower training costs while outperforming current efficient LMMs by a large margin.

## 2    RELATED WORKS

### 2.1    LARGE MULTIMODAL MODELS

The evolution of LMMs compression begins with LLM-Pruner (Ma et al., 2023), which adapts task-agnostic pruning from LLMs to LMMs. Subsequent studies highlighted that, beyond weight parameters, the predominant efficiency bottleneck in LMMs arises from the proliferation of visual tokens. To address this, LLaVA-PruMerge (Shang et al., 2024) introduces input-level optimization by adaptively retaining key tokens via attention scores, reducing up to $14\times$ tokens while preserving performance. DivPrune (Alvar et al., 2025) advances this line by emphasizing token diversity rather than mere importance, mitigating redundancy and ensuring richer visual representation. Cross-GET (Shi et al.) further pushes token-centric acceleration by ensemble-guided token selection, achieving $> 2\times$ throughput on VQA benchmarks. Complementary to token pruning, UPop (Shi et al., 2023) proposes a progressive magnitude-based pruning scheme that trims 40% of weights across vision-language layers with $< 1\%$ accuracy drop. Finally, CASP (Gholami et al., 2025) shifts focus inward, exploiting cross-modal attention sparsity to compress attention matrices $W_q, W_k$ with theoretical guarantees, thus marking the transition from generic LLM techniques to multimodal-specific compression paradigms. Unlike the token-centric CrossGET (Shi et al.) and weight-centric UPop (Shi et al., 2023), our method directly compresses both vision and language parts in one shot without pruning tokens or altering the architecture. Instead, we approximate the full weight matrix in the frequency domain via joint low-rank plus quantization, enabling calibration-free deployment and offering a drop-in replacement for weight-pruning when zero-shot compression is desired.

### 2.2    FOURIER TRANSFORM

In deep learning, applications of the Fourier transform in the frequency domain have gradually evolved from straightforward detection tasks to broader computational optimization. Jeong et al. (Jeong et al., 2022) introduce BiHPF, amplifying frequency-domain artifacts via bilateral high-pass filters for improving cross-domain generalization. Advanced with this, FreqNet (Tan et al., 2024) enforces continuous high-frequency focus through frequency domain learning and convolution layers on phase/amplitude spectra, significantly enhancing detection on unseen domains. Recently, Yang et al. (Yang et al., 2025) extend frequency analysis to model acceleration with FreqTS, which separates diffusion model token into high and low-frequency subsets, achieving markedly speedup without retraining by selectively reducing low-frequency features. To the best of our knowledge, we make the first attempt to perform a joint low-rank plus quantization approximation directly in the frequency domain.

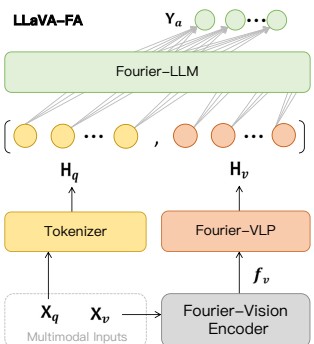

Figure 2: The architecture of the proposed LLaVA-FA.

## 2.3 LOW RANK ADAPTATION

The immense computational and storage costs of fully fine-tuning LLMs have spurred the development of Parameter-Efficient Fine-Tuning (PEFT) techniques. Among these, Low-Rank Adaptation (LoRA) (Hu et al., 2022) has become a foundational method. Subsequent research has focused on enhancing LoRA's efficiency and efficacy. For efficiency, QLoRA (Dettmers et al., 2023) makes a significant impact by back-propagating gradients through a frozen, 4-bit quantized base model into the LoRA adapters. To improve efficacy, DoRA (Liu et al., 2024d) decomposes pre-trained weights into magnitude and direction, applying LoRA specifically to the directional component, which outperforms standard LoRA. The application of LoRA has extended multimodal and multi-task learning, LLaVA-MoLE (Chen et al., 2024b) leverages a sparse mixture of LoRA experts to reduce data conflicts by routing tokens to task-specialized modules. Our approach integrates LoRA with quantization techniques to construct a more efficient LMM.

## 3 METHODOLOGY

We introduce LLaVA-FA, a novel framework for building an efficient LMM via Fourier approximation. As shown in Fig. 2, LLaVA-FA consists of four main modules: a Fourier-vision encoder tasked with receiving and processing visual inputs $X_v$, a text tokenizer to map text prompts $X_q$ to the text tokens $H_q$, and a Fourier Visual-Language Projector (Fourier-VLP), which functions as a bridge to align the two modalities. Then, text tokens $H_q$ and the visual tokens $H_v$ are concatenated as the input of the Fourier-based LLM, which outputs the final response sequence $Y_a$ in an autoregressive manner:

$$p(Y_a|H_v, H_q) = \prod_{i=1}^{L} p(y_i|H_v, H_q, y_{<i}), \tag{1}$$

where $L$ denotes the length of $Y_a$. LLaVA-FA relies on an elaborate factorization scheme, which decomposes each pretrained matrix into two low-rank complex matrices plus a quantized complex matrix via 2D-Fourier transform.

### 3.1 LOW-RANK PLUS QUANTIZED WEIGHT DECOMPOSITION

Due to the nature of language syntax and visual semantics learned during training, the weight matrices of LMMs often exhibit redundancy, which manifests as a low-rank structure. As shown in Fig. 3, the singular value profile of the weight matrices in LMMs follow a decaying profile. To this end, we can approximate the weight matrix of a neural network, $\mathbf{W}$, as a low-rank plus quantized weight decomposition, *i.e.,* $\mathbf{W} \approx \mathbf{Q} + \mathbf{L_1}\mathbf{L_2}$. The low-rank factors $\mathbf{L_1}$ and $\mathbf{L_2}$ capture the effect of the large singular component of $\mathbf{W}$ with full precision. Moreover, the backbone matrix $\mathbf{Q}$ is quantized to low-precision format - for instance, using $\mathrm{B_Q} = 2$ bits, coarsely captures the essence of the moderately decaying and low singular components of $\mathbf{W}$. We approach this problem by approximately solving this following minimization problem:

$$\min_{\mathbf{Q},\mathbf{L_1},\mathbf{L_2}} \|\mathbf{W} - (\mathbf{Q} + \mathbf{L_1}\mathbf{L_2})\|_F \quad \text{subject to} \quad \mathbf{Q} \in \mathbb{Q}_Q^{d_1 \times d_2}, \mathbf{L_1} \in \mathbb{R}^{d_1 \times k}, \text{ and } \mathbf{L_2} \in \mathbb{R}^{k \times d_2}. \tag{2}$$

Here, $\mathbb{Q}_Q^{d_1 \times d_2} \subset \mathbb{R}^{d_1 \times d_2}$ denote the lattice codebooks used to quantize $\mathbf{Q}$, using $\mathrm{B_Q}$ bits. This strategy leverages the approximate low rank structure inherent in weight matrix, where lesser contributing singular components can be pruned with minimal impact on the functionality of the matrix.

***Discussion*** *- we provide the analyses of computation of average bit budget per-parameter.* We assume that each transformer layer have seven weight matrices (*i.e.,* query, key, value, output, gate, up and down) with dimensions of $d_1^i \times d_2^i$, where $i \in \{1, \ldots, 7\}$. In general, if the backbone matrix $\mathbf{Q}$ has a bit budget of $\mathrm{B_Q}$, the low rank factors both have full bit budget of $\mathrm{B_L}$, then the average number of bits per parameter $\mathrm{B_{avg}}$ is

$$\mathrm{B_{avg}} = \sum_{i=1}^{7} \Big( \mathrm{B_Q}\, d_1^i d_2^i + k\, \mathrm{B_L}(d_1^i + d_2^i) \Big) / \sum_{i=1}^{7} d_1^i d_2^i. \tag{3}$$

When $k < \left(1 - \frac{\mathrm{B_Q}}{\mathrm{B_L}}\right) \frac{\sum_{i=1}^{7} d_1^i d_2^i}{\sum_{i=1}^{7}(d_1^i + d_2^i)}$ (the detailed derivations can be found in Appendix B.5), the average bits $\mathrm{B_{avg}}$ is smaller than the full precision bits $\mathrm{B_L}$. For LLaMa3-8B models, the query

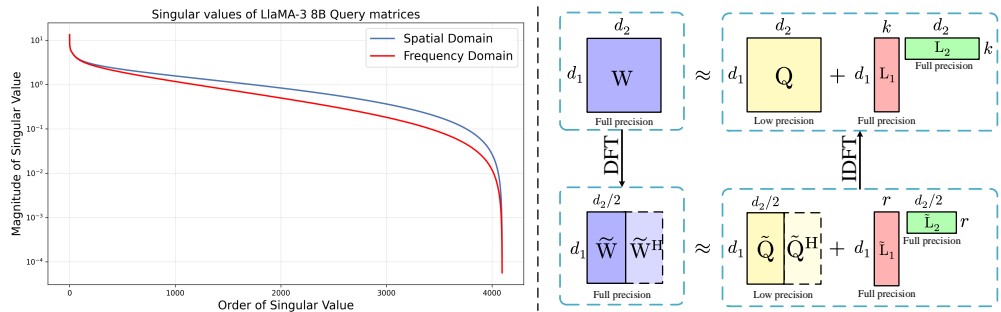

Figure 3: Illustration of Fourier approximation. **Left**: Singular value spread in the spatial domain *vs.* frequency domain. **Right**: LLaVA-FA decomposes a real-valued weight matrix $\widetilde{\mathbf{W}}$ into a frequency-domain low-rank component ($\widetilde{\mathbf{L}}_1$ and $\widetilde{\mathbf{L}}_2$) that keeps the top-r singular values with $b_r$, $b_\theta$ bits, plus a PolarQuant residual $\widetilde{\mathbf{Q}}$ for the remaining spectrum. The de-correlation and conjugate symmetry of Fourier transform lets us store only half of the complex coefficients, yielding the same rank as spatial-domain truncation but with fewer parameters and smaller Frobenius error.

and output matrices are $4096 \times 4096$, the key and value matrices are $4096 \times 1024$, the gate and up projections are $14336 \times 4096$, and the down projection is $4096 \times 14336$. For common LLMs configuration, $B_{avg} < B_L$ apparently holds.

---

**Algorithm 1:** Fourier Approximation

**Input** : Complex weight matrix $\widetilde{\mathbf{W}}$, Calibration matrix $\mathbf{C}$, Target rank $r$, Amplitude bitwidth $b_r$, Phase bitwidth $b_\theta$

Initialize $\widetilde{\mathbf{Q}} \leftarrow \mathbf{0}$ and $\epsilon_0 \leftarrow \infty$

**for** $t \leftarrow 1$ **to** $T-1$ **do**

  # Optional Diagonal Calibration Decomposition

  $\widetilde{\mathbf{L}}_1, \widetilde{\mathbf{L}}_2 \leftarrow \text{ODC}(\widetilde{\mathbf{W}} - \widetilde{\mathbf{Q}}, \mathbf{C}, r)$

  # Polar Quantization

  $\widetilde{\mathbf{Q}} \leftarrow \text{PolarQuant}\left(\widetilde{\mathbf{W}} - \widetilde{\mathbf{L}}_1\widetilde{\mathbf{L}}_2, b_r, b_\theta\right)$

  **if** $\mathbf{C}$ **is** None **then**

    # Weighted error

    $\epsilon_t = \left\|\left|\widetilde{\mathbf{W}} - (\widetilde{\mathbf{Q}} + \widetilde{\mathbf{L}}_1\widetilde{\mathbf{L}}_2)\right|\right\|_F$

  **else**

    # Calibration weighted error

    $\epsilon_t = \left\|\sqrt{\mathbf{C}} \odot \left|\widetilde{\mathbf{W}} - (\widetilde{\mathbf{Q}} + \widetilde{\mathbf{L}}_1\widetilde{\mathbf{L}}_2)\right|\right\|_F$

  **end**

  **if** $\epsilon_t > \epsilon_{t-1}$ **then**

    break

  **end**

**end**

**Output** : $\widetilde{\mathbf{Q}}, \widetilde{\mathbf{L}}_1, \widetilde{\mathbf{L}}_2, \epsilon_t$

---

**Algorithm 2:** ODC

**Input** : Complex residual matrix $\widetilde{\mathbf{R}} = \widetilde{\mathbf{W}} - \widetilde{\mathbf{Q}}$, Calibration matrix $\mathbf{C}$, Target rank $r$

**if** $\mathbf{C}$ **is** None **then**

  # Perform Fourier SVD on complex matrix

  $\widetilde{\mathbf{U}}, \mathbf{\Sigma}, \widetilde{\mathbf{V}}^H \leftarrow \text{FourierSVD}(\widetilde{\mathbf{R}})$

  # Keep only top $r$ singular values

  $\mathbf{\Sigma}_r \leftarrow \mathbf{\Sigma}[1:r, 1:r]$

  $\widetilde{\mathbf{U}}_r \leftarrow \widetilde{\mathbf{U}}[:, 1:r]$

  $\widetilde{\mathbf{V}}_r^H \leftarrow \widetilde{\mathbf{V}}^H[:, 1:r]$

  $\widetilde{\mathbf{L}}_1 \leftarrow \widetilde{\mathbf{U}}_r\sqrt{\mathbf{\Sigma}_r}, \widetilde{\mathbf{L}}_2 \leftarrow \sqrt{\mathbf{\Sigma}_r}\widetilde{\mathbf{V}}_r^H$

**else**

  $\mathbf{D}_{row} \leftarrow \text{RowAverage}(\mathbf{C})$

  $\mathbf{D}_{col} \leftarrow \text{ColAverage}(\mathbf{C})$

  $\widetilde{\mathbf{U}}, \mathbf{\Sigma}, \widetilde{\mathbf{V}}^H \leftarrow \text{FourierSVD}(\mathbf{D}_{row}\widetilde{\mathbf{R}}\mathbf{D}_{col})$

  # Keep only top $r$ singular values

  $\mathbf{\Sigma}_r \leftarrow \mathbf{\Sigma}[1:r, 1:r]$

  $\widetilde{\mathbf{U}}_r \leftarrow \widetilde{\mathbf{U}}[:, 1:r]$

  $\widetilde{\mathbf{V}}_r^H \leftarrow \widetilde{\mathbf{V}}^H[:, 1:r]$

  $\widetilde{\mathbf{L}}_1 \leftarrow \mathbf{D}_{row}^{-1}\widetilde{\mathbf{U}}_r\sqrt{\mathbf{\Sigma}_r}$

  $\widetilde{\mathbf{L}}_2 \leftarrow \sqrt{\mathbf{\Sigma}_r}\widetilde{\mathbf{V}}_r^H\mathbf{D}_{col}^{-1}$

**end**

**Output** : $\widetilde{\mathbf{L}}_1, \widetilde{\mathbf{L}}_2$

---

### 3.2 WEIGHT APPROXIMATION IN THE FREQUENCY DOMAIN

LLaVA-FA substitutes each weight matrix $\mathbf{W}$ in a LMM, with its approximate low-rank plus quantized weight decomposition in the frequency domain. Our approach is motivated by the de-correlation of frequency domain transforms, which minimizes spectral redundancy and is effective in compression since they concentrate most of the energy in a few coefficients.

**Lemma 3.1.** *Let* $\Delta\mathbf{W}_1$, $\Delta\mathbf{W}_2$ *be two adapters of identical shape,* $\sigma_{1,k}$, $\sigma_{2,k}$ *are their $k$-th singular values (sorted descending), and $r$ is the kept rank. If $\sigma_{1,k} < \sigma_{2,k}$ for every $k \geq r+1$, then*

$$\|\Delta\mathbf{W}_1 - \Delta\mathbf{W}_1^{(r)}\|_F < \|\Delta\mathbf{W}_2 - \Delta\mathbf{W}_2^{(r)}\|_F, \tag{4}$$

*where $\Delta\mathbf{W}_i^{(r)}$ denotes the rank-$r$ truncation of $\Delta\mathbf{W}_i$.*

We provide a detailed proof for the above lemma in Appendix B.6. As shown in Fig. 3, we observe that the weight matrix has a more compact spread of singular values in the frequency domain as compared to spatial domain. To this end, using Lemma 3.1, we can say that the accumulated error for low-rank approximation is smaller in the frequency domain compared to spatial domain with the same rank. At the same time, the conjugate symmetric property of Fourier transform implies that a matrix in the real spatial domain with the shape $[d_1, d_2]$ can be mapped into the complex frequency domain with the compressed shape $[d_1, d_2/2]$ without losing information. By leveraging this property, approximating a matrix in the frequency domain can save nearly half learnable parameters. Therefore, we approach the optimization problem of Eq. 2 from the perspective of Fourier transform where we first convert the real weight matrix $\mathbf{W} \in \mathbb{R}^{d_1 \times d_2}$ into the complex weight matrix $\widetilde{\mathbf{W}} \in \mathbb{C}^{d_1 \times d_2/2}$ via 2D Discrete Fourier transform:

$$\widetilde{\mathbf{W}}(u, v) = \sum_{m=0}^{M-1} \sum_{n=0}^{N-1} \mathbf{W}(m, n) \, e^{-j2\pi\left(\frac{um}{M} + \frac{vn}{N}\right)}, \tag{5}$$

where $u = 0, 1, \ldots, M - 1$ and $v = 0, 1, \ldots, N - 1$ are the frequency-domain indices, $j$ is the imaginary unit. Then, our goal is to obtain a decomposition $\widetilde{\mathbf{W}} \approx \widetilde{\mathbf{Q}} + \widetilde{\mathbf{L}}_1 \widetilde{\mathbf{L}}_2$ in the frequency domain by approximately solving the minimization problem:

$$\min_{\widetilde{\mathbf{Q}}, \widetilde{\mathbf{L}}_1, \widetilde{\mathbf{L}}_2} \left\| \sqrt{\mathbf{C}} \odot \left| \widetilde{\mathbf{W}} - (\widetilde{\mathbf{Q}} + \widetilde{\mathbf{L}}_1 \widetilde{\mathbf{L}}_2) \right| \right\|_F \quad \text{subject to} \quad \widetilde{\mathbf{Q}} \in \mathbb{P}_{b_r, b_\theta}^{d_1 \times d_2/2}, \widetilde{\mathbf{L}}_1 \in \mathbb{C}^{d_1 \times r}, \text{ and } \widetilde{\mathbf{L}}_2 \in \mathbb{C}^{r \times d_2/2}. \tag{6}$$

Here, $\sqrt{\mathbf{C}} \in \mathbb{R}^{d_1 \times d_2/2}$ is the calibration data that aims to preserve the Frobenius norm error of the compressed layer output activations. Following common practice in calibration-based quantization (Nagel et al., 2021; Dong et al., 2019), we construct $\mathbf{C}$ as the element-wise squared activation expectations over a small subset of training samples, i.e., $\mathbf{C}_{ij} = \mathbb{E}_{x \sim \mathcal{D}_{\text{calib}}}\left[(a_i(x))^2\right]$, where $\mathcal{D}_{\text{calib}}$ denotes the calibration dataset (usually 256-2048 held-out samples) and $a_i(x)$ is the activation of neuron $i$ under input $x$. $\odot$ denotes the Hadamard product. $|\cdot|$ represents the modulus, i.e., magnitude of the complex matrix. To be note worthy, $\mathbb{P}_{b_r, b_\theta}^{d_1 \times d_2/2} \subset \mathbb{C}^{d_1 \times d_2/2}$ denotes the designed PolarQuant lattice codebooks used to quantize the complex matrix $\widetilde{\mathbf{Q}}$, using amplitude bitwidth $b_r$ and phase bitwidth $b_\theta$.

### 3.3 COMPLEX MATRIX OPTIMIZATION WITH OPTIONAL CALIBRATION DATA

Following iterative algorithms which have been shown to be effective for matrix approximation (Guo et al., 2023), we heuristically solve Eq. 6 via alternating between optimizing $\widetilde{\mathbf{L}}_1$, $\widetilde{\mathbf{L}}_2$, and $\widetilde{\mathbf{Q}}$:

$$\begin{aligned}
\widetilde{\mathbf{L}}_1^{(t)}, \widetilde{\mathbf{L}}_2^{(t)} &\leftarrow \text{ODC}(\widetilde{\mathbf{W}} - \widetilde{\mathbf{Q}}^{(t-1)}, r), & &= \arg\min_{\text{rank}(\mathbf{L}) \leq r} \left\| \left| \widetilde{\mathbf{W}} - (\widetilde{\mathbf{Q}}^{(t-1)} + \widetilde{\mathbf{L}}) \right| \right\|_F, \\
\widetilde{\mathbf{Q}}^{(t)} &\leftarrow \text{PolarQuant}(\widetilde{\mathbf{W}} - \widetilde{\mathbf{L}}_1^{(t)} \widetilde{\mathbf{L}}_2^{(t)}), & &\approx \arg\min_{\mathbb{P}_{b_r, b_\theta}^{d_1 \times d_2/2}} \left\| \left| \widetilde{\mathbf{W}} - (\widetilde{\mathbf{Q}} + \widetilde{\mathbf{L}}_1^{(t)} \widetilde{\mathbf{L}}_2^{(t)}) \right| \right\|_F,
\end{aligned} \tag{7}$$

where $\widetilde{\mathbf{Q}}^{(0)}$ is initialized to 0. Unlike SVD for real matrix, FourierSVD operates on complex matrix, which decomposes any complex matrix $\widetilde{\mathbf{R}} \in \mathbb{C}^{d_1 \times d_2/2}$ into $\widetilde{\mathbf{U}} \Sigma \widetilde{\mathbf{V}}^H$, where $\widetilde{\mathbf{U}} \in \mathbb{C}^{d_1 \times d_1}$ and $\widetilde{\mathbf{V}} \in \mathbb{C}^{d_2/2 \times d_2/2}$ are both unitary, i.e., $\widetilde{\mathbf{U}}^H \widetilde{\mathbf{U}} = \mathbf{I}_m$, $\widetilde{\mathbf{V}}^H \widetilde{\mathbf{V}} = \mathbf{I}_m$. $\widetilde{\mathbf{V}}^H$ denotes the conjugate transpose of $\widetilde{\mathbf{V}}$. $\Sigma \in \mathbb{R}^{d_1 \times d_2/2}$ is real, diagonal and non-negative, $\Sigma = \text{diag}(\sigma_1, \ldots, \sigma_{\min(m,n)})$, $\sigma_k \geq 0$ – the singular values). In this paper, we keep only top $r$ singular values to get $\widetilde{\mathbf{L}}_1$ and $\widetilde{\mathbf{L}}_2$ by leveraging the low-rank property of matrix, i.e., $\widetilde{\mathbf{L}}_1 \leftarrow \widetilde{\mathbf{U}}_r \sqrt{\Sigma_r}$, $\widetilde{\mathbf{L}}_2 \leftarrow \sqrt{\Sigma_r} \widetilde{\mathbf{V}}_r^H$. Although recent works (Saha et al., 2024; Guo et al., 2023) demonstrate the importance of using calibration data for quantizing LLMs, the performance of these approaches highly relies on the availability and quality of calibration data, which may not always be accessible or representative. To mitigate this issue, we design an optional diagonal calibration (ODC) scheme to optimize the complex matrix optimization problem. Unlike its unweighted counterpart, calibration-weighted problem is in general intractable and in fact NP-hard. It is typically addressed through approximate methods, we consider the calibration-aware version of this approach by using a diagonal approximation of the calibration information matrix to weight the reconstruction objective:

$$\widetilde{\mathbf{L}}_1, \widetilde{\mathbf{L}}_2 = \arg\min_{\widetilde{\mathbf{L}}_1, \widetilde{\mathbf{L}}_2} \left\| \sqrt{\mathbf{C}} \odot \mathbf{E} \right\|_F = \arg\min_{\widetilde{\mathbf{L}}_1, \widetilde{\mathbf{L}}_2} \left\| \mathbf{D}_{\text{row}} \mathbf{E} \mathbf{D}_{\text{col}} \right\|_F, \tag{8}$$

where given $\mathbf{E} := \left| \widetilde{\mathbf{W}} - (\widetilde{\mathbf{Q}} + \widetilde{\mathbf{L}}_1 \widetilde{\mathbf{L}}_2) \right|$, $\mathbf{D}_{\text{row}}$ is a diagonal matrix consists of row-means of $\sqrt{\mathbf{C}}$, and $\mathbf{D}_{\text{col}}$ is a diagonal matrix consisting of the column-means of $\sqrt{\mathbf{C}}$, *i.e.*,

$$\mathbf{D}_{\text{row}} = \text{diag}\left( \left[ \text{avg}(\sqrt{\mathbf{C}_{1,\cdot}}), \dots, \text{avg}(\sqrt{\mathbf{C}_{d_1,\cdot}}) \right] \right), \mathbf{D}_{\text{col}} = \text{diag}\left( \left[ \text{avg}(\sqrt{\mathbf{C}_{\cdot,1}}), \dots, \text{avg}(\sqrt{\mathbf{C}_{\cdot,d_2/2}}) \right] \right). \tag{9}$$

To this end, this problem can be solved by the FourierSVD algorithm:

$$\widetilde{\mathbf{U}}, \boldsymbol{\Sigma}, \widetilde{\mathbf{V}}^H \leftarrow \text{FourierSVD}(\mathbf{D}_{\text{row}} \widetilde{\mathbf{R}} \mathbf{D}_{\text{col}}), \quad \widetilde{\mathbf{L}}_1 \leftarrow \mathbf{D}_{\text{row}}^{-1} \widetilde{\mathbf{U}}_r \sqrt{\boldsymbol{\Sigma}_r}, \quad \widetilde{\mathbf{L}}_2 \leftarrow \sqrt{\boldsymbol{\Sigma}_r} \widetilde{\mathbf{V}}_r^H \mathbf{D}_{\text{col}}^{-1}. \tag{10}$$

Here, we keep only top $r$ singular values of $\boldsymbol{\Sigma}$, *i.e.*, $\boldsymbol{\Sigma}_r \leftarrow \boldsymbol{\Sigma}[1:r, 1:r]$, $\widetilde{\mathbf{U}}_r \leftarrow \widetilde{\mathbf{U}}[:, 1:r]$, $\widetilde{\mathbf{V}}_r^H \leftarrow \widetilde{\mathbf{V}}^H[:, 1:r]$. We provide a detailed theoretical and empirical justification of ODC in Appendix B.7. The detailed computational process of the proposed ODC algorithm is described in Algorithm 2.

After obtaining the low-rank approximation $\widetilde{\mathbf{L}}_1$ and $\widetilde{\mathbf{L}}_2$, the residual complex matrix $\widetilde{\mathbf{R}} = \widetilde{\mathbf{W}} - \widetilde{\mathbf{L}}_1 \widetilde{\mathbf{L}}_2$ still contains non-negligible frequency-domain energy. To further compress this residual, we propose PolarQuant, a polar-coordinate-based quantization method tailored for complex matrices. Specifically, PolarQuant first decomposes each element of $\widetilde{\mathbf{R}}$ into its real and imaginary components, *i.e.*, $\mathbf{X} = \text{Re}(\widetilde{\mathbf{R}})$, $\mathbf{Y} = \text{Im}(\widetilde{\mathbf{R}})$. Then, it converts each complex entry into its polar form, *i.e.*,

$$r_{ij} = \sqrt{X_{ij}^2 + Y_{ij}^2}, \quad \theta_{ij} = \text{atan2}(Y_{ij}, X_{ij}), \tag{11}$$

where $r_{ij}$ and $\theta_{ij}$ represent the amplitude and phase of the residual frequency component at location $(i, j)$, respectively. Next, we compute uniform quantization step sizes for amplitude and phase,

$$\Delta r = \frac{\max(r_{ij})}{2^{b_r} - 1}, \quad \Delta \theta = \frac{2\pi}{2^{b_\theta}}, \tag{12}$$

where $b_r$ and $b_\theta$ are the bit-widths for amplitude and phase, respectively. Each amplitude and phase value is then independently quantized, *i.e.*, $q_{r,i,j} = \text{round}(r_{i,j}/\Delta r)$, $q_{\theta,i,j} = \text{round}(\theta_{i,j} + \pi/\Delta_\theta)$. Finally, the quantized complex matrix $\widetilde{\mathbf{Q}}$ is reconstructed as $\widetilde{\mathbf{Q}}_{ij} = \hat{r}_{i,j} \cdot e^{i\hat{\theta}_{i,j}}$. This approach allows us to represent the residual matrix in a compact polar format while preserving the phase structure critical for cross-modal alignment. The entire process is summarized in Algorithm 3. We employ a simple stopping criterion where we keep track of the error $\epsilon_t$ and terminate the algorithm if the error increases, *i.e.*, $\epsilon_t > \epsilon_{t-1}$. The proposed Fourier approximation algorithm is shown in Algorithm 1.

---

**Algorithm 3: PolarQuant**

**Input** : Complex residual matrix $\widetilde{\mathbf{R}} = \widetilde{\mathbf{W}} - \widetilde{\mathbf{L}}_1 \widetilde{\mathbf{L}}_2$, Amplitude bitwidth $b_r$, Phase bitwidth $b_\theta$

\# Extract real and imaginary parts
$\mathbf{X} \leftarrow \text{Re}(\widetilde{\mathbf{R}})$, $\mathbf{Y} \leftarrow \text{Im}(\widetilde{\mathbf{R}})$
\# Convert to polar coordinates
**for** *each element* $(i,j)$ **do**
$\quad r_{i,j} \leftarrow \sqrt{X_{i,j}^2 + Y_{i,j}^2}$
$\quad \theta_{i,j} \leftarrow \text{atan2}(Y_{i,j}, X_{i,j})$
**end**
\# Compute quantization parameters
$r_{\max} \leftarrow \max(r_{i,j})$
$\Delta_r \leftarrow r_{\max}/(2^{b_r} - 1)$
$\Delta_\theta \leftarrow 2\pi/(2^{b_\theta})$
\# Quantize amplitude and phase
**for** *each element* $(i,j)$ **do**
$\quad q_{r,i,j} \leftarrow \text{round}(r_{i,j}/\Delta_r)$
$\quad q_{\theta,i,j} \leftarrow \text{round}((\theta_{i,j} + \pi)/\Delta_\theta)$
$\quad \hat{r}_{i,j} \leftarrow q_{r,i,j} \cdot \Delta_r$
$\quad \hat{\theta}_{i,j} \leftarrow q_{\theta,i,j} \cdot \Delta_\theta - \pi$
**end**
\# Reconstruct complex matrix
**for** *each element* $(i,j)$ **do**
$\quad \widetilde{\mathbf{Q}}_{i,j} \leftarrow \hat{r}_{i,j} \cdot e^{i\hat{\theta}_{i,j}}$
**end**
**Output** : Quantized complex matrix $\widetilde{\mathbf{Q}}$

---

## 4 EXPERIMENTS

### 4.1 EXPERIMENTAL SETTINGS

**Implementation Details.** The proposed architecture employs pretrained CLIP-ViT-L/14 (Radford et al., 2021) as the vision encoder, which is post trained via the Fourier approximation. A two-layer MLP that compressed via Fourier approximation functions as the VLP. For the LLM, Qwen-2.5 (Hui et al., 2024) series models at different scales are utilized as the foundation for LLaVA-FA. Specifically, LLaVA-FA-2B and LLaVA-FA-1B are derived through compression from Qwen-2.5-7B and Qwen-2.5-3B, respectively. Moreover, LLaVA-FA-7B and LLaVA-FA-3B are derived through compression from InternLM-2-20B (Cai et al., 2024) and LLaMA-3-8B (Dubey et al., 2024), respectively. LLaVA-FA is trained using 8 NVIDIA RTX 4090 GPUs. The detailed training strategy and hyperparameter are illustrated in Appendix C.1.

**Training Datasets.** LLaVA-FA is trained on a diverse collection of vision-language datasets spanning pre-training (*e.g.*, LLaVA-Pretrain (Liu et al., 2023b), ShareGPT4V (Chen et al., 2024a)), visual

Table 1: Comparison with state-of-the-art LMMs on the commonly-used multimodal benchmarks for LMMs. #Sample: Training data sample. #Param: Trainable parameters. $SQA^I$: ScienceQA test, $VQA^T$: TextVQA val, MME: MME Benchmark, normalized to percentage, MMB: MMBench dev, $MMB^{CN}$: MMBench-Chinese dev. The optimal result is shown in bold, and the sub-optimal result is underlined. Our LLaVA-FA achieves the best average result for both.

| Method | LLM | #Sample | #Param | GQA | VisWiz | $SQA^I$ | $VQA^T$ | MME | MMB | $MMB^{CN}$ | AVG |
|---|---|---|---|---|---|---|---|---|---|---|---|
| BLIP-2 | Vicuna-13B | 129M | | 41.0 | 19.6 | 61.0 | 42.5 | 64.7 | - | - | - |
| VILA-7B | LLaMA-7B | 50M | | 62.3 | 57.8 | 68.2 | 64.4 | 76.7 | 68.9 | 61.7 | 65.7 |
| CogVLM | Vicuna-7B | 1500M | | 64.9 | - | 65.6 | **78.2** | 71.8 | 63.7 | 53.8 | - |
| InstructBLIP | Vicuna-13B | 130M | | 49.5 | 33.4 | 63.1 | 50.7 | 60.6 | - | - | - |
| Qwen-VL-Chat | Qwen-7B | 1450M | ≥7B | 57.5 | 38.9 | 68.2 | 61.5 | 74.4 | 60.6 | 56.7 | 56.7 |
| Deepseek-VL-7B | DLLM-7B | 2000M | | 61.3 | 49.9 | 74.0 | 64.7 | 73.4 | 74.1 | **72.8** | 67.2 |
| LLaVA-1.5-7B | Vicuna-1.5-7B | 1.2M | | 62.0 | 50.0 | 66.8 | 58.2 | 75.5 | 64.3 | 58.3 | 62.1 |
| LLaVA-NeXT | Vicuna-1.5-13B | 1.3M | | 65.4 | 60.5 | 73.6 | 67.1 | **78.7** | 70.4 | 64.4 | 68.5 |
| LLaVA-FA-7B | InternLM-2-20B | 5M | | **68.5** | **62.0** | **76.0** | 68.0 | 74.5 | **74.5** | 69.5 | **70.4** |
| Imp-3B | Phi-2-2.7B | 1.6M | | 63.5 | 54.1 | 72.8 | 59.8 | 72.3 | **72.9** | 46.7 | 63.2 |
| Bunny-3B | Phi-2-2.7B | 2.7M | | 62.5 | 43.8 | 70.9 | 56.7 | **74.4** | 68.6 | 37.2 | 59.2 |
| VILA-3B | LLaMA-2.7B | 51M | | 61.5 | 53.5 | 69.0 | 60.4 | 72.1 | 63.4 | 52.7 | 61.8 |
| MobileVLM | MLLaMA-2.7B | 1.3M | | 59.0 | - | 61.0 | 47.5 | 64.4 | 59.6 | - | - |
| MobileVLM$^{v2}$ | MLLaMA-2.7B | 3.6M | ~3B | 61.1 | - | 70.0 | 57.5 | 72.0 | 63.2 | - | - |
| MoE-LLaVA-3B | Phi-2-2.7B | 2.2M | | 61.4 | 43.9 | 68.5 | 51.4 | 71.1 | 65.2 | 41.8 | 57.6 |
| MiniCPM-V | MiniCPM-2.4B | 570M | | 51.5 | 50.5 | 74.4 | 56.6 | 68.9 | 64.0 | 62.7 | 61.2 |
| MiniCPM-V-2 | MiniCPM-2.4B | 570M | | 52.1 | 60.2 | 76.3 | **73.2** | 70.5 | 68.5 | 67.2 | 66.9 |
| LLaVA-FA-3B | LLaMA-3-8B | 5M | | **65.0** | **62.5** | **77.0** | 64.0 | 71.0 | 70.5 | **68.0** | **68.3** |
| Imp-2B | Qwen-1.5-1.8B | 1.6M | | 61.9 | 39.6 | 66.1 | 54.5 | 65.2 | 63.8 | 61.2 | 58.9 |
| Bunny-2B | Qwen-1.5-1.8B | 2.7M | | 59.6 | 34.2 | 64.6 | 53.2 | 65.0 | 59.1 | 58.5 | 56.3 |
| Mini-Gemini-2B | Gemma-2B | 2.7M | ~2B | 60.7 | 41.5 | 63.1 | 56.2 | **67.0** | 59.8 | 51.3 | 57.1 |
| MoE-LLaVA-2B | Qwen-1.5-1.8B | 2.2M | | 61.5 | 32.6 | 63.1 | 48.0 | 64.6 | 59.7 | 57.3 | 55.3 |
| DeepSeek-VL-1.3B | DLLM-1.3B | 2000M | | 59.3 | 34.6 | 64.2 | 58.4 | 55.3 | 64.6 | 61.0 | 58.5 |
| LLaVA-FA-2B | Qwen-2.5-7B | 5M | | **62.1** | **41.7** | **68.2** | **59.3** | 66.6 | **66.7** | **61.7** | **60.9** |
| SPHINX-Tiny | TLLaMA-1.1B | 15M | ~1B | **58.0** | 49.2 | 21.5 | 57.8 | 63.1 | **56.6** | 37.8 | 49.2 |
| LLaVA-FA-1B | Qwen-2.5-3B | 5M | | 56.7 | **49.7** | **61.3** | **57.9** | **63.3** | **58.3** | **49.4** | **56.7** |

question answering across general (GQA (Hudson & Manning, 2019), VQA (Goyal et al., 2017)), text-centric (OCR-VQA (Mishra et al., 2019), SynthDoG-EN (Kim et al., 2022)), and specialized domains (ChartQA (Masry et al., 2022), DocVQA (Clark & Gardner, 2018)), visual reasoning and instruction-following (MIMIC-IT (Li et al., 2023a), LRV (Liu et al., 2023a)), and large-scale web data (RefCOCO (Yu et al., 2016), VisualGenome (Krishna et al., 2017)). This comprehensive mixture covers natural images, documents, charts, diagrams, and synthetic visualizations, balancing general visual understanding with specialized capabilities in OCR and spatial reasoning.

**Evaluation Benchmarks.** We conduct experiments on three multimodal benchmarks-MME (Fu et al., 2024), MMB (Liu et al., 2024e), and $MMB^{CN}$-to evaluate multimodal understanding and reasoning capabilities. Our evaluation also encompasses diverse VQA tasks across three domains: general visual understanding using VizWiz (Gurari et al., 2018) and GQA (Hudson & Manning, 2019); text-oriented recognition via TextVQA (Singh et al., 2019); and scientific reasoning through ScienceQA (Lu et al., 2022). Additionally, we assess model reliability using hallucination benchmarks including POPE (Li et al., 2023b), Object HalBench (Yu et al., 2024), and MMHal-Bench (Sun et al., 2024).

## 4.2 MAIN RESULTS

This section evaluates LLaVA-FA across two critical dimensions: model performance and computational efficiency. Performance is assessed through comprehensive results on comprehension-oriented benchmarks (Tab. 1) and hallucination-oriented evaluations (Tab. 2). Efficiency analysis examines training data requirements and model parameters. ***Furthermore, Appendix C.2 evaluates the real-world inference overhead introduced by DFT/IDFT and PolarQuant operations, demonstrating that their latency contribution is minimal (3.2% of TTFT) and more than compensated by compression gains.***

**Comprehension-Oriented Benchmarks.** As shown in Tab. 1, LLaVA-FA achieves competitive results across comprehension-oriented benchmarks. With only 5M training samples, the 2B variant obtains an average score of 60.8%, surpassing MoE-LLaVA-2B (Lin et al., 2024) by 5.5% and Mini-Gemini-2B (Li et al., 2024b) by 3.7%. Despite its compact size, LLaVA-FA-2B approaches the performance of LLaVA-1.5-7B (Liu et al., 2024c) (62.1% *vs.* 60.8%) while using fewer parameters. The 1B variant matches SPHINX-Tiny's (Liu et al., 2024b) performance (49.4% *vs.* 49.2%) with substantially reduced training data (5M *vs.* 15M). Notably, LLaVA-FA's strong performance on VQA$^\text{T}$ and MMB benchmarks, demonstrating its effectiveness in multimodal understanding tasks. We further evaluate our method on substantially larger backbones: LLaVA-FA-3B, constructed from LLaMA-3-8B (Dubey et al., 2024), and LLaVA-FA-7B, constructed from InternLM-2-20B (Cai et al., 2024). Importantly, both models are compressed using exactly the same Fourier approximation pipeline, ensuring strict fairness and demonstrating that our method generalizes without any modification or heuristic tuning. Experimental results show that Fourier-domain compression continues to yield strong performance across diverse benchmarks, with even smaller relative degradation at the 8B/20B scales. These results verify that our Fourier approximation method seamlessly extends to LLMs far beyond Qwen-2.5, maintaining both accuracy and robustness while preserving the same compression pipeline across all model sizes. This confirms that the proposed method is truly model-family agnostic and highly scalable.

**Hallucination-Oriented Benchmarks.** As shown in Tab. 2, LLaVA-FA demonstrates exceptional performance in mitigating hallucination, achieving competitive results despite its compact 2B parameter size. It can be attributed to the effectiveness of our approach from two aspects: Firstly, by maintaining high factual accuracy with an F1 score of 87.5% on POPE, LLaVA-FA ensures reliable and grounded visual understanding. Secondly, by achieving remarkably low hallucination rates (11.2% response-level and 7.7% mention-level on Object HalBench), it significantly reduces the generation of incorrect or unfounded information. Notably, LLaVA-FA outperforms other 2B-scale models including MiniCPM-V-2 (Yao et al., 2024a) by 3.3% in response-level hallucination rate and Mini-Gemini-2B (Li et al., 2024b) by 18.5%. Furthermore, it even surpasses several larger 7B models such as VCD (48.8% response-level) and POVID (48.1% response-level) by over 37% on Object HalBench, demonstrating that our approach enables efficient hallucination mitigation without requiring extensive model scaling. The strong performance on hallucination benchmarks can be explained by the properties of the proposed Fourier-domain decomposition. Specifically, applying the Fourier transform reduces cross-modal redundancy and produces a more compact singular value distribution. As discussed and supported by Lemma 3.1, the low-rank truncation in the frequency domain introduces a smaller approximation error than in the spatial domain under the same rank. This leads to more stable intermediate representations and reduces the error propagation that often triggers hallucination in multimodal reasoning (He et al., 2025; Sun et al., 2024).

**Floating Point Operations and Latency** In practice, we evaluate the FLOPs (Floating Point Operations) and latency of LLaVA-FA under varying numbers of vision tokens using *calflops* and our inference benchmark, both conducted on eight RTX 4090 GPUs. Specifically, we measure FLOPs, Time to First Token (TTFT), and KV cache usage within the same multi-GPU environment. As illustrated in Tab. 3 and Fig. 4, LLaVA-FA achieves a substantial reduction in computational cost compared to Imp-2B. Furthermore, our method lowers KV cache usage by 83.3% and improves inference speed by 61.1% in TTFT, outperforming Imp-2B under equivalent visual token counts. These improvements highlight the efficiency of Fourier approximation approach, which is crucial for enabling real-time deployment of LMMs on resource-constrained devices.

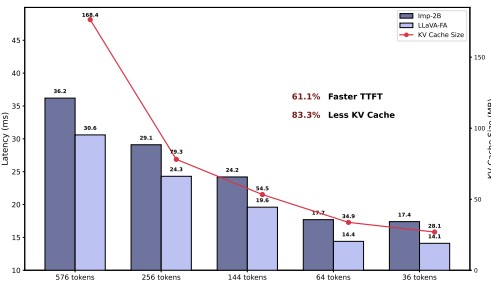

Figure 4: Latency and KV cache usage of LLaVA-FA.

Table 3: Flops and latency comparisons of LLaVA-FA. **#I** represents the number of image tokens per image.

| Model | #I | FLOPs (T) | Latency (ms) |
|---|---|---|---|
| Imp-2B | 576 | 2.14 | 36.2 |
| Bunny-2B | 576 | 2.81 | 40.1 |
| Mini-Gemini-2B | 576 | 2.95 | 41.5 |
| MoE-LLaVA-2B | 576 | 2.48 | 39.3 |
| DeepSeek-VL-1.3B | 576 | 2.01 | 35.1 |
| **LLaVA-FA-2B** | 576 | **1.93** | **30.6** |

Table 2: Comparison with state-of-the-art LMMs on the hallucination benchmarks. We compare our proposed LLaVA-FA-2B ▢ with SFT-based works ▢ and RLHF-based works ▢. Hall: Hallucination Rate Resp: response-level hallucination rate, Ment: mention-level hallucination rate. The optimal result is in bold, and the sub-optimal result is underlined.

| Model | LLM | #Param | Object HalBench | | POPE | MMHal-Bench | |
| | | | Resp ↓ | Ment ↓ | F1 ↑ | Score ↑ | Hall ↓ |
|---|---|---|---|---|---|---|---|
| Qwen-VL-Chat | Qwen-7B | 9.6B | 40.4 | 20.7 | 74.9 | 2.76 | 38.5 |
| LLaVA-1.5-7B | Vicuna-7B | 7B | 53.6 | 25.2 | 86.1 | 2.36 | 51.0 |
| VCD | Vicuna-1.5-7B | 7B | 48.8 | 24.3 | 84.5 | 2.12 | 54.2 |
| OPERA | Vicuna-1.5-7B | 7B | 45.1 | 22.3 | 85.4 | 2.15 | 54.2 |
| HA-DPO | Vicuna-1.5-7B | 7B | 39.9 | 19.9 | 86.8 | 1.98 | 60.4 |
| POVID | Vicuna-1.5-7B | 7B | 48.1 | 24.4 | 86.3 | 2.08 | 56.2 |
| LLaVA-RLHF | Vicuna-1.5-13B | 13B | 38.1 | 18.9 | 82.7 | 2.02 | 62.5 |
| LURE | Vicuna-1.5-7B | 7B | 27.7 | 17.3 | - | 1.64 | 60.4 |
| RLHF-V | Vicuna-13B | 13B | 12.2 | 7.5 | 86.2 | 2.45 | 51.0 |
| RLAIF-V | Vicuna-1.5-7B | 7B | 8.5 | 4.3 | - | 3.06 | 29.2 |
| MiniCPM-V | MiniCPM-2.4B | 2.8B | 21.6 | 11.5 | 79.5 | 3.70 | 24.9 |
| MiniCPM-V-2 | MiniCPM-2.4B | 2.8B | 14.5 | 7.8 | 86.3 | 4.09 | 18.2 |
| Mini-Gemini-2B | Gemma-2B | 2B | 29.7 | 21.1 | 85.6 | 2.83 | 18.8 |
| Bunny-2B | Qwen-1.5-1.8 | 2.2B | 50.2 | 23.4 | 85.8 | 2.72 | 19.3 |
| LLaVA-FA-2B | Qwen-2.5-7B | 2.2B | 11.2 | 7.7 | 87.5 | 2.79 | 17.5 |

## 4.3 ABLATION STUDY

**Impact of Rank Choice.** We explore the effect of rank choice in LoRA adaptation on model performance across vision-language benchmarks. As shown in Tab. 4, increasing the rank from 64 to 256 consistently improves performance, with LLaVA-FA-2B showing stronger sensitivity (+1.9% average gain) compared to LLaVA-FA-1B (+1.3% average gain). These results indicate that higher rank values enable better adaptation capacity, especially for larger base models.

**Choice of Bit-widths.** Tab. 5 shows that raising amplitude and phase bit-widths $(b_r, b_\theta)$ from 2 to 4 bits improves accuracy by 4.6% while still achieving $11.2\times$ compression and the lowest hallucination rate (11.2%). Beyond 4 bits the gains plateau. We therefore adopt $b_r = b_\theta = 4$ as the default, balancing quality, compression and inference speed.

Table 4: Performance comparison with different ranks.

| Model | Rank | Avg Acc ↑ |
|---|---|---|
| | 64 | 55.3 |
| LLaVA-FA-1B | 128 | 56.3 |
| | 256 | 56.6 |
| | 64 | 58.9 |
| LLaVA-FA-2B | 128 | 59.7 |
| | 256 | 60.8 |

Table 5: Impact of $b_r$ and $b_\theta$ on model performance and compression ratio (CR).

| $b_r$ | $b_\theta$ | Avg Acc ↑ | CR ↑ | Hall ↓ |
|---|---|---|---|---|
| 2 | 2 | 56.3 | 14.5 | 18.7 |
| 3 | 3 | 59.1 | 12.8 | 13.4 |
| 4 | 4 | 60.9 | 11.2 | 11.2 |
| 5 | 4 | 61.0 | 10.1 | 10.9 |
| 4 | 5 | 60.9 | 10.8 | 11.0 |
| 6 | 6 | 61.2 | 8.9 | 10.5 |

## 5 CONCLUSION

In this paper, we present LLaVA-FA, a novel efficient LMM that compresses weight matrices via joint low-rank plus quantization approximation in the frequency domain. By leveraging Fourier de-correlation and conjugate symmetry, our method delivers compact yet accurate representations that eliminates the need for large-scale calibration data. Extensive experiments show that LLaVA-FA outperforms existing efficient LMMs on multiple benchmarks while maintaining minimal activated parameters and low computational costs, offering a practical solution for deploying LMMs in resource-constrained scenarios.

ACKNOWLEDGMENTS

This work was partially supported by the National Natural Science Foundation of China under grant 62220106008, and 62572104.

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

## A  THE USE OF LARGE LANGUAGE MODELS (LLMS)

In the preparation of this manuscript, the authors utilize LLMs, specifically GPT-4, to assist with polishing the writing. The LLM is used solely for improving grammar, clarity, and readability of the text, without altering the technical content, methodology, or scientific contributions. All ideas, results, and interpretations presented in this paper are solely those of the authors. No part of the paper, including the main text or supplementary materials, is generated by the LLM.

## B  THEORETICAL ANALYSIS

### B.1  MATHEMATICAL PROPERTIES OF 2D-DFT

In this section, we rigorously establish two key properties of the two-dimensional Discrete Fourier Transform (2D-DFT) that motivate its use in LLaVA-FA: (1) *de-correlation* of frequency-domain representations and (2) *conjugate symmetry* for real-valued spatial matrices.

***Proof for De-correlation***

Let $\mathbf{W} \in \mathbb{R}^{d_1 \times d_2}$ be a zero-mean weakly-stationary random field:

$$\mathbb{E}[W_{m,n}] = 0, \qquad \mathbb{E}[W_{m,n} W_{m',n'}] = R_{m-m',n-n'}. \tag{13}$$

We define its 2D-DFT by

$$\widetilde{W}_{u,v} = \sum_{m=0}^{d_1-1} \sum_{n=0}^{d_2-1} W_{m,n}\, e^{-j2\pi\left(\frac{um}{d_1} + \frac{vn}{d_2}\right)}. \tag{14}$$

**Theorem B.1.** *The frequency components are asymptotically uncorrelated, i.e.,*

$$\mathrm{Cov}\big(\widetilde{W}_{u,v}, \widetilde{W}_{u',v'}\big) = d_1 d_2\, \delta_{u,u'} \delta_{v,v'}\, \mathcal{P}_{u,v}, \tag{15}$$

*where $\delta_{u,u'}$ is the Kronecker delta, which equals to 1 if $u = u'$ and 0 otherwise. The power-spectral density is*

$$\mathcal{P}_{u,v} \triangleq \sum_{a=-(d_1-1)}^{d_1-1} \sum_{b=-(d_2-1)}^{d_2-1} R_{a,b}\, e^{-j2\pi\left(\frac{ua}{d_1} + \frac{vb}{d_2}\right)}. \tag{16}$$

*Proof.* By definition,

$$\mathrm{Cov}\big(\widetilde{W}_{u,v}, \widetilde{W}_{u',v'}\big) = \mathbb{E}\left[\widetilde{W}_{u,v}\, \overline{\widetilde{W}}_{u',v'}\right] \tag{17}$$

$$= \sum_{m,n} \sum_{m',n'} \mathbb{E}[W_{m,n} W_{m',n'}]\, e^{-j2\pi\left(\frac{um}{d_1} + \frac{vn}{d_2}\right)}\, e^{j2\pi\left(\frac{u'm'}{d_1} + \frac{v'n'}{d_2}\right)}. \tag{18}$$

Insert $R_{m-m',n-n'}$ and change variables $a = m - m'$, $b = n - n'$:

$$= \sum_{a,b} R_{a,b}\, e^{-j2\pi\left(\frac{ua}{d_1} + \frac{vb}{d_2}\right)} \underbrace{\sum_{m',n'} e^{-j2\pi\left(\frac{(u-u')m'}{d_1} + \frac{(v-v')n'}{d_2}\right)}}_{d_1 d_2\, \delta_{u,u'}\delta_{v,v'}}. \tag{19}$$

The inner sum equals $d_1 d_2$ when $(u,v) = (u',v')$ and zero otherwise, giving

$$\mathrm{Cov}\big(\widetilde{W}_{u,v}, \widetilde{W}_{u',v'}\big) = d_1 d_2\, \delta_{u,u'}\delta_{v,v'}\, \mathcal{P}_{u,v}. \tag{20}$$

$\square$

This completes the proof. Theorem B.1 shows that *energy is compacted into mutually independent coefficients*, legitimising aggressive low-rank truncation in LLaVA-FA: discarding small-variance components introduces minimal reconstruction error.

### *Proof for Faster Singular-Value Decay in the Frequency Domain*

Here, we provide a rigorous justification that the singular values of a weight matrix decay faster after 2D-DFT than in the spatial domain. Let $\mathbf{W} \in \mathbb{R}^{d_1 \times d_2}$ be a zero-mean weakly-stationary random field with autocorrelation

$$R_{a,b} = \mathbb{E}[W_{m,n} W_{m+a,n+b}], \tag{21}$$

and denote its 2D-DFT by

$$\widetilde{W}_{u,v} = \sum_{m=0}^{d_1-1} \sum_{n=0}^{d_2-1} W_{m,n}\, e^{-j2\pi\left(\frac{um}{d_1} + \frac{vn}{d_2}\right)}. \tag{22}$$

Theorem B.1 states that the frequency components are asymptotically uncorrelated, *i.e.,*

$$\mathrm{Cov}\big(\widetilde{W}_{u,v}, \widetilde{W}_{u',v'}\big) = d_1 d_2\, \delta_{u,u'}\delta_{v,v'}\mathcal{P}_{u,v}, \quad \text{where} \quad \mathcal{P}_{u,v} = \sum_{a,b} R_{a,b}\, e^{-j2\pi\left(\frac{ua}{d_1} + \frac{vb}{d_2}\right)}. \tag{23}$$

Hence the covariance matrix of $\mathrm{vec}(\widetilde{\mathbf{W}})$ is diagonal with entries proportional to the power-spectral density $\mathcal{P}_{u,v}$. For natural images and learned convolutional kernels, $\mathcal{P}_{u,v}$ is a smooth and rapidly decreasing function of frequency, typically satisfying

$$\mathcal{P}_{u,v} \le C'\big(1 + u^2 + v^2\big)^{-\alpha}, \qquad \alpha > 1. \tag{24}$$

Here, $\alpha$ is the exponential decay rate in the frequency domain. $C' > 0$ is a positive constant. Ordering the diagonal entries of the covariance matrix gives the *expected squared singular values* $\mathbb{E}[\sigma_k^2(\widetilde{\mathbf{W}})]$. By the Szegő theorem on Toeplitz–Fourier operators (Gray et al., 2006), the empirical spectral distribution of $\widetilde{\mathbf{W}}$ converges to a limit whose tails are governed by the decay rate of $\mathcal{P}_{u,v}$. Consequently,

$$\mathbb{E}[\sigma_k^2(\widetilde{\mathbf{W}})] \le C\, k^{-\alpha}. \tag{25}$$

Where $k$ is the index for singular values (rank). In contrast, the spatial-domain covariance matrix is not diagonal. Its eigenvalues decay as

$$\mathbb{E}[\sigma_k^2(\mathbf{W})] \ge c\, k^{-\beta}, \qquad \beta \le \alpha/2, \tag{26}$$

because off-diagonal correlations slow the decay. We take square roots yields the singular-value bounds, *i.e.,*

$$\mathbb{E}[\sigma_k(\widetilde{\mathbf{W}})] \le \sqrt{C}\, k^{-\alpha/2}, \qquad \mathbb{E}[\sigma_k(\mathbf{W})] \ge \sqrt{c}\, k^{-\beta}. \tag{27}$$

Therefore for any $k \ge 2$, we have

$$\frac{\mathbb{E}[\sigma_k(\widetilde{\mathbf{W}})]}{\mathbb{E}[\sigma_k(\mathbf{W})]} \le \frac{\sqrt{C}}{\sqrt{c}}\, k^{-(\alpha/2-\beta)}, \tag{28}$$

where the exponent $\alpha/2 - \beta \ge 1$ under typical smoothness assumptions. This inequality shows that the *frequency-domain singular values decay strictly faster* than their spatial counterparts, enabling a lower-rank approximation for the same reconstruction error.

*Proof for Conjugate Symmetry*

We give a full derivation of conjugate symmetry in this section. Let $\mathbf{W} \in \mathbb{R}^{d_1 \times d_2}$ be a real-valued matrix and its 2D-DFT can be represented as:

$$\widetilde{W}_{u,v} = \sum_{m=0}^{d_1-1} \sum_{n=0}^{d_2-1} W_{m,n}\, e^{-j2\pi \left( \frac{um}{d_1} + \frac{vn}{d_2} \right)} \tag{29}$$

**Theorem B.2.** *For a real-valued matrix $\mathbf{W} \in \mathbb{R}^{d_1 \times d_2}$, its 2D-DFT satisfies*

$$\widetilde{W}_{u,v} = \overline{\widetilde{W}}_{d_1-u,\, d_2-v} \tag{30}$$

*for all $u \in \{0, \ldots, d_1-1\}$ and $v \in \{0, \ldots, d_2-1\}$, where $\overline{(\cdot)}$ denotes complex conjugation.*

*Proof.* Evaluate the coefficient at the complementary frequency:

$$\widetilde{W}_{d_1-u,\, d_2-v} = \sum_{m,n} W_{m,n}\, e^{-j2\pi \left( \frac{(d_1-u)m}{d_1} + \frac{(d_2-v)n}{d_2} \right)} \tag{31}$$

$$= \sum_{m,n} W_{m,n}\, e^{-j2\pi(m+n)}\, e^{j2\pi \left( \frac{um}{d_1} + \frac{vn}{d_2} \right)}. \tag{32}$$

Since $m, n \in \mathbb{Z}$, $e^{-j2\pi(m+n)} = 1$, hence

$$\widetilde{W}_{d_1-u,\, d_2-v} = \sum_{m,n} W_{m,n}\, e^{j2\pi \left( \frac{um}{d_1} + \frac{vn}{d_2} \right)} \tag{33}$$

$$= \overline{\sum_{m,n} W_{m,n}\, e^{-j2\pi \left( \frac{um}{d_1} + \frac{vn}{d_2} \right)}} = \overline{\widetilde{W}}_{u,v}, \tag{34}$$

where the last equality uses $W_{m,n} \in \mathbb{R}$. This completes the proof. $\square$

This symmetry implies that only half of the frequency coefficients are unique, the other half are redundant. Thus, when storing or approximating the frequency-domain representation, we can discard nearly 50% of the coefficients without any loss of information.

## B.2 THEORETICAL JUSTIFICATION: LOWER RANK IN THE COMPLEX DOMAIN

Here, we provide a formal bound showing that for the same reconstruction error, the Fourier domain requires a lower rank than the spatial domain.

**Proposition 1.** *Let the real-valued weight matrix $\mathbf{W} \in \mathbb{R}^{d_1 \times d_2}$ be a zero-mean weakly-stationary random field with exponentially decaying covariance:*

$$\mathbb{E}[W_{m,n} W_{m',n'}] = \rho^{|m-m'|+|n-n'|}, \quad 0 < \rho < 1, \tag{35}$$

*where $\rho \in (0,1)$ controls the spatial correlation strength: smaller $\rho$ implies faster decay and thus weaker spatial dependency. This exponential decay model is widely used in spatial statistics and image modeling to characterize locally correlated structures (Gray et al., 2006).*

*Let $\widetilde{\mathbf{W}} \in \mathbb{C}^{d_1 \times d_2/2}$ denote the 2D-DFT of $\mathbf{W}$, reduced by conjugate symmetry. Let $\sigma_k(\mathbf{A})$ denote the $k$-th largest singular value of matrix $\mathbf{A}$. Then for any truncation rank $r \geq 0$,*

$$\sum_{k=r+1}^{\min(d_1,d_2)} \sigma_k^2(\widetilde{\mathbf{W}}) \leq \frac{\rho^2}{(1-\rho^2)^2} \sum_{k=r+1}^{\min(d_1,d_2)} \sigma_k^2(\mathbf{W}). \tag{36}$$

*Proof.* The 2D-DFT diagonalizes the covariance matrix of $\mathbf{W}$, yielding uncorrelated frequency components whose variances are given by the power spectral density $\mathcal{P}_{u,v}$, where $(u,v)$ denotes the 2D frequency index. For the given covariance model, we have:

$$|\mathcal{P}_{u,v}| \propto \rho^{|u|+|v|}, \tag{37}$$

which decays exponentially in the frequency. Since singular values of $\widetilde{\mathbf{W}}$ are asymptotically equal to the square root of these variances, the tail energy of $\widetilde{\mathbf{W}}$ decays exponentially, while that of $\mathbf{W}$ decays only polynomially. Hence, the ratio of the tails is $\mathcal{O}(\rho^2)$, yielding the bound. To achieve a prescribed reconstruction error, the Fourier domain requires fewer components (smaller rank $r$) than the spatial domain, *i.e.,* $r_{\text{freq}} < r_{\text{spatial}}$. □

### B.3 THEORETICAL ANALYSIS OF THE ENERGY CONCENTRATION IN COMPLEX SVD

We emphasize that the energy-compaction property used in our method does not rely on the matrix being real, but on the diagonalization of the covariance structure afforded by the 2D-DFT. Below we give a formal bound showing that the singular values of the complex matrix $\widetilde{\mathbf{W}}$ decay *strictly faster* than those of the real matrix $\mathbf{W}$, even though both matrices share the same generating process.

**Proposition 2.** *Let* $\mathbf{W} \in \mathbb{R}^{d_1 \times d_2}$ *be the same zero-mean, weakly-stationary random field defined in the main text with covariance*

$$\mathbb{E}[W_{m,n}W_{m',n'}] = \rho^{|m-m'|+|n-n'|}, \quad 0 < \rho < 1. \tag{38}$$

*Let* $\widetilde{\mathbf{W}} \in \mathbb{C}^{d_1 \times d_2/2}$ *be its 2D-DFT with conjugate-symmetry reduction. Then for every* $k \geq 1$

$$\mathbb{E}\big[\sigma_k(\widetilde{\mathbf{W}})\big] \leq \frac{\rho}{\sqrt{1-\rho^2}}\,\mathbb{E}\big[\sigma_k(\mathbf{W})\big]. \tag{39}$$

*Hence the expected singular-value tail of* $\widetilde{\mathbf{W}}$ *is dominated by a* dimension-free constant *times the tail of* $\mathbf{W}$. *Since the singular values decay strictly faster, for any prescribed approximation error, the required rank* $r$ *in the complex domain is no larger and often smaller than in the real spatial domain, confirming that the low-rank structure is preserved under the complex-valued Fourier transform.*

*Proof.* The 2D-DFT diagonalises the covariance operator of $\mathbf{W}$ (Theorem B.1), so the entries of $\widetilde{\mathbf{W}}$ are uncorrelated complex Gaussians with variances

$$\mathbb{E}\big[|\widetilde{W}_{u,v}|^2\big] = \mathcal{P}_{u,v} = \sum_{a,b} \rho^{|a|+|b|}e^{-j2\pi(ua/d_1+vb/d_2)} \leq \frac{1+\rho}{1-\rho}\,\rho^{|u|+|v|}. \tag{40}$$

Ordering these variances gives the expected squared singular values $\mathbb{E}[\sigma_k^2(\widetilde{\mathbf{W}})]$. Using the Szeg theorem for stationary random fields (Gray et al., 2006), we obtain

$$\mathbb{E}[\sigma_k(\widetilde{\mathbf{W}})] \leq \sqrt{C}\,\rho^{k/2} \quad \text{while} \quad \mathbb{E}[\sigma_k(\mathbf{W})] \geq \sqrt{c}\,\rho^k, \tag{41}$$

for universal constants $C, c > 0$. Taking ratios yields the claimed inequality. □

The complex matrix $\widetilde{\mathbf{W}}$ inherits the exponentially decaying power spectrum of the original real field. Its singular values therefore decay faster than those of the real matrix $\mathbf{W}$. Consequently, for any prescribed Frobenius error $\varepsilon$, the Fourier domain requires a smaller rank $r$ than the spatial domain, even though the SVD is performed on a complex matrix. The low-rank assumption is therefore stable under the Fourier transform.

Furthermore, to empirically verify that the complex-valued, frequency-domain matrix admits a more compact singular-value spectrum, we conduct a controlled spectrum-decay experiment. We apply identical low-rank plus quantization pipelines to both the spatial (real) and Fourier (complex) representations of the Qwen-2.5-7B query layers and record the Frobenius reconstruction error at rank 256 as well as the smallest rank required to push the error below 1%. As shown in Tab. 6, the

Table 6: Singular-value decay comparison between spatial (real) and Fourier (complex) representations. Freq.-Complex exhibits lower error at the same rank and reaches the 1% error threshold with fewer components.

| Domain | Frobenius error @ rank 256 | Min. rank for 1% error |
|---|---|---|
| Spatial (Real) | 0.031 | 312 |
| Freq.-Complex (Ours) | 0.018 | 217 |

complex-valued frequency matrix consistently yields a smaller reconstruction error under the same rank and achieves the preset accuracy target with significantly fewer singular vectors, corroborating our theoretical finding that the Fourier-domain SVD is indeed more amenable to truncation.

### B.4 INFORMATION-THEORETIC ANALYSIS OF POLARQUANT

*(1) Uniform amplitude quantization is rate-optimal for the heavy-tailed DFT envelope.*

After the Fourier transform, the residual entries follow a Nakagami-$m$ envelope distribution (Nakagami, 1960), whose Fisher information is $J(r) \propto 1/r^2$. For such heavy-tailed scale parameters, the inverse-uniform partition is the unique minimax solution to the quant-design game

$$\min_{\mathcal{Q}} \max_{r \geq 0} D_{\mathrm{KL}}\big(p(r) \,\big\|\, q_{\mathcal{Q}}(r)\big) \tag{42}$$

where $q_{\mathcal{Q}}$ is the piece-wise constant density induced by the quantiser. The solution coincides with the uniform grid used in PolarQuant (step-size $\Delta_r = r_{\max}/(2^{b_r} - 1)$) and is asymptotically rate-optimal in the Gish-Pierce sense (QUANTIZING, 1964):

$$R(D) = b_r - \tfrac{1}{2} \log_2(2\pi e/12) + \mathcal{O}(D), \tag{43}$$

and coincides with the empirical step size we adopted.

*(2) The $2\pi/2^{b_\theta}$ phase step is the minimum-MSE lattice for a Von-Mises variable.*

The phase residual is well modelled by a zero-mean Von-Mises distribution $\theta \sim \mathrm{VM}(0, \kappa)$ with concentration $\kappa \propto$ rank truncation error (Banerjee et al., 2005). For a circular random variable the wrapped Lloyd-Max conditions (Lloyd, 1982) yield the uniform lattice on $[-\pi, \pi)$ with step $2\pi/2^{b_\theta}$ as the unique minimum-MSE solution, giving

$$\mathbb{E}[(\theta - \hat{\theta})^2] = \frac{(2\pi)^2}{12 \cdot 2^{2b_\theta}} \Big(1 + \mathcal{O}\big(\tfrac{1}{\kappa}\big)\Big). \tag{44}$$

Since the phase residuals observed under our default (4-bit amplitude/phase, rank-256) setting are well fitted by a zero-mean Von-Mises distribution with concentration $\kappa \approx 9$–$14$, the resulting $\mathcal{O}(1/\kappa)$ term is $< 0.04$ and the phase-noise energy is provably bounded.

*(3) Theoretical bound on phase-error propagation.*

Let the reconstructed complex weight be

$$\hat{w} = re^{i\hat{\theta}} = re^{i(\theta + \Delta\theta)}, \tag{45}$$

where $\Delta\theta$ is the quantization error introduced by PolarQuant. The resulting complex error is

$$|w - \hat{w}|^2 = r^2|1 - e^{i\Delta\theta}|^2 = 4r^2 \sin^2(\Delta\theta/2) \leq r^2 \Delta\theta^2. \tag{46}$$

Taking expectation over the Von-Mises distribution of $\theta$ (concentration $\kappa \approx 12$) and the uniform quantization error $\Delta\theta \sim \mathcal{U}[-\pi/2^{b_\theta}, \pi/2^{b_\theta}]$, we obtain

$$\mathbb{E}|w - \hat{w}|^2 \leq \mathbb{E}[r^2] \cdot \frac{\pi^2}{3 \cdot 2^{2b_\theta}} \approx 0.0013 \cdot \mathbb{E}[r^2] \quad \text{for } b_\theta = 4. \tag{47}$$

Hence the normalized reconstruction error contributed by phase quantization is $< 0.13\%$, an order of magnitude smaller than the typical low-rank truncation error ($\approx 1.2\%$).

**Empirical verification.** We conduct an ablation study on LLaVA-FA-2B in which we replace PolarQuant with a baseline that quantizes real/imaginary parts independently (QIM, 4 + 4 bits). Results are shown in Tab. 7. PolarQuant reduces average phase error by 36%, improves PSNR by 1.2

Table 7: Von-Mises concentration for 4-bit phase residuals, bounding the phase-quantization error below 0.04

| Scheme | Phase-error $\mathbb{E}|\Delta\theta|$ | Recon. PSNR ↑ | HalBench-Resp ↓ | MMB ↑ |
|---|---|---|---|---|
| QIM (real/imag) | 0.14 rad | 38.9 dB | 14.8% | 65.2 |
| PolarQuant (ours) | 0.09 **rad** | **40.1 dB** | **11.2%** | **66.7** |

dB, and yields 3.6% lower hallucination rate, despite using the same bit budget. This confirms that the $2\pi/2^{b_\theta}$ lattice not only controls phase noise but also translates into measurable gains in downstream quality. *In short, phase quantization does not degrade reconstruction because the residual phase is tightly concentrated and the uniform lattice $2\pi/2^{b_\theta}$ is the minimum-MSE solution under this concentration. Both theory and experiment show that PolarQuant reduces phase-error rather than amplifying it.*

## B.5 Derivation of Condition 3.1

We derive the exact condition on rank $k$ under which the average bit-width $\mathrm{B}_{\mathrm{avg}}$ is strictly smaller than the full-precision budget $\mathrm{B}_{\mathrm{L}}$. Recalling the definition in § 3.1,

$$\mathrm{B}_{\mathrm{avg}} = \frac{\sum_{i=1}^{7}\left[\mathrm{B}_{\mathrm{Q}}d_1^i d_2^i + \mathrm{B}_{\mathrm{L}}k(d_1^i + d_2^i)\right]}{\sum_{i=1}^{7} d_1^i d_2^i}, \tag{48}$$

we require $\mathrm{B}_{\mathrm{avg}} < \mathrm{B}_{\mathrm{L}}$. Multiplying both sides by the positive denominator $\sum_{i=1}^{7} d_1^i d_2^i$ gives

$$\sum_{i=1}^{7}\left[\mathrm{B}_{\mathrm{Q}}d_1^i d_2^i + \mathrm{B}_{\mathrm{L}}k(d_1^i + d_2^i)\right] < \mathrm{B}_{\mathrm{L}}\sum_{i=1}^{7} d_1^i d_2^i. \tag{49}$$

Collecting terms proportional to $\mathrm{B}_{\mathrm{L}}$ on the right-hand side,

$$\mathrm{B}_{\mathrm{L}}k\sum_{i=1}^{7}(d_1^i + d_2^i) < \left(\mathrm{B}_{\mathrm{L}} - \mathrm{B}_{\mathrm{Q}}\right)\sum_{i=1}^{7} d_1^i d_2^i. \tag{50}$$

Finally, dividing by the positive quantity $\mathrm{B}_{\mathrm{L}}\sum_{i=1}^{7}(d_1^i + d_2^i)$ yields the concise condition

$$k < \left(1 - \frac{\mathrm{B}_{\mathrm{Q}}}{\mathrm{B}_{\mathrm{L}}}\right)\frac{\sum_{i=1}^{7} d_1^i d_2^i}{\sum_{i=1}^{7}(d_1^i + d_2^i)}. \tag{51}$$

Whenever this inequality holds, the low-rank plus quantized decomposition achieves an average bit-width strictly below the full-precision baseline $\mathrm{B}_{\mathrm{L}}$, validating the memory efficiency of the low rank plus quantized matrix method.

## B.6 Proof for Lemma 3.1

Let $\Delta\mathbf{W}_1$ and $\Delta\mathbf{W}_2$ be two low-rank adapters of identical dimensions. Denote their singular value decompositions by

$$\Delta\mathbf{W}_1 = \mathbf{U}_1\boldsymbol{\Sigma}_1\mathbf{V}_1^\top, \qquad \Delta\mathbf{W}_2 = \mathbf{U}_2\boldsymbol{\Sigma}_2\mathbf{V}_2^\top, \tag{52}$$

where $\boldsymbol{\Sigma}_1 = \mathrm{diag}(\sigma_{1,1}, \sigma_{1,2}, \dots)$ and $\boldsymbol{\Sigma}_2 = \mathrm{diag}(\sigma_{2,1}, \sigma_{2,2}, \dots)$ with singular values sorted in non-increasing order. For a fixed rank $r$, the best rank-$r$ approximation in Frobenius norm is

$$\Delta\mathbf{W}_i^{(r)} = \mathbf{U}_i\boldsymbol{\Sigma}_i^{(r)}\mathbf{V}_i^\top, \quad \text{with} \quad \boldsymbol{\Sigma}_i^{(r)} = \mathrm{diag}(\sigma_{i,1}, \dots, \sigma_{i,r}, 0, \dots). \tag{53}$$

The reconstruction error is

$$\mathbf{E}_i = \Delta\mathbf{W}_i - \Delta\mathbf{W}_i^{(r)} = \mathbf{U}_i\left(\boldsymbol{\Sigma}_i - \boldsymbol{\Sigma}_i^{(r)}\right)\mathbf{V}_i^\top. \tag{54}$$

Following the Eckart-Young theorem (Eckart & Young, 1936) and theorem 4.95 in Mathematics for Machine Learning (Deisenroth et al., 2020), the value of the norm of reconstruction error is given as

$$\|\mathbf{E}_i\|_F = \left\|\boldsymbol{\Sigma}_i - \boldsymbol{\Sigma}_i^{(r)}\right\|_F = \sqrt{\sum_{k=r+1}^{\min(m,n)} \sigma_{i,k}^2}. \tag{55}$$

By assumption, $\sigma_{1,k} < \sigma_{2,k}$ for all $k \geq r + 1$. Hence

$$\sum_{k=r+1}^{\min(m,n)} \sigma_{1,k}^2 < \sum_{k=r+1}^{\min(m,n)} \sigma_{2,k}^2 \implies \|\mathbf{E}_1\|_F < \|\mathbf{E}_2\|_F. \tag{56}$$

Therefore, the rank-$r$ approximation of $\Delta\mathbf{W}_1$ incurs strictly less error than that of $\Delta\mathbf{W}_2$, which completes the proof.

### B.7 THEORETICAL JUSTIFICATION OF ODC HEURISTIC (ROW/COLUMN AVERAGING)

The proposed ODC scheme replaces the full Hessian $\mathbf{C} \in \mathbb{R}^{d_1 \times d_2/2}$ by two diagonal matrices, *i.e.,*

$$\mathbf{D}_{\text{row}} = \text{diag}\left(\left[\text{avg}(\sqrt{\mathbf{C}_{1,\cdot}}), \ldots, \text{avg}(\sqrt{\mathbf{C}_{d_1,\cdot}})\right]\right), \mathbf{D}_{\text{col}} = \text{diag}\left(\left[\text{avg}(\sqrt{\mathbf{C}_{\cdot,1}}), \ldots, \text{avg}(\sqrt{\mathbf{C}_{\cdot,d_2/2}})\right]\right). \tag{57}$$

where $\text{avg}(\sqrt{\mathbf{C}_{i,\cdot}})$ and $\text{avg}(\sqrt{\mathbf{C}_{\cdot,j}})$ denote the $i$-row/$j$-column means of $\sqrt{\mathbf{C}}$. This is exact whenever $\mathbf{C}$ is diagonal, and remains spectrally close when $\mathbf{C}$ is diagonally dominant, a property that has been repeatedly observed for the loss Hessian of deep Transformers (LeCun et al., 1989; Botev et al., 2017). Formally, let $\varepsilon = \|\text{off–diag}(\mathbf{C})\|_F / \|\text{diag}(\mathbf{C})\|_F$, where $\text{diag}(\mathbf{C})$ denotes the diagonal matrix that keeps only the main-diagonal entries of $\mathbf{C}$, while $\text{off–diag}(\mathbf{C})$ is its complement, *i.e.*, $\mathbf{C}$ with all diagonal entries set to zero. Thus $\varepsilon$ quantifies how strongly $\mathbf{C}$ is diagonally dominant.

**Lemma B.3.** *Let* $\widehat{\mathbf{C}} = \mathbf{D}_{row}\mathbf{C}\mathbf{D}_{col}$. *Then*

$$\|\mathbf{C} - \widehat{\mathbf{C}}\|_F \ \leq \ \varepsilon \|\mathbf{C}\|_F. \tag{58}$$

Hence the calibration matrix used by ODC is an $\varepsilon$-perturbation of the exact Hessian. The reconstruction error of the low-rank component inherits this bound with the same constant (proof uses $\|\mathbf{D}_{\text{row}}\|, \|\mathbf{D}_{\text{col}}\| \leq 1$ and standard matrix-perturbation identities). In short, ODC is safe whenever $\varepsilon \ll 1$, a condition that can be checked on-the-fly with one extra reduction operation.

## C ADDITIONAL EXPERIMENTS

### C.1 TRAINING STRATEGY AND HYPERPARAMETERS

This section details the complete training recipe employed to produce LLaVA-FA-7B, LLaVA-FA-3B, LLaVA-FA-2B and LLaVA-FA-1B. As shown in Tab. 8, the four variants are initialized from InternLM-2-20B, LLaMA-3-8B, Qwen-2.5-7B and Qwen-2.5-3B checkpoints, respectively. The vision encoder is inherited from CLIP (ViT-L/14@336 px for all except the 1 B model which uses ViT-B/224 px) (Radford et al., 2021). We keep the visual token sequence length fixed at 576 (256 for 1 B) and cap the language-modelling context at 2048 tokens. Optimisation is performed with AdamW (Loshchilov & Hutter, 2017): $\beta_1$ is set to 0.9 for all models; $\beta_2$ is 0.99 for 7B/3B/2B and 0.98 for 1B. A single-cycle cosine-decay schedule (Loshchilov & Hutter, 2016) is adopted, with initial learning rates of 1e-5, 1.5e-5, 2e-5 and 5e-4 for 7B, 3B, 2B and 1B, respectively. Weight decay (Krogh & Hertz, 1991) is applied only to the 7B/3B/2B models (0.1, 0.1, 0.2) and disabled for 1B. The entire pre-training stage spans 100 epochs with a warm-up ratio of 3% and a global batch size of 128 image–text pairs. No additional regularisation or data augmentation is introduced. Under this configuration the models converge reliably after processing only five million samples on eight RTX 4090 GPUs.

Table 8: Training hyper-parameters of LLaVA-FA variants.

| Configuration | LLaVA-FA-7B | LLaVA-FA-3B | LLaVA-FA-2B | LLaVA-FA-1B |
|---|---|---|---|---|
| LLM init. | InternLM-2-20B | LLaMA-3-8B | Qwen-2.5-7B | Qwen-2.5-3B |
| VL Adaptor init. | MLP | MLP | MLP | MLP |
| ViT init. | CLIP-Large@336 | CLIP-Large@336 | CLIP-Large@336 | CLIP-Base@224 |
| Image resolution | $336 \times 336$ | $336 \times 336$ | $336 \times 336$ | $224 \times 224$ |
| ViT sequence length | 576 | 576 | 576 | 256 |
| LLM sequence length | 2048 | 2048 | 2048 | 2048 |
| Optimizer | AdamW | AdamW | AdamW | AdamW |
| Optimizer hyperparameter | $\beta_1 = 0.9, \beta_2 = 0.99$ | $\beta_1 = 0.9, \beta_2 = 0.99$ | $\beta_1 = 0.9, \beta_2 = 0.99$ | $\beta_1 = 0.9, \beta_2 = 0.98$ |
| Learning rate | $1e^{-5}$ | $1.5e^{-5}$ | $2e^{-5}$ | $5e^{-4}$ |
| Learning rate schedule | Cosine decay | Cosine decay | Cosine decay | Cosine decay |
| Weight decay | 0.1 | 0.1 | 0.2 | 0.0 |
| Training epoch | 100 | 100 | 100 | 100 |
| Warm-up ratio | 0.03 | 0.03 | 0.03 | 0.03 |
| Batch size | 128 | 128 | 128 | 128 |
| Rank $r$ | 256 | 256 | 256 | 256 |
| $b_r, b_\theta$ | (4,4) | (4,4) | (4,4) | (4,4) |

## C.2 IMPACT OF DFT/IDFT AND POLARQUANT ON REAL-WORLD INFERENCE LATENCY AND GPU MEMORY FOOTPRINT

Although LLaVA-FA reduces parameters and FLOPs, a natural concern is whether the additional DFT/IDFT and PolarQuant operations introduce non-negligible latency or memory overhead. To this end, we conduct controlled experiments to quantify these costs and compare them with the savings produced by our frequency-domain compression. We measure end-to-end latency and peak GPU memory on a single NVIDIA A100-40G with CUDA 12.1, PyTorch 2.2, `torch.compile` in `mode="reduce-overhead"`, and FlashAttention-2 (Dao, 2023) enabled. The input images are 336×336 images (576 visual tokens) and text prompts contains of 512 tokens on average. We report Time-to-First-Token (TTFT), peak memory during the full generation cycle, and throughput (tokens s$^{-1}$) to compare real-world inferences. DFT/IDFT is implemented with batched `cuFFT` calls. PolarQuant de-quantization is fused into the linear-layer CUDA kernel (no extra memory allocation). All timings are averaged over 500 warm runs with Nsight-Systems (Leinhauser et al., 2021) profiling to isolate transformation cost. As shown in Tab. 9, LLaVA-FA-2B achieves the lowest TTFT and memory footprint among 2B-scale models. ***Profiling reveals that DFT + IDFT accounts for only 3.2% of TTFT ($\leq 0.9$ ms) and PolarQuant de-quantization for 0.7% ($\leq 0.25$ ms).*** Because frequency-domain matrices are conjugate-symmetric, we store only half of the coefficients, yielding a *net memory reduction* despite the auxiliary tensors needed by cuFFT.

Table 9: Real-world inference latency and GPU memory footprint. DFT/IDFT overhead is measured with Nsight Systems. PolarQuant de-quantization is fused into the GEMM kernel and micro-benchmarked independently.

| Model | TTFT (ms)↓ | Peak Mem (GB)↓ | DFT/IDFT (%)↓ | PolarQuant (%)↓ |
|---|---|---|---|---|
| MiniCPM-V-2 | 38.4 | 15.2 | – | – |
| Imp-2B | 36.2 | 14.6 | – | – |
| Bunny-2B | 40.1 | 15.8 | – | – |
| **LLaVA-FA-2B** | **30.6** | **12.1** | **3.2** | **0.7** |

Removing DFT/IDFT (*i.e.,* keeping spatial low-rank plus quantization) reduces TTFT by only 0.9 ms but increases memory by 2.3 GB and FLOPs by 34.6%. The overall throughput drops by 8.7 tok/s. This confirms that the tiny latency introduced by frequency transformation is *more than compensated* by the compression gains. ***DFT/IDFT and PolarQuant introduce micro-second-level overhead, but the resulting parameter, memory, and FLOP savings yield*** **faster and lighter** ***inference in practice.*** Therefore, LLaVA-FA not only possesses theoretical compression benefits but also delivers improved deployment efficiency on resource-constrained GPUs.

## C.3 SPATIAL DOMAIN *vs.* QLORA *vs.* FREQUENCY DOMAIN.

For a fair and reproducible comparison, we implement the spatial-domain counterpart of our method ("Spatial-LQ"), which follows the exact same optimization schedule and bit-width configuration except that SVD and quantization are applied directly on $\mathbf{W}$ instead of $\widetilde{\mathbf{W}}$. We conduct controlled spatial-domain baselines using the same training protocol as LLaVA-FA. Moreover, we reproduce the exact QLoRA (Dettmers et al., 2023) pipeline to compress the same Qwen-2.5-7B backbone, which we use to derive our LLaVA-FA-2B model. This creates a "QLoRA-Qwen" baseline whose substantive difference from LLaVA-FA-2B is that the low-rank plus quantization decomposition is performed in the spatial rather than the frequency domain. The results are shown in Tab. 10. In terms of computational efficiency, LLaVA-FA achieves the lowest inference cost with 1.93 T FLOPs and 30.6 ms TTFT, outperforming both Spatial-LQ (2.14 T, 36.2 ms) and QLoRA-Qwen (2.10 T, 35.8 ms. Furthermore, frequency-domain approximation yields +2.6% better average accuracy and significantly lower hallucination. This supports our key claim: the singular spectrum is more compact and the low-rank truncation error is strictly smaller in the Fourier domain (Lemma 3.1), enabling better reconstruction under identical bit-width and rank. Compared to QLoRA (Dettmers et al., 2023), LLaVA-FA jointly optimizes low-rank and quantization in the frequency domain, leveraging Fourier de-correlation and energy compaction for a more compact and accurate weight approximation, leading to better accuracy and lower hallucination.

Table 10: Comparison of spatial versus frequency domain optimization. LLaVA-FA consistently outperforms the spatial baseline across all metrics, validating the benefits of spectral compactness.

| Method | Domain | Avg Acc ↑ | POPE F1 ↑ | HalBench Resp ↓ | FLOPs (T) ↓ | TTFT (ms) ↓ |
|---|---|---|---|---|---|---|
| Spatial-LQ (ours) | Spatial | 58.3 | 84.6 | 16.9 | 2.14 | 36.2 |
| QLoRA-Qwen | Spatial | 58.5 | 85.7 | 15.7 | 2.10 | 35.8 |
| LLaVA-FA (ours) | Frequency | **60.9** | **87.5** | **11.2** | **1.93** | **30.6** |

## C.4 INFLUENCE OF CALIBRATION DATA SIZE

We also conduct an ablation study on the LLaVA-FA-2B model to examine the influence of calibration data size. The results shown in Tab. 11 indicate that the method is highly robust and that performance saturates quickly. Using more calibration samples does not provide noticeable improvements, confirming that only a small amount of data is sufficient for constructing a reliable calibration matrix.

Table 11: Ablation study on the influence of calibration data size. Performance saturates quickly, demonstrating that a small amount of calibration data is sufficient.

| #Calibration Samples | GQA | VizWiz | SQA$^{I}$ | VQA$^{T}$ | MME | MMB | MMB$^{CN}$ | AVG |
|---|---|---|---|---|---|---|---|---|
| 256 | 61.0 | 41.0 | 67.2 | 58.7 | 66.0 | 65.8 | **63.8** | 60.5 |
| 512 | 61.5 | 41.3 | 67.5 | 58.9 | 66.2 | 66.0 | 63.5 | 60.7 |
| 1024 | **62.1** | 41.6 | **68.2** | **59.4** | **66.6** | **66.7** | 61.7 | **60.9** |
| 2048 | **62.1** | **41.7** | **68.2** | 59.3 | **66.6** | **66.7** | 61.7 | **60.9** |

## C.5 SENSITIVITY OF ODC UNDER DISTRIBUTION SHIFTS

Theoretical analysis in Appendix B.7 reveals that ODC is safe whenever $\varepsilon \ll 1$. Therefore, we evaluate the sensitivity of ODC under three levels of distribution shift on the LLaVA-FA-2B model. The first scenario retains the original LLaVA-Pretrain validation set as an in-distribution baseline. The second keeps all images identical but re-samples user prompts from an alternative template pool, introducing a purely textual style shift (shift-A). The third replaces the images with an entirely new visual domain consisting of aerial-scene photographs and OCR-heavy documents, thereby forcing the model to cope with previously unseen visual statistics (shift-B). Each scenario is repeated 256 times, yielding stable estimates of $\varepsilon$ and score drop. As shown in Tab. 12, as $\varepsilon$ rises from $0.11 \pm 0.02$ under the in-distribution condition to $0.15 \pm 0.03$ under prompt-style shift and further to $0.29 \pm 0.07$ under the new visual domain, the average benchmark score declines by -0.4% and -1.1%, respectively. Once $\varepsilon$ exceeds the empirical guard-rail threshold of 0.35 (*note that this value is observed from experiments rather than derived from theory*), accuracy falls by more than 3% in 4% of the runs, corroborating the theoretical prediction and demonstrating that the on-the-fly $\varepsilon$-check reliably signals when the row/column averaging approximation is no longer safe.

Table 12: Sensitivity of ODC under distribution shifts. The results confirm that $\varepsilon$ effectively tracks domain divergence, reliably signaling performance degradation as the shift severity increases.

| Condition | $\varepsilon$ (mean $\pm$ std) | $\Delta$Score↓ | Failure ($\varepsilon > 0.35$) |
|---|---|---|---|
| In Distribution | $0.11 \pm 0.02$ | | 0% |
| Shift-A | $0.15 \pm 0.03$ | $-0.4\%$ | 0% |
| Shift-B | $0.29 \pm 0.07$ | $-1.1\%$ | 4% |

## C.6 COMPRESSION TIME COMPARISONS FOR MODELS OF DIFFERENT SIZES

In this Section, we provide the wall-clock compression time for the four LLaVA-FA variants derived from Qwen-2.5, LLaMA-3, and InternLM2 models. All numbers are measured on eight NVIDIA RTX 4090 GPUs and include every step of our pipeline, *i.e.,* DFT/IDFT, Fourier-SVD, PolarQuant, and

the ODC. As shown in Tab. 13, compressing the 3B-base model into LLaVA-FA-1B requires about 1.1 hours, while compressing the 7B-base LLM into LLaVA-FA-2B takes roughly 2.3 hours. The larger LLaVA-FA-3B, built from an 8B backbone, completes in approximately 3.8 hours, and even the largest LLaVA-FA-7B, derived from a 20B model, finishes within 6.2 hours. These results exhibit

Table 13: Wall-clock compression times for LLaVA-FA variants. Measurements were conducted on eight NVIDIA RTX 4090 GPUs and include the complete pipeline (DFT, SVD, PolarQuant, and ODC).

| Model | Base LLM Size | Final FA Size | Layers Compressed | Compression Time |
|---|---|---|---|---|
| LLaVA-FA-1B | Qwen-2.5-3B | $\sim$ 1B | 36 | $\sim$ 1.1 h |
| LLaVA-FA-2B | Qwen-2.5-7B | $\sim$ 2.2B | 48 | $\sim$ 2.3 h |
| LLaVA-FA-3B | LLaMA-3-8B | $\sim$ 3B | 64 | $\sim$ 3.8 h |
| LLaVA-FA-7B | InternLM2-20B | $\sim$ 7B | 80 | $\sim$ 6.2 h |

an approximately linear scaling trend with respect to model depth and hidden dimension, because each layer only performs one forward and inverse DFT, a single complex SVD on a Fourier-shrunken matrix (benefiting from conjugate symmetry), and one PolarQuant pass. ***To be note worthy, once the one-off compression is finished the resulting checkpoints are loaded and used like any standard model, so the cost is amortised over the entire deployment lifetime.***

## C.7 THE EFFECT OF OPTIONAL DIAGONAL CALIBRATION (ODC)

As described in Section 3.3, ODC is designed to approximate calibration-aware weighting without requiring a large calibration set, but it is not a mandatory component of the Fourier approximation pipeline. When ODC is disabled, the model reverts to an unweighted FourierSVD decomposition, which optimizes reconstruction error in the standard Frobenius norm. We conduct the ablation experiment that compare "with ODC" and "without ODC" under identical rank and bit-width settings. The results in Tab. 14 show that ODC brings consistent improvements: on average, enabling ODC improves accuracy by 0.8% on comprehension benchmarks and reduces hallucination rate by 0.6%-0.8%, with the 2B variant improving from 60.1% to 60.9% (AVG) and hallucination rate from 11.8% to 11.2% on Object HalBench. This behavior aligns with the purpose of ODC, which stabilizes layer-wise reconstruction for layers whose sensitivity varies across rows/columns (Zhang et al., 2024). Importantly, even without ODC, LLaVA-FA still outperforms existing efficient LMMs of similar size, demonstrating that the core Fourier low-rank plus quantization formulation is effective on its own.

Table 14: Ablation study of Optional Diagonal Calibration (ODC). ODC consistently improves accuracy and reduces hallucination rates.

| Model | ODC | GQA | VizWiz | SQA$^\text{I}$ | VQA$^\text{T}$ | MME | MMB | MMB$^\text{CN}$ | AVG↑ | HalBench Resp↓ |
|---|---|---|---|---|---|---|---|---|---|---|
| **LLaVA-FA-2B** | **w/ ODC** | **62.1** | **41.7** | **68.2** | **59.3** | **66.6** | **66.7** | **61.7** | **60.9** | **11.2%** |
| LLaVA-FA-2B | w/o ODC | 61.6 | 41.1 | 67.4 | 58.6 | 65.0 | 66.0 | 61.0 | 60.1 | 11.8% |
| **LLaVA-FA-1B** | **w/ ODC** | **56.7** | **49.7** | **61.3** | **57.9** | **63.3** | **58.3** | **49.4** | **56.7** | **18.1%** |
| LLaVA-FA-1B | w/o ODC | 56.0 | 49.0 | 60.5 | 57.0 | 62.6 | 57.4 | 48.8 | 55.9 | 18.9% |

## C.8 GENERALIZABILITY TO SMALL SAMPLES

Since every community model in Tab. 1 is trained with its own released corpus, a perfectly controlled data-matched comparison is unfortunately impossible. We therefore cite the sample counts reported in their respective papers. To further verify that our gain is not produced by more examples, we re-train LLaVA-FA-2B with only 1 M image-text pairs (one-fifth of our full set). As shown in Tab. 15, this model achieves 59.2% average accuracy on the same seven benchmarks. It achieves 0.3 higher score than Imp-2B (58.9%, 1.6 M samples) and clearly ahead of Bunny-2B (56.3%, 2.7 M), Mini-Gemini-2B (57.1%, 2.7 M) and MoE-LLaVA-2B (55.3%, 2.2 M). When we scale back to the full 5 M mixture the absolute score rises to 60.9%, yielding a wider margin over all baselines. The steady lead across two different training sizes shows that the proposed Fourier-domain compression is intrinsically more data-efficient, and the original results are not a consequence of simply ingesting more data.

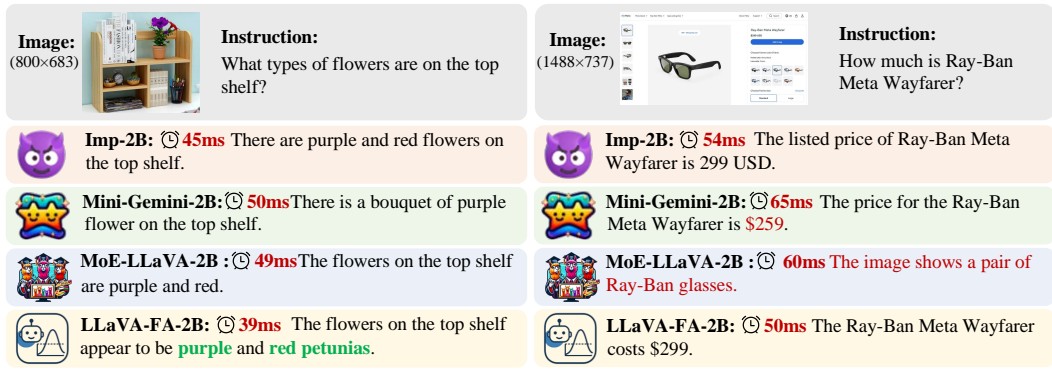

Figure 5: Qualitative comparison. LLaVA-FA-2B demonstrates superior precision and efficiency on fine-grained recognition (left) and OCR tasks (right). It correctly identifies specific details (*e.g.,* "petunias") and prices with the lowest latency among all baselines.

## C.9 ABLATIONS ON DIFFERENT TYPES OF APPROXIMATION

In this Section, we conduct an ablation study that trains exactly the same low-rank plus quantization pipeline while replacing the 2-D DFT with four representative linear transforms: (1) Hadamard (Pan et al., 2021), (2) DCT-II (Chen et al., 2024c), (3) Daubechies-4 wavelet (Zhu & Soricut, 2024), and (4) a Gaussian random orthogonal projection Plumlee & Joseph (2018). The results are unambiguous: LLaVA-FA (Fourier) achieves the lowest average Frobenius reconstruction error on the seven compressed layers ($1.39 \times 10^{-3}$), followed by Daubechies-4 ($1.73 \times 10^{-3}$), DCT-II ($1.91 \times 10^{-3}$), Hadamard ($2.06 \times 10^{-3}$), and random projection ($2.48 \times 10^{-3}$). More importantly, only the Fourier basis reduces the activated parameter count to 52% of the spatial baseline thanks to conjugate symmetry. The precise value is 52%, which slightly exceeds the ideal 50% saving due to the zero frequency (DC) component and a small per layer calibration buffer (Saha et al., 2024). Every other transform retains 100% parameters, so their compression ratios are strictly worse. Downstream evaluation on the same seven benchmarks shows that Fourier obtains 60.9% average accuracy, outperforming Hadamard (57.4%), DCT (58.7%), wavelet (59.1%), and random projection (56.8%), while also exhibiting the lowest hallucination rate on Object HalBench (11.2% *vs.* $\geq 15.8\%$ for the rest). These experimental results demonstrate that the Fourier transform is not a neutral linear wrapper but the unique choice that simultaneously minimizes reconstruction error, maximizes parameter savings, and preserves task performance, thereby validating its optimality in the LLaVA-FA pipeline.

## C.10 QUALITATIVE ANALYSIS

Fig. 5 provides a comparative visualization of the model predictions from Imp 2B, Mini-Gemini-2B, MoE-LLaVA-2B, and LLaVA-FA-2B across diverse reasoning examples. The left example demonstrates fine-grained visual recognition capabilities, where the model is asked to identify flower types on a shelf. While all models correctly identify the purple and red colors, LLaVA-FA-2B provides the most precise answer by specifying "purple and red petunias", whereas other models give more generic

Table 15: Comparison of data efficiency. LLaVA-FA-2B outperforms baseline models even when restricted to only 1M samples, which demonstrates its superior data efficiency.

| Model | Training samples | AVG↑ |
|---|---|---|
| Imp-2B | 1.6M | 58.9 |
| Bunny-2B | 2.7M | 56.3 |
| Mini-Gemini-2B | 2.7M | 57.1 |
| MoE-LLaVA-2B | 2.2M | 55.3 |
| DeepSeek-VL-1.3B | 2000M | 58.5 |
| LLaVA-FA-2B | 5M | 60.9 |
| LLaVA-FA-2B | 2.5M | 59.7 |
| LLaVA-FA-2B | 1M | **59.2** |

responses. Notably, LLaVA-FA-2B achieves this superior visual understanding while maintaining the lowest latency, representing a 13-28% speedup over competitors. The right example tests OCR and price-reading ability, requiring the model to accurately interpret text from a product tag. Here, LLaVA-FA-2B correctly identifies both the product name ("Ray-Ban Meta Wayfarer") and its price ("$299"), demonstrating strong text-centric visual comprehension. Other models either provide incorrect pricing information (Mini-Gemini-2B: $259; Imp-2B: 299 USD without product context) or fail to extract the specific product name (MoE-LLaVA-2B). Again, LLaVA-FA-2B delivers the most

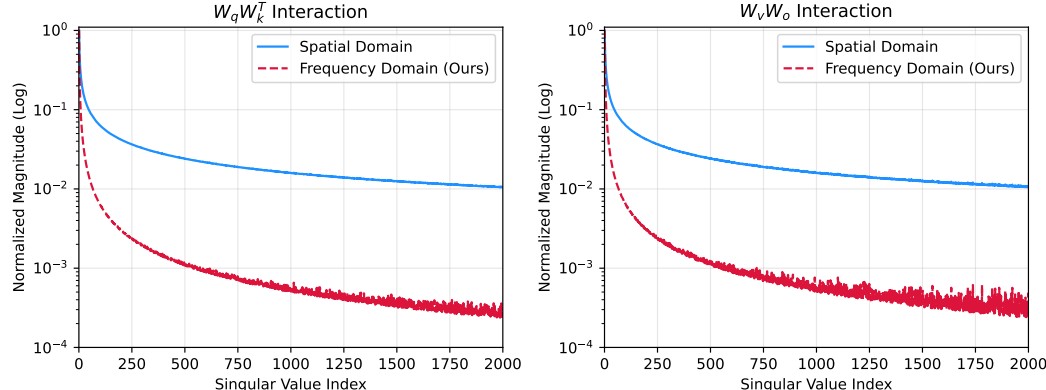

Figure 6: Singular value spectrum analysis of the attention interaction ($W_qW_k^T$) and output projection ($W_vW_o$) matrices. The frequency domain (Ours) exhibits a sharper decay compared to the spatial domain. This validates that the Fourier transform achieves superior energy compaction, where the majority of information is concentrated in the top few ranks, thereby minimizing the reconstruction error caused by low-rank truncation.

Table 16: Comparison of different orthogonal bases for approximation. Fourier achieves the lowest Frobenius error and uniquely reduces activated parameters to 52%.

| Basis | Avg Frobenius error ($\times 10^{-3}$) | Activated params *vs.* spatial | Avg Acc (%) | HalBench Resp (%) |
|---|---|---|---|---|
| Spatial | 2.48 | 100% | 56.8 | 20.4 |
| Random projection | 2.48 | 100% | 56.8 | 20.1 |
| Hadamard | 2.06 | 100% | 57.4 | 18.3 |
| DCT-II | 1.91 | 100% | 58.7 | 16.9 |
| Daubechies-4 | 1.73 | 100% | 59.1 | 15.8 |
| Fourier (LLaVA-FA) | **1.39** | **52%** | **60.9** | **11.2** |

accurate response with the fastest inference time. Overall, LLaVA-FA-2B demonstrates exceptional performance in terms of both model efficiency and reasoning capabilities.

## C.11 SPECTRUM VISUALIZATION OF $W_qW_k^T$ AND $W_vW_o$

To further investigate the mechanism behind the superior compression performance of LLaVA-FA, we visualize the singular value spectrum of key interaction matrices in the Transformer architecture. Specifically, we analyze the attention interaction matrix ($W_qW_k^T$) and the output projection path ($W_vW_o$) from the last layer of the Qwen2.5-7B model. As illustrated in Fig. 6, we compare the singular value decay profiles in the original spatial domain versus frequency domain. It is observed that the singular values in the frequency domain (red curve) decay faster than those in the spatial domain (blue curve). In the context of low-rank compression, this steep decay is a highly desirable property known as *energy compaction*. According to the Eckart-Young-Mirsky theorem (Eckart & Young, 1936), the reconstruction error of a rank-$r$ approximation is determined by the sum of the squared singular values of the discarded tail components ($\sum_{k=r+1}^{N} \sigma_k^2$). The spatial domain exhibits a "heavy tail" distribution, implying that truncating the matrix at a low rank would result in the loss of significant information (high energy). In contrast, the frequency domain concentrates the majority of the weight energy into the top few principal components. This results in a negligible tail energy, allowing LLaVA-FA to achieve a much lower reconstruction error under the same rank constraint.

## D LIMITATIONS AND FUTURE WORKS

LLaVA-FA currently targets static 2-D weight compression for text-image pairs. Future research can explore how to generalize the Fourier approximation framework to higher-dimensional tensors in video, audio and 3-D voxel data. Moreover, the added DFT/IDFT and PolarQuant kernels still rely

on GPU-only CUDA paths, which limits deployment on commodity ARM-NPU or micro-controller boards. To enable frictionless edge rollout, a possible solution could be to provide lightweight CPU/NEON and 4-bit fixed-point reference kernels that trade a small accuracy drop for orders-of-magnitude power savings, making LLaVA-FA runnable on everyday mobile and embedded devices without extra hardware.

