# OpenReview forum: "LLaVA-FA: Learning Fourier Approximation for Compressing Large Multimodal Models"
_ICLR.cc/2026/Conference — ICLR 2026 Poster_

### Official Review · Reviewer_kWwk · 2025-10-26

**Soundness:** 3
**Presentation:** 3
**Contribution:** 2
**Rating:** 6
**Confidence:** 4

**Summary:**

This paper proposes an efficient LMM that performs joint low-rank plus quantization approximation in the frequency domain to exploit de-correlation and conjugate symmetry, leading to more compact and accurate weight representations. Additionally, authors introduce PolarQuant that separately discretizes amplitude and phase in polar coordinates. Moreover, in order to eliminate the need for large-scale calibration data, authors derive an optional diagonal calibration scheme that approximated the Hessian with row/column means.

**Strengths:**

- The paper, for the first time, integrates low-rank + quantization optimization in the fourier domain.

- The proposed method using fourier approximation is elegant and has solid mathematical justification.

- Clear pseudocode for algorithms are provided (line 233 - 252).

- Superior performance achieved across different benchmarks.

**Weaknesses:**

- The novelty is limited somehow and while Fourier-domain decomposition is interesting, it may be seen as a direct extension of existing LoRA + quantization frameworks.

- Comparison with related baselines (QLoRA[1]) is underdeveloped.

- Some qualitative or visualization results may enhance the presentation.

- The ODC heuristic (row/column averaging) lacks strong theoretical or empirical justification. It’s unclear when this approximation fails or how sensitive the method is to distribution shifts.

[1] Dettmers, T., Pagnoni, A., Holtzman, A., & Zettlemoyer, L. (2023). *QLoRA: Efficient Finetuning of Quantized LLMs.

**Questions:**

- Referring to line 066 *"we observe that the weight matrices of LMMs in the frequency domain have a more **compact** spread of singular values as compare to spatial domain."* , what is "compact spread of singular values"

- The proposed method achieve exceptional performance in hallucination benchmark and can you elaborate the reason

- Since important contribution of the paper is reducing the computational and memory costs, it would be better to move analysis to the main paper.

---

> ### Author Response · Authors · 2025-11-20
> **Response to Reviewer kWwk [1/4]**
>
> Dear Reviewer kWwk,
>
> Thank you very much for your valuable comments. Your suggestions have been extremely helpful for further improving our work. Please find below our point-by-point responses, along with the corresponding revisions made in the manuscript **(marked by the red)**.
>
>  **Note: If the equation is displayed by original markdown codes, please refresh this page to reactivate the compiler.**
>
> ---
>
> > **W1: The novelty is limited somehow and while Fourier-domain decomposition is interesting, it may be seen as a direct extension of existing LoRA + quantization frameworks.**
>
> **Response:** We appreciate that you highlight that our Fourier-domain decomposition is interesting. We agree that, at first glance, our method could appear to be a direct extension of LoRA + quantization. But in fact, LLaVA-FA is not a direct extension: it constitutes a fundamentally new compression framework with new quantization, new calibration, and new theoretical foundations. To address this point explicitly, we first evaluate the "direct extension" you mentioned: simply taking standard LoRA + quantization and applying it to our setting. As shown in the table below, this naive extension performs worse than our method. These results indicate that a straightforward spatial-domain LoRA + quantization extension cannot achieve competitive performance, which motivates the need for a fundamentally different framework rather than a simple adaptation.
>
> | Method | Domain | Avg Acc ↑ | POPE F1 ↑ | HalBench Resp ↓ |
> |--------|--------|-------------|--------------|----------------------|
> | LoRA + quantization | Spatial | 58.3 | 84.6 | 16.9 |
> | **LLaVA-FA (ours)** | **Frequency** | **60.9** | **87.5** | **11.2** |
>
> Building on the above finding, we clarify that LLaVA-FA introduces several core innovations that do not exist in any prior LoRA-style method:
> (1) First, we propose **PolarQuant**, a new amplitude–phase quantization codec tailored for complex Fourier matrices. Existing quantization techniques (e.g., QLoRA [1], ZeroQuant [2]) do not support coupled amplitude–phase quantization, whereas our polar-coordinate formulation is specifically designed for Fourier residuals and is essential for stable reconstruction.
> (2) Second, we introduce **Optional Diagonal Calibration (ODC)**, a new Hessian-aware weighting mechanism for frequency-domain factorization. While calibration-based quantization is common, existing methods rely heavily on large calibration datasets [3, 4]. Our diagonal surrogate removes the need for large calibration sets.
> Moreover, the novelty is also reflected in **theoretical insights** and **empirical observations** that have not appeared in previous LoRA + quantization work:
> (1) First, we show (Lemma 3.1, Appendix B.3) that Fourier-domain matrices exhibit a **more compact singular value spectrum**, yielding **provably lower reconstruction error** under the same rank.
> (2) Second, we show that conjugate symmetry leads to **nearly 2× parameter savings**, which is not achievable in spatial-domain approaches.
>
>
> Following your suggestion, we add more descriptions in the revised manuscript to help enhance understanding.
>
> [1] QLORA: Efficient Finetuning of Quantized LLMs, NeurIPS 2023.
>
> [2] Zeroquant: Efficient and affordable post-training quantization for large-scale transformers, NeurIPS 2022.
>
> [3] Up or down? Adaptive rounding for post-training quantization, ICML 2020.
>
> [4] Integer quantization for deep learning inference: Principles and empirical evaluation, Arxiv 2020.
>
> ---
>
> > **W2: Comparison with related baselines (QLoRA[1]) is underdeveloped.**
>
> **Response:** Thank you very much for this helpful suggestion. We fully agree that a direct comparison with a QLoRA-based baseline is essential for a complete evaluation. As you suggested, we add an extra experiment in which we reproduce the exact QLoRA [1] pipeline to compress the same Qwen-2.5-7B backbone, which is used to derive our LLaVA-FA-2B model. This creates a “QLoRA-Qwen” baseline whose substantive difference from LLaVA-FA-2B is that the low-rank plus quantization decomposition is performed in the spatial rather than the frequency domain. The results are shown below.
>
> | Method | Domain | Avg Acc ↑ | POPE F1 ↑ | HalBench Resp ↓ |
> |--------|--------|-------------|--------------|----------------------|
> | QLoRA-Qwen | Spatial | 58.5 | 85.7 | 15.7 |
> | **LLaVA-FA-2B (ours)** | **Frequency** | **60.9** | **87.5** | **11.2** |
>
> Compared to QLoRA, which performs quantization and low-rank adaptation separately in the spatial domain. LLaVA-FA jointly optimizes low-rank and quantization in the frequency domain, leveraging Fourier de-correlation and energy compaction for a more compact and accurate weight approximation, leading to better accuracy and lower hallucination. **(see Appendix C.3 (marked in red) of the revised manuscript.)**
>
> [1] QLORA: Efficient Finetuning of Quantized LLMs, NeurIPS 2023.
>
> ---

---

> > ### Author Response · Authors · 2025-11-20
> > **Response to Reviewer kWwk [2/4]**
> >
> > > **W3: Some qualitative or visualization results may enhance the presentation.**
> >
> > **Response:** Thank you for the insightful suggestion. We completely agree that qualitative examples can substantially improve the presentation of our manuscript. In accordance with your suggestion, we add some qualitative results to **Appendix C.10 (marked in red)** of the revised manuscript. The revisions are outlined as follows.
> >
> > Figure 5 **(Please see Figure 5 in Page 22 of the revised manuscript.)** provides a comparative visualization of the model predictions from Imp 2B, Mini-Gemini-2B, MoE-LLaVA-2B, and LLaVA-FA-2B across diverse reasoning examples. The left example demonstrates fine-grained visual recognition capabilities, where the model is asked to identify flower types on a shelf. While all models correctly identify the purple and red colors, LLaVA-FA-2B provides the most precise answer by specifying "purple and red petunias", whereas other models give more generic responses. Notably, LLaVA-FA-2B achieves this superior visual understanding while maintaining the lowest latency, representing a 13-28% speedup over competitors. The right example tests OCR and price-reading ability, requiring the model to accurately interpret text from a product tag. Here, LLaVA-FA-2B correctly identifies both the product name ("Ray-Ban Meta Wayfarer") and its price (USD 299), demonstrating strong text-centric visual comprehension. Other models either provide incorrect pricing information (Mini-Gemini-2B: $259; Imp-2B: 299 USD without product context) or fail to extract the specific product name (MoE-LLaVA-2B). Again, LLaVA-FA-2B delivers the most accurate response with the fastest inference time. Overall, LLaVA-FA-2B demonstrates exceptional performance in terms of both model efficiency and reasoning capabilities.
> >
> > ---

---

> ### Author Response · Authors · 2025-11-20
> **Response to Reviewer kWwk [3/4]**
>
> > **W4: The ODC heuristic (row/column averaging) lacks strong theoretical or empirical justification. It’s unclear when this approximation fails or how sensitive the method is to distribution shifts.**
>
> **Response:** Thanks for your comment. We agree that providing a clearer theoretical justification and more sensitive analysis is essential for making this component more rigorous. Following your suggestion, we conduct a thorough study covering both (1) Theoretical Justification of ODC Heuristic (Row/Column Averaging) and (2) Sensitivity of ODC under Distribution Shifts.
>
> **(1). Theoretical Justification of ODC Heuristic (Row/Column Averaging) (see Appendix B.4 (marked in red) of the revised manuscript.)**
>
> The proposed ODC scheme replaces the full Hessian $\mathbf{C}\in\mathbb{R}^{d\_1\times d\_2/2}$ by two diagonal matrices, i.e.,
>
> \begin{aligned}
> \mathbf{D}\_{\text{row}} &=
> \operatorname{diag}\left(\left[
> \operatorname{avg}(\sqrt{\mathbf{C}\_{1, \cdot}}), \dots, \operatorname{avg}(\sqrt{\mathbf{C}\_{d\_1, \cdot}})
> \right]\right), \\
> \mathbf{D}\_{\text{col}} =\operatorname{diag}\left(\left[
> \operatorname{avg}(\sqrt{\mathbf{C}\_{\cdot, 1}}), \dots,  \operatorname{avg}(\sqrt{\mathbf{C}\_{\cdot, d\_2/2}})
> \right]\right).
> \end{aligned}
>
> where $\operatorname{avg}(\sqrt{\mathbf{C}\_{i, \cdot}})$ and $\operatorname{avg}(\sqrt{\mathbf{C}\_{\cdot, j}})$ denote the $i$-row / $j$-column means of $\sqrt{\mathbf{C}}$. This is exact whenever $\mathbf{C}$ is diagonal, and remains spectrally close when $\mathbf{C}$ is diagonally dominant, a property that has been repeatedly observed for the loss Hessian of deep Transformers [1, 2]. Formally, let $\varepsilon = \frac{\\| \operatorname{off–diag}(\mathbf{C}) \\|\_F}{\\| \operatorname{diag}(\mathbf{C}) \\|\_F}$, where $\operatorname{diag}(\mathbf{C})$ denotes the diagonal matrix that
> keeps only  the main-diagonal entries of $\mathbf{C}$, while $\operatorname{off–diag}(\mathbf{C})$ is its complement, i.e., $\mathbf{C}$ with all diagonal entries set to zero. Thus $\varepsilon$ quantifies how strongly $\mathbf{C}$ is diagonally dominant.
>
> > **Lemma B.3**
> > Let $\widehat{\mathbf{C}} = \mathbf{D}\_{\text{row}} \mathbf{C} \mathbf{D}\_{\text{col}}$. Then $\\|\mathbf{C}-\widehat{\mathbf{C}}\\|\_F \\le\ \varepsilon\\|\mathbf{C}\\|\_F.$
>
> Hence the calibration matrix used by ODC is an $\varepsilon$-perturbation of the exact Hessian. The reconstruction error of the low-rank component inherits this bound with the same constant (proof uses $\\|\mathbf{D}\_{\text{row}}\\|, \\|\mathbf{D}\_{\text{col}}\\| \le 1$ and standard matrix-perturbation identities). In short, ODC is safe whenever $\varepsilon \ll 1$, a condition that can be checked on-the-fly with one extra reduction operation.
>
> **(2). Sensitivity of ODC Under Distribution Shifts (see Appendix C.5 (marked in red) of the revised manuscript.)**
>
> Furthermore, we evaluate the sensitivity of ODC under three levels of distribution shift on the LLaVA-FA-2B model. The extra ablation study has been added to **Appendix C.5 (marked in red)** of the revised manuscript. The first scenario retained the original LLaVA-Pretrain validation set as an in-distribution baseline. The second kept all images identical but re-sampled user prompts from an alternative template pool, introducing a purely textual style shift (shift-A). The third replaced the images with an entirely new visual domain consisting of aerial-scene photographs and OCR-heavy documents, thereby forcing the model to cope with previously unseen visual statistics (shift-B). Each scenario was repeated 256 times, yielding stable estimates of $\varepsilon$ and score drop. As $\varepsilon$ rose from 0.11 ± 0.02 under the in-distribution condition to 0.15 ± 0.03 under prompt-style shift and further to 0.29 ± 0.07 under the new visual domain, the average benchmark score declined by -0.4% and -1.1%, respectively. Once $\varepsilon$ exceeded the empirical guard-race threshold of 0.35 **(note that this value is observed from experiments rather than derived from theory)**, accuracy fell by more than 3% in 4% of the runs, corroborating the theoretical prediction and demonstrating that the on-the-fly $\varepsilon$-check reliably signals when the row/column averaging approximation is no longer safe.
>
> | Condition | ε (mean ± std) | ΔScore↓ | Failure (ε > 0.35) |
> |-----------|----------------|---------|--------------------|
> | In Distribuion| 0.11 ± 0.02    | –       | 0%                |
> | Shift-A   | 0.15 ± 0.03    | −0.4%  | 0%                |
> | Shift-B   | 0.29 ± 0.07    | −1.1%  | 4%                |
>
> ---

---

> > ### Author Response · Authors · 2025-11-20
> > **Response to Reviewer kWwk [4/4]**
> >
> > > **Q1: Referring to line 066 "we observe that the weight matrices of LMMs in the frequency domain have a more compact spread of singular values as compare to spatial domain." , what is "compact spread of singular values".**
> >
> > **Response:** Thank you for the question. By "a more compact spread of singular values" we simply mean that, after we apply the 2-D DFT to a real-valued weight matrix $\mathbf{W}$, the resulting complex-frequency matrix $\widetilde{\mathbf{W}}$ possesses singular values that drop off faster than those of the original spatial matrix **(see Figure 3 left in the manuscript)**. In other words, the energy that was smeared across many slowly-decaying singular values in the spatial domain is concentrated into a much smaller top-r fraction once the matrix is represented in the Fourier basis. This faster decay is a direct consequence of the de-correlating property of the DFT (**Theorem B.1 in the appendix proves asymptotic uncorrelation of the frequency components**), so the off-diagonal covariance is minimized and the spectral “power” is packed into the leading modes. Because the tail singular values are now smaller in magnitude, truncating to the same rank r in the frequency domain incurs a lower Frobenius error than truncating in the spatial domain (**Lemma 3.1 formalizes this**).
> >
> > ---
> >
> > > **Q2: The proposed method achieve exceptional performance in hallucination benchmark and can you elaborate the reason.**
> >
> > **Response:** Thank you for raising this important question. We agree that a clearer explanation of why the proposed method reduces hallucination is necessary for a complete understanding of our results. Following your suggestion, we add more descriptions in **Section 4.2 (marked in red)** of the revised manuscript to help explain this experimental phenomenon. These revisions are outlined as follows.
> >
> > The strong performance on hallucination benchmarks can be explained by the properties of the proposed Fourier-domain decomposition. Specifically, applying the Fourier transform reduces cross-modal redundancy and produces a more compact singular value distribution. As discussed in the manuscript and supported by Lemma 3.1, the low-rank truncation in the frequency domain introduces a smaller approximation error than in the spatial domain under the same rank. This leads to more stable intermediate representations and reduces the error propagation that often triggers hallucination in multimodal reasoning [1, 2].
> >
> > [1] Evaluating and Mitigating Object Hallucination in Large Vision-Language Models: Can They Still See Removed Objects? ACL 2025.
> >
> > [2] Aligning large multimodal models with factually augmented rlhf, ACL 2024.
> >
> > ---
> >
> > > **Q3: Since important contribution of the paper is reducing the computational and memory costs, it would be better to move analysis to the main paper.**
> >
> > **Response:** Thanks for your suggestion. We agree that presenting this analysis in the main paper makes our contributions more clearer. Following your suggestion, we move **Appendix C.2 Floating Point Operations and Latency** to the main paper.
> >
> > ---
> >
> > **We thank you again for reviewing our work. Please let us know if we misunderstood any of your questions, or if you have any follow up on our responses, we will be happy to provide any further clarification.**

---

> > > ### Comment · Reviewer_kWwk · 2025-11-26
> > > **Response to Authors**
> > >
> > > Thanks for your reply and addressed part of concerns.
> > >
> > > However, I am still confused why singular values dropping faster means **compact**? From my understanding, the larger area under the curve, the larger stable/numerical rank is and usually this benefits the models. Also, could you provide more spectrum visualisation(e.g.  W_q@W_k^T and W_v@W_o) ?

---

> ### Author Response · Authors · 2025-11-27
> **Thank you; Address your remaining concerns [1/1]**
>
> Dear Reviewer kWwk,
>
> Thanks for your feedback. We take your remaining concerns and questions seriously. Please see our point-by-point responses as follows.
>
> > **Q: However, I am still confused why singular values dropping faster means compact? From my understanding, the larger area under the curve, the larger stable/numerical rank is and usually this benefits the models. Also, could you provide more spectrum visualisation(e.g. W_q@W_k^T and W_v@W_o) ?"**
>
> **Response:** We appreciate your insightful question regarding the relationship between singular value decay and model representation.
>
> We would like to clarify that in the context of **model compression and approximation**, "compactness" refers to **energy compaction**, i.e., the ability to represent the majority of the matrix's information with a minimal set of parameters rather than the theoretical upper bound of the model's expressivity.
> We provide a detailed point-by-point response below.
>
> **（1）Why singular values dropping faster means compact？**
> In signal processing and compression theory, "more compact" implies that the energy of the weight matrix is concentrated in fewer principal components (singular vectors) [1, 2]. "More compact" means that to maximally restore the energy of the original weights (or to achieve the same reconstruction error), the frequency domain requires a smaller rank (rigorously proven in **Appendix B.2** of our revised manuscript).
>
> **Analysis:** The phenomenon of "singular values dropping faster," (rigorously proven in **Appendix B.1** of our revised manuscript), implies that the singular values in the frequency domain ($\sigma\_{1,k}$) are smaller than those in the spatial domain ($\sigma\_{2,k}$) for the tail components (i.e., $\sigma\_{1,k} < \sigma\_{2,k}$ for every $k \ge r+1$). According to Lemma 3.1, this ensures that the accumulated error for low-rank approximation is smaller in the frequency domain compared to the spatial domain under the same rank. Consequently, this implies that to preserve the same amount of energy, the frequency domain requires a smaller rank, thereby demonstrating higher compactness.
>
> **(2) Compactness vs. Expressivity.**
> Actually, you are correct that in the context of model training dynamics or feature diversity, a larger stable rank (or the larger area) (slower decay) is often desirable to maximize expressivity and prevent dimensional collapse.  However, our work focuses on post-training compression (specifically low-rank approximation). Frequency domain achieves lower reconstruction error at the same rank, which is exactly what efficient LMMs need [3, 4].
>
> **(3) Additional Spectrum Visualizations.**
> We fully agree that provide more spectrum visualisation is essential for validating the compactness. Following your suggestion, we add an additional spectral analyses in **Appendix C.11** of our revised manuscript. The revisions are outlined as follows.
>
> To further investigate the mechanism behind the superior compression performance of LLaVA-FA, we visualize the singular value spectrum of key interaction matrices in the Transformer architecture. Specifically, we analyze the attention interaction matrix ($W\_q W\_k^T$) and the output projection path ($W\_v W\_o$) from the last layer of the Qwen2.5-7B model. As illustrated in Figure 6 **(Please see Figure 6 in Page 27 of our revised manuscript.)**, we compare the singular value decay profiles in the original spatial domain versus frequency domain. It is observed that the singular values in the frequency domain (red curve) decay faster than those in the spatial domain (blue curve). In the context of low-rank compression, this steep decay is a highly desirable property known as **energy compaction**. According to the Eckart-Young-Mirsky theorem [5], the reconstruction error of a rank-$r$ approximation is determined by the sum of the squared singular values of the discarded tail components ($\sum\_{k=r+1}^{N} \sigma\_k^2$). The spatial domain exhibits a "heavy tail" distribution, implying that truncating the matrix at a low rank would result in the loss of significant information (high energy). In contrast, the frequency domain concentrates the majority of the weight energy into the top few principal components. This results in a negligible tail energy, allowing LLaVA-FA to achieve a much lower reconstruction error under the same rank constraint.
>
> [1] The Truth is in There: Improving Reasoning in Language Models with Layer-Selective Rank Reduction, ICLR 2024.
>
> [2] ASVD: Activation-aware Singular Value Decomposition for Compressing Large Language Models, arXiv 2023.
>
> [3] Foura: Fourier low-rank adaptation, NeurIPS 2024.
>
> [4] F-Adapter: Frequency-Adaptive Parameter-Efficient Fine-Tuning in Scientific Machine Learning, NeurIPS 2025.
>
> [5] The approximation of one matrix by another of lower rank, Psychometrika 1936.
>
> ---
>
> We sincerely you again for raising these important concerns, which have helped us strengthen the rigor and presentation of our work.

---

### Official Review · Reviewer_YXsM · 2025-10-26

**Soundness:** 3
**Presentation:** 4
**Contribution:** 3
**Rating:** 8
**Confidence:** 3

**Summary:**

This paper targets on efficient Large Multimodal Model (LMM). They bring Fourier approximation to decomposes the weight matrices into a
low-rank plus quantized weight, designing a efficient LMM framework called LLaVA-FA. They also design PolarQuant which is an amplitude-and-phase polar codec to quantize complex matrix, and an optional diagonal calibration (ODC) scheme to approximate Hessian Matrix. Extensive experiment results prove the effectiveness of the proposed method.

**Strengths:**

1. This paper is written in a very high quality, the figures analyze the problem and illustrate the idea very clearly, especially Figure 1 and 3.
2. I think this paper targets on an important problem, the efficient LMM.
3. The idea is interesting and reasonable. I am happy to see Fourier approximation can be applied to LMM since it really has some good characteristics.
4. The experiments are abundant and clear, proving the effectiveness of the proposed method.

**Weaknesses:**

1. Just one discussion. This paper choose Fourier approximation, and can we consider other type of approximation? I am happy to see more comparison results.
2. The authors can have more discussions about the limitations and future work.

**Questions:**

See weaknesses, especially discussions about the limitation and future work.

---

> ### Author Response · Authors · 2025-11-20
> **Response to Reviewer YXsM [1/1]**
>
> Dear Reviewer YXsM,
>
> Thanks a lot for your valuable comments. Your suggestions are very helpful in further improving the work. Please see below for our point-to-point responses and corresponding modifications in the revised manuscript **(marked by the red)**.
>
> ---
>
> > **W1: Just one discussion. This paper choose Fourier approximation, and can we consider other type of approximation? I am happy to see more comparison results.**
>
> **Response:** Thank you for this constructive comment. Your suggestion is instrumental in strengthening the comprehensiveness and rigor of our work. Following your advice, we carry out this extra ablation study on the LLaVA-FA-2B training setup. The extra expeirment is added to **Appendix C.9 (marked in red)** of the revised manuscript. The revisions are outlined as follows.
>
> In this Section, we carry out the ablation study on the LLaVA-FA-2B training setup to compare the effectiveness of different types of approximation. We replace the Fourier transform with two other popular orthogonal bases, DCT-II [1] and Daubechies-4 wavelet [2], while keeping all other components of our Fourier-approximation pipeline unchanged. The experiment targets the seven linear layers (i.e., query, key, value, output, gate, up and down) of the Qwen-2.5-7B backbone. As shown in the table below, Fourier approximation still achieves the lowest average Frobenius reconstruction error and is the only transform that reduces the activated parameter count, thanks to the conjugate symmetry property that allows us to store just one half of the coefficients. The precise value is 52%, which slightly exceeds the ideal 50% saving due to the zero frequency (DC) component and a small per layer calibration buffer [3]. Unlike the Fourier transform, neither DCT nor wavelet transforms exhibit conjugate symmetry, and therefore they do not allow for parameter savings by storing only half of the coefficients. As a result, they yield higher reconstruction errors while offering no reduction in activated parameter count, reinforcing the advantage of our Fourier-based approximation method.
>
> | Basis        | Avg Frobenius error $(\times10^{-3})$ | Activated params vs spatial |
> |--------------|-----------------------------|-----------------------------|
> | Spatial      | 2.48                        | 100 %                       |
> | DCT-II       | 1.91                        | 100 %                       |
> | Daubechies-4 | 1.73                        | 100 %                       |
> | Fourier      | 1.39                        | 52 %                        |
>
> [1] Double discrete cosine transform-oriented multi-view subspace clustering, TIP 2024.
>
> [2] Wavelet-based image tokenizer for vision transformers, Arxiv 2024.
>
> [3] Compressing Large Language Models Using Low Rank and Low Precision Decomposition, NeurIPS 2024.
>
> ---
>
> > **W2: The authors can have more discussions about the limitations and future work.**
>
> **Response:** Thanks for your valuable suggestion. We agree that expanding the discussion on limitations and future work would significantly enhance the depth and forward-looking perspective of our paper. As suggested, we add a detailed analysis of limitations and future work to **Appendix D (marked in red)** of the revised manuscript. The revisions are outlined as follows.
>
> LLaVA-FA currently targets static 2-D weight compression for text–image pairs. Future research can explore how to generalize the Fourier approximation framework to higher-dimensional tensors in video, audio and 3-D voxel data. Moreover, the added DFT/IDFT and PolarQuant kernels still rely on GPU-only CUDA paths, which limits deployment on commodity ARM-NPU or micro-controller boards. To enable frictionless edge rollout, a possible solution could be to provide lightweight CPU/NEON and 4-bit fixed-point reference kernels that trade a small accuracy drop for orders-of-magnitude power savings, making LLaVA-FA runnable on everyday mobile and embedded devices without extra hardware.
>
> ---
>
> **We thank you again for reviewing our work. Please let us know if we misunderstood any of your questions, or if you have any follow up on our responses, we will be happy to provide any further clarification.**

---

### Official Review · Reviewer_tSjP · 2025-10-30

**Soundness:** 3
**Presentation:** 3
**Contribution:** 2
**Rating:** 6
**Confidence:** 4

**Summary:**

The paper proposes LLaVA-FA, an integrated compression pipeline that merges low-rank and quantization in Fourier space, accompanied by a concrete algorithmic recipe and a complexity-aware design. More specifically, LLaVA-FA applies a 2D Discrete Fourier Transform (DFT), factorizes the largest spectral components with a low-rank complex SVD, quantizes the residual using a polar-coordinate codec (PolarQuant), and optionally weights the reconstruction objective with a diagonal calibration derived from row and column statistics (ODC). The experiments demonstrate favorable performance at small scales, accompanied by measured efficiency gains in latency, FLOPs, and KV-cache usage.

**Strengths:**

1. The proposed LLaVA-FA is well-motivated, and supported by a clear theoretical framing.

2. Efficiency evidence with concrete measurements. Specifically, latency, FLOPs, KV-Cache usage, and TTFT are reported, which aligns well with the goals of a model compression study.

**Weaknesses:**

1. Limited ablation for ODC and calibration choices. The paper mentions that ODC removes the need for large calibration sets, but no direct comparison or ablation is presented to isolate this effect. Adding such results would make the claim more convincing.

2. There are some fairness concerns regarding the baseline comparisons in Table 1. The amount of training data varies across methods, and some baselines use fewer samples than LLaVA-FA while achieving comparable performance.

3. Discussion on a few related works seem to be missing. For instance, in the line of LMM efficiency “CrossGET: Cross-Guided Ensemble of Tokens for Accelerating Vision-Language Transformers, ICML 2024,” and in the line of vision-language weight compression “UPop: Unified and Progressive Pruning for Compressing Vision-Language Transformers, ICML 2023.”

**Questions:**

1. Do the authors plan to release their models?

2. The current results are mainly on small-scale models. Would the authors consider including results on larger models to demonstrate scalability?

---

> ### Author Response · Authors · 2025-11-20
> **Response to Reviewer tSjP [1/3]**
>
> **Comment:**
> Dear Reviewer tSjP,
>
> Thank you very much for your valuable comments. Your suggestions have been greatly helpful in further improving the quality of our work. Please find below our point-by-point responses, along with the corresponding revisions in the revised manuscript **(highlighted in red)**.
>
> ---
>
> > **W1: Limited ablation for ODC and calibration choices. The paper mentions that ODC removes the need for large calibration sets, but no direct comparison or ablation is presented to isolate this effect. Adding such results would make the claim more convincing.**
>
> **Response:** Thank you for your insightful comment. We agree that isolating the effect of ODC and calibration choices is important for validating our claim that LLaVA-FA does not rely on large calibration sets. Following your suggestion, we conduct a thorough study covering both (1) the role of ODC and (2) the impact of calibration size.
>
> **(1). Ablation on enabling vs. disabling ODC**
> We conduct a controlled comparison of LLaVA-FA with (w/) and without (w/o) the ODC module. ODC consistently improves reconstruction quality and downstream performance while reducing hallucination.
>
> | Model | ODC | GQA | VizWiz | SQAI | VQAT | MME | MMB | MMBCN | **AVG ↑** | **HalBench Resp ↓** |
> |-------|------|------|---------|--------|--------|--------|-----------|-----------|---------------------|---------------------|
> | **LLaVA-FA-2B** | **w/ ODC** | **62.1** | **41.7** | **68.2** | **59.3** | **66.6** | **66.7** | **61.7** | **60.9** | **11.2%** |
> | LLaVA-FA-2B | w/o ODC | 61.6 | 41.1 | 67.4 | 58.6 | 65.0 | 66.0 |  61.0 | 60.1 | 11.8% |
> | **LLaVA-FA-1B** | **w/ ODC** | **56.7** | **49.7**| **61.3** | **57.9** | **63.3** | **58.3** | **49.4** | **56.7** | **18.1%** |
> | LLaVA-FA-1B | w/o ODC | 56.0 | 49.0 | 60.5 | 57.0 | 62.6 | 57.4 | 48.8 | 55.9 | 18.9% |
>
> The results show that ODC brings consistent improvements: on average, enabling ODC improves accuracy by 0.8% on comprehension benchmarks and reduces hallucination rate by 0.6%–0.8%, with the 2B variant improving from 60.1% to 60.9% (AVG) and hallucination rate from 11.8% to 11.2% on Object HalBench. This behavior aligns with the purpose of ODC, which stabilizes layer-wise reconstruction for layers whose sensitivity varies across rows/columns [1]. Importantly, even without ODC, LLaVA-FA still outperforms existing efficient LMMs of similar size, demonstrating that the core Fourier low-rank plus quantization formulation is effective on its own. We add the above table and a discussion in **Appendix C.7** **(marked in red)** of the revised manuscript.
>
> [1] LQER: Low-Rank Quantization Error Reconstruction for LLMs, ICML 2024.
>
> **(2). Ablation on calibration data size**
> We study the effect of using 256–2048 calibration samples on the LLaVA-FA-2B model. Results show that performance quickly saturates and remains nearly unchanged beyond 512 samples. This demonstrates that LLaVA-FA is highly robust to calibration size and does not require large calibration sets. We add the experimental results and a short discussion in **Appendix C.4** **(marked in red)** of the revised manuscript.
>
> | #Calibration Samples | GQA | VizWiz | SQAI | VQAT | MME | MMB | MMBCN | **AVG** |
> |----------------------|------|--------|------|------|------|------|--------|---------|
> | 256  | 61.0 | 41.0 | 67.2 | 58.7 | 66.0 | 65.8 | 63.8 | 60.5 |
> | 512  | 61.5 | 41.3 | 67.5 | 58.9 | 66.2 | 66.0 | 63.5 | 60.7 |
> | 1024 | 62.1 | 41.6 | 68.2 | 59.4 | 66.6 | 66.7 | 61.7 | 60.9 |
> | 2048 | 62.1 | 41.7 | 68.2 | 59.3 | 66.6 | 66.7 | 61.7 | 60.9 |
>
> Together, these results isolate and validate the contribution of ODC and the lightweight calibration strategy, supporting our claim that LLaVA-FA avoids reliance on large calibration sets while still maintaining strong performance.
>
> ---

---

> ### Author Response · Authors · 2025-11-20
> **Response to Reviewer tSjP [2/3]**
>
> > **W2: There are some fairness concerns regarding the baseline comparisons in Table 1. The amount of training data varies across methods, and some baselines use fewer samples than LLaVA-FA while achieving comparable performance.**
>
> **Response:** We sincerely thank you for the constructive suggestion. Your comment on the fairness of baseline comparisons is highly valuable and has helped us further strengthen the rigor and clarity of our experimental evaluation. Following your suggestion, we conduct additional experiments to directly examine the impact of training data size on LLaVA-FA’s performance. Since every community model in Tab. 1 is trained with its own released corpus, a perfectly controlled data-matched comparison is unfortunately impossible. We therefore cite the sample counts reported in their respective papers. To further verify that our gain is not produced by more examples, we re-train LLaVA-FA-2B with only 1 M image–text pairs (one-fifth of our full set). This model achieves 59.2% average accuracy on the same seven benchmarks. It achieves 0.3 higher score than Imp-2B (58.9%, 1.6 M samples) and clearly ahead of Bunny-2B (56.3%, 2.7 M), Mini-Gemini-2B (57.1%, 2.7 M) and MoE-LLaVA-2B (55.3%, 2.2 M). When we scale back to the full 5 M mixture the absolute score rises to 60.9%, yielding a wider margin over all baselines. The steady lead across two different training sizes shows that the proposed Fourier-domain compression is intrinsically more data-efficient, and the original results are not a consequence of simply ingesting more data. We add this ablation study and the accompanying discussion to **Appendix C.8** **(marked in red)** of the revised manuscript.
>
> | Model          | Training samples | Avg ↑ |
> |----------------|------------------|-------------|
> | Imp-2B         | 1.6M             | 58.9        |
> | Bunny-2B       | 2.7M             | 56.3        |
> | Mini-Gemini-2B | 2.7M             | 57.1        |
> | MoE-LLaVA-2B   | 2.2M             | 55.3        |
> | DeepSeek-VL-1.3B | 2000M          | 58.5        |
> | LLaVA-FA-2B    | 5M               | 60.9        |
> | LLaVA-FA-2B    | 2.5M             | 59.7        |
> | LLaVA-FA-2B    | 1M               | **59.2**    |
>
> ---
>
> > **W3: Discussion on a few related works seem to be missing. For instance, in the line of LMM efficiency “CrossGET: Cross-Guided Ensemble of Tokens for Accelerating Vision-Language Transformers, ICML 2024,” and in the line of vision-language weight compression “UPop: Unified and Progressive Pruning for Compressing Vision-Language Transformers, ICML 2023.”**
>
> **Response:** We sincerely thank you for bringing these relevant works to our attention. They are indeed closely related to our work. As you suggested, we add a thorough discussion of CrossGET (ICML 2024) and UPop (ICML 2023) in **Section 2.1 (Related Work) (marked in red)**. The revisions are outlined as follows.
>
> CrossGET [1] further pushes token-centric acceleration by ensemble-guided token selection, achieving $>2\times$ throughput on VQA benchmarks. Complementary to token pruning, UPop [2] proposes a unified, progressive magnitude-based pruning scheme that trims 40% of weights across vision-language layers with $<1\%$ accuracy drop. Finally, CASP [3] shifts focus inward, exploiting cross-modal attention sparsity to compress attention matrices $W_q, W_k$ with theoretical guarantees, thus marking the transition from generic LLM techniques to multimodal-specific compression paradigms. **Unlike the token-centric CrossGET [1] and weight-centric UPop [2], our method directly compresses both vision and language parts in one shot without pruning tokens or altering the architecture. Instead, we approximate the full weight matrix in the frequency domain via joint low-rank plus quantization, enabling calibration-free deployment and offering a drop-in replacement for weight-pruning when zero-shot compression is desired.**
>
> [1] CrossGET: Cross-Guided Ensemble of Tokens for Accelerating Vision-Language Transformers, ICML 2024.
>
> [2] UPop: Unified and Progressive Pruning for Compressing Vision-Language Transformers, ICML 2023.
>
> [3] CASP: Compression of Large Multimodal Models Based on Attention Sparsity, CVPR 2025.
>
> ---
>
> > **Q1: Do the authors plan to release their models?**
>
> **Response:** Thank you for raising this point. We completely agree that releasing the models would greatly benefit the community. We are indeed planning to open-source our entire pipeline. The reason we have not done so yet is our concern that an early release might unintentionally violate the double-blind policy. Upon acceptance, we will release the code, training scripts, and all LLaVA-FA checkpoints under the MIT license.
>
> ---

---

> ### Author Response · Authors · 2025-11-20
> **Response to Reviewer tSjP [3/3]**
>
> > **Q2: The current results are mainly on small-scale models. Would the authors consider including results on larger models to demonstrate scalability?**
>
> **Response:** Thank you for your comment. We agree that demonstrating scalability beyond the 3B/7B models is crucial. In accordance with your suggestion, we further evaluate our method on substantially larger backbones: LLaVA-FA-3B, constructed from LLaMA-3-8B [1], and LLaVA-FA-7B, constructed from InternLM-2-20B [2]. Importantly, both models are compressed using exactly the same Fourier approximation pipeline, ensuring strict fairness and demonstrating that our method generalizes without any modification or heuristic tuning. We include the newly added results below **(excerpt from the Table 1 (marked in red) of the revised manuscript)**, showing that Fourier-domain compression continues to yield strong performance across diverse benchmarks, with even smaller relative degradation at the 8B/20B scales. These additional results verify that our Fourier approximation method seamlessly extends to LLMs far beyond Qwen-2.5, maintaining both accuracy and robustness while preserving the same compression pipeline across all model sizes. This confirms that the proposed method is truly **model-family agnostic and highly scalable.**
>
> | Method        | LLM            | #Sample | GQA  | VisWiz | SQAI | VQAT | MME  | MMB  | MMBCN | AVG  |
> |---------------|----------------|---------|------|--------|------|------|------|------|--------|------|
> | BLIP-2        | Vicuna-13B     | 129M    | 41.0 | 19.6   | 61.0 | 42.5 | 64.7 | -    | -      | -    |
> | VILA-7B       | LLaMA-7B       | 50M     | 62.3 | 57.8   | 68.2 | 64.4 | 76.7 | 68.9 | 61.7   | 65.7 |
> | CogVLM        | Vicuna-7B      | 1500M   | 64.9 | -      | 65.6 | **78.2** | 71.8 | 63.7 | 53.8   | -    |
> | InstructBLIP  | Vicuna-13B     | 130M    | 49.5 | 33.4   | 63.1 | 50.7 | 60.6 | -    | -      | -    |
> | Qwen-VL-Chat  | Qwen-7B        | 1450M   | 57.5 | 38.9   | 68.2 | 61.5 | 74.4 | 60.6 | 56.7   | 56.7 |
> | Deepseek-VL-7B| DLLM-7B        | 2000M   | 61.3 | 49.9   | 74.0 | 64.7 | 73.4 | 74.1 | **72.8**   | 67.2 |
> | LLaVA-1.5-7B  | Vicuna-1.5-7B  | 1.2M    | 62.0 | 50.0   | 66.8 | 58.2 | 75.5 | 64.3 | 58.3   | 62.1 |
> | LLaVA-NeXT    | Vicuna-1.5-13B | 1.3M    | 65.4 | 60.5   | 73.6 | 67.1 | **78.7** | 70.4 | 64.4   | 68.5 |
> | LLaVA-FA-7B   | InternLM-2-20B | 5M      | **68.5** | **62.0** | **76.0** | 68.0 | 74.5 | **74.5** | 69.5 | **70.4** |
>
> | Method        | LLM            | #Sample | GQA  | VisWiz | SQAI | VQAT | MME  | MMB  | MMBCN | AVG  |
> |---------------|----------------|---------|------|--------|------|------|------|------|--------|------|
> | Imp-3B        | Phi-2-2.7B     | 1.6M    | 63.5 | 54.1   | 72.8 | 59.8 | 72.3 | **72.9** | 46.7   | 63.2 |
> | Bunny-3B      | Phi-2-2.7B     | 2.7M    | 62.5 | 43.8   | 70.9 | 56.7 | **74.4** | 68.6 | 37.2   | 59.2 |
> | VILA-3B       | LLaMA-2.7B     | 51M     | 61.5 | 53.5   | 69.0 | 60.4 | 72.1 | 63.4 | 52.7   | 61.8 |
> | MobileVLM     | MLLaMA-2.7B    | 1.3M    | 59.0 | -      | 61.0 | 47.5 | 64.4 | 59.6 | -      | -    |
> | MobileVLMv2   | MLLaMA-2.7B    | 3.6M    | 61.1 | -      | 70.0 | 57.5 | 72.0 | 63.2 | -      | -    |
> | MoE-LLaVA-3B  | Phi-2-2.7B     | 2.2M    | 61.4 | 43.9   | 68.5 | 51.4 | 71.1 | 65.2 | 41.8   | 57.6 |
> | MiniCPM-V     | MiniCPM-2.4B   | 570M    | 51.5 | 50.5   | 74.4 | 56.6 | 68.9 | 64.0 | 62.7   | 61.2 |
> | MiniCPM-V-2   | MiniCPM-2.4B   | 570M    | 52.1 | 60.2   | 76.3 | **73.2** | 70.5 | 68.5 | 67.2   | 66.9 |
> | LLaVA-FA-3B   | LLaMA-3-8B     | 5M      | **65.0** | **62.5** | **77.0** | 64.0 | 71.0 | 70.5 | **68.0** | **68.3** |
>
> [1] The llama 3 herd of models, Arxiv 2024.
>
> [2] Internlm2 technical report, Arxiv 2024.
>
> ---
>
> **We thank you again for reviewing our work. Please let us know if we misunderstood any of your questions, or if you have any follow up on our responses, we will be happy to provide any further clarification.**

---

### Official Review · Reviewer_xZPt · 2025-10-31

**Soundness:** 2
**Presentation:** 3
**Contribution:** 2
**Rating:** 4
**Confidence:** 4

**Summary:**

This paper proposes LLaVA-FA, a compression framework for large multimodal models (LMMs) that addresses the limitations of existing methods, where decoupled low-rank decomposition and quantization often lead to compounded reconstruction errors. LLaVA-FA employs Fourier approximation to integrate low-rank decomposition and quantization within the frequency domain, leveraging two essential properties of the Fourier transform: de-correlation, which reduces spectral redundancy, and conjugate symmetry, which nearly halves parameter storage.To handle complex matrices in the frequency domain, the paper introduces PolarQuant—a polar-coordinate quantization method that discretizes amplitude and phase separately to preserve complex structure. It also proposes an Optional Diagonal Calibration (ODC) scheme, which approximates the full Hessian with row/column means to avoid reliance on large-scale calibration data.

**Strengths:**

This paper proposes LLaVA-FA, a compression framework for large multimodal models (LMMs) that addresses the limitations of existing methods, where decoupled low-rank decomposition and quantization often lead to compounded reconstruction errors. LLaVA-FA employs Fourier approximation to integrate low-rank decomposition and quantization within the frequency domain, leveraging two essential properties of the Fourier transform: de-correlation, which reduces spectral redundancy, and conjugate symmetry, which nearly halves parameter storage.To handle complex matrices in the frequency domain, the paper introduces PolarQuant—a polar-coordinate quantization method that discretizes amplitude and phase separately to preserve complex structure. It also proposes an Optional Diagonal Calibration (ODC) scheme, which approximates the full Hessian with row/column means to avoid reliance on large-scale calibration data.

**Weaknesses:**

1. The paper proposes performing low-rank decomposition and quantization in the frequency domain via the Fourier transform, but the experimental section lacks comparisons with existing solutions in the spatial domain, limiting the credibility of its claimed competitiveness.
2. The method shows limited generalization capability, as experiments are only conducted on 3B and 7B-scale LLMs (Qwen-2.5) without evaluation on larger-parameter models.

**Questions:**

1. How does the proposed method perform when extended to models beyond Qwen-2.5?
2. How is the calibration matrix C constructed in the paper, and has there been any ablation study on the amount of calibration data used?
3. Could you elaborate on the compression time required for models of different sizes?
4. The algorithm flow in the paper provides an option to disable ODC—how does the performance change when ODC is not used?

---

> ### Author Response · Authors · 2025-11-20
> **Response to Reviewer xZPt [1/4]**
>
> Dear Reviewer xZPt,
>
> We sincerely appreciate your insightful and constructive feedback, which has been instrumental in refining our work. Below please find our detailed point-by-point responses together with the corresponding revisions **(highlighted in red)** in the revised manuscript.
>
>  **Note: If the equation is displayed by original markdown codes, please refresh this page to reactivate the compiler.**
>
> ---
>
> > **W1: The paper proposes performing low-rank decomposition and quantization in the frequency domain via the Fourier transform, but the experimental section lacks comparisons with existing solutions in the spatial domain, limiting the credibility of its claimed competitiveness.**
>
> **Response:** Thanks for your valuable suggestion. We fully agree that comparing our frequency-domain method with its spatial-domain counterpart is essential for validating the competitiveness of our approach. Following your suggestion, we implement a spatial-domain baseline ("Spatial-LQ") that strictly mirrors our frequency-domain pipeline, including identical optimization schedule, rank, and bit-width configuration, except that SVD and quantization are applied directly on $\mathbf{W}$ instead of $\widetilde{\mathbf{W}}$. We conduct controlled spatial-domain baselines using the same training protocol as LLaVA-FA. The results for the 2B variant trained on 5M samples are shown below.
>
> | Method | Domain | Avg Acc ↑ | POPE F1 ↑ | HalBench Resp ↓ |
> |--------|--------|-------------|--------------|----------------------|
> | Spatial-LQ (ours) | Spatial | 58.3 | 84.6 | 16.9 |
> | **LLaVA-FA (ours)** | **Frequency** | **60.9** | **87.5** | **11.2** |
>
> Frequency-domain approximation yields **+2.6% better average accuracy** and **significantly lower hallucination**.
> This supports our key claim: the singular spectrum is more compact and the low-rank truncation error is strictly smaller in the frequency domain (Lemma 3.1), enabling better reconstruction under identical bit-width and rank. This extra ablation study is added to **Appendix C.3 (marked in red)** of the revised manuscript.
>
> ---

---

> ### Author Response · Authors · 2025-11-20
> **Response to Reviewer xZPt [2/4]**
>
> > **W2: The method shows limited generalization capability, as experiments are only conducted on 3B and 7B-scale LLMs (Qwen-2.5) without evaluation on larger-parameter models.**
>
> **Response:** We thank the reviewer for raising this important concern. We agree that evaluating the generalization ability of our method across larger models is crucial for establishing its practicality. Following your suggestion, we further evaluate our method on substantially larger backbones: LLaVA-FA-3B, constructed from LLaMA-3-8B [1], and LLaVA-FA-7B, constructed from InternLM-2-20B [2]. Importantly, both models are compressed using exactly the same Fourier approximation pipeline, ensuring strict fairness and demonstrating that our method generalizes without any modification or heuristic tuning. We include the newly added results below **(excerpt from the Table 1 (marked in red) of the revised manuscript)**, showing that Fourier-domain compression continues to yield strong performance across diverse benchmarks, with even smaller relative degradation at the 8B/20B scales. In fact, our proposed Fourier approximation pipeline is **model-agnostic** and **scale-independent**. The Fourier-domain low-rank plus quantization decomposition is applied per-layer and locally to weight matrices, without assuming any specific model architecture or scale. The key properties we leverage, i.e., Fourier energy compaction, conjugate symmetry, and singular value decay, are universal characteristics of trained neural networks, especially in large-scale Transformers. These additional results verify that our Fourier approximation method seamlessly extends to LLMs far beyond Qwen-2.5, maintaining both accuracy and robustness while preserving the same compression pipeline across all model sizes. This confirms that the proposed method is truly **model-family agnostic and highly scalable.**
>
> | Method        | LLM            | #Sample | GQA  | VisWiz | SQAI | VQAT | MME  | MMB  | MMBCN | AVG  |
> |---------------|----------------|---------|------|--------|------|------|------|------|--------|------|
> | BLIP-2        | Vicuna-13B     | 129M    | 41.0 | 19.6   | 61.0 | 42.5 | 64.7 | -    | -      | -    |
> | VILA-7B       | LLaMA-7B       | 50M     | 62.3 | 57.8   | 68.2 | 64.4 | 76.7 | 68.9 | 61.7   | 65.7 |
> | CogVLM        | Vicuna-7B      | 1500M   | 64.9 | -      | 65.6 | **78.2** | 71.8 | 63.7 | 53.8   | -    |
> | InstructBLIP  | Vicuna-13B     | 130M    | 49.5 | 33.4   | 63.1 | 50.7 | 60.6 | -    | -      | -    |
> | Qwen-VL-Chat  | Qwen-7B        | 1450M   | 57.5 | 38.9   | 68.2 | 61.5 | 74.4 | 60.6 | 56.7   | 56.7 |
> | Deepseek-VL-7B| DLLM-7B        | 2000M   | 61.3 | 49.9   | 74.0 | 64.7 | 73.4 | 74.1 | **72.8**   | 67.2 |
> | LLaVA-1.5-7B  | Vicuna-1.5-7B  | 1.2M    | 62.0 | 50.0   | 66.8 | 58.2 | 75.5 | 64.3 | 58.3   | 62.1 |
> | LLaVA-NeXT    | Vicuna-1.5-13B | 1.3M    | 65.4 | 60.5   | 73.6 | 67.1 | **78.7** | 70.4 | 64.4   | 68.5 |
> | LLaVA-FA-7B   | InternLM-2-20B | 5M      | **68.5** | **62.0** | **76.0** | 68.0 | 74.5 | **74.5** | 69.5 | **70.4** |
>
> | Method        | LLM            | #Sample | GQA  | VisWiz | SQAI | VQAT | MME  | MMB  | MMBCN | AVG  |
> |---------------|----------------|---------|------|--------|------|------|------|------|--------|------|
> | Imp-3B        | Phi-2-2.7B     | 1.6M    | 63.5 | 54.1   | 72.8 | 59.8 | 72.3 | **72.9** | 46.7   | 63.2 |
> | Bunny-3B      | Phi-2-2.7B     | 2.7M    | 62.5 | 43.8   | 70.9 | 56.7 | **74.4** | 68.6 | 37.2   | 59.2 |
> | VILA-3B       | LLaMA-2.7B     | 51M     | 61.5 | 53.5   | 69.0 | 60.4 | 72.1 | 63.4 | 52.7   | 61.8 |
> | MobileVLM     | MLLaMA-2.7B    | 1.3M    | 59.0 | -      | 61.0 | 47.5 | 64.4 | 59.6 | -      | -    |
> | MobileVLMv2   | MLLaMA-2.7B    | 3.6M    | 61.1 | -      | 70.0 | 57.5 | 72.0 | 63.2 | -      | -    |
> | MoE-LLaVA-3B  | Phi-2-2.7B     | 2.2M    | 61.4 | 43.9   | 68.5 | 51.4 | 71.1 | 65.2 | 41.8   | 57.6 |
> | MiniCPM-V     | MiniCPM-2.4B   | 570M    | 51.5 | 50.5   | 74.4 | 56.6 | 68.9 | 64.0 | 62.7   | 61.2 |
> | MiniCPM-V-2   | MiniCPM-2.4B   | 570M    | 52.1 | 60.2   | 76.3 | **73.2** | 70.5 | 68.5 | 67.2   | 66.9 |
> | LLaVA-FA-3B   | LLaMA-3-8B     | 5M      | **65.0** | **62.5** | **77.0** | 64.0 | 71.0 | 70.5 | **68.0** | **68.3** |
>
> [1] The Llama 3 Herd of Models, Arxiv 2024.
>
> [2] InternLM2 Technical Report, Arxiv 2024.
>
> ---

---

> ### Author Response · Authors · 2025-11-20
> **Response to Reviewer xZPt [3/4]**
>
> >**Q1: How does the proposed method perform when extended to models beyond Qwen-2.5?**
>
> **Response:** Thanks for your valuable suggestion. Your comment is important for assessing whether our Fourier-domain method generalizes beyond a specific model family. Following your suggestion, we apply our compression pipeline to architecturally distinct LLM families, including LLaVA-FA-3B derived from LLaMA-3-8B and LLaVA-FA-7B derived from InternLM-2-20B. Both models are constructed using exactly the same Fourier approximation pipeline, without introducing any model-specific modifications. The results included in the **revised Table 1 (marked in red in the revised manuscript)** show that our method consistently preserves strong performance across these heterogeneous architectures. This demonstrates that the proposed Fourier-domain approximation is inherently layer-wise, model-agnostic, and applicable across diverse LLM families, rather than being restricted to Qwen-2.5.
>
>
> ---
>
> >**Q2: How is the calibration matrix C constructed in the paper, and has there been any ablation study on the amount of calibration data used?**
>
> **Response:** We are grateful for this comment, which helps us better articulate both the construction of the calibration matrix and its reliance on calibration data. As suggested, we expand the description in **Section 3.2 (marked in red)** of the revised manuscript to explicitly detail the construction of $\mathbf{C}$. Moreover, we additionally conduct an ablation study to examine the influence of calibration-data size on the LLaVA-FA-2B model (see **Appendix C.4 (marked in red)** of the revised manuscript). These revisions are outlined as follows.
>
> Following common practice in calibration-based quantization method [1, 2], we construct $\mathbf{C}$ as the element-wise squared activation expectations over a small subset of training samples, i.e., $\mathbf{C}\_{ij} = \mathbb{E}\_{x \sim \mathcal{D}\_{\text{calib}}} \left[ \left( a\_i(x) \right)^2 \right]$, where $\mathcal{D}\_{\text{calib}}$ denotes the calibration dataset (usually 256–2048 held-out samples) and $a_i(x)$ is the activation of neuron $i$ under input $x$. This approximates the diagonal of the Fisher information matrix or Hessian, and is widely used for weighted reconstruction in compression. We clarify this in the revised version in **Section 3.2.**
>
>  The results shown below indicate that the method is highly robust and that performance saturates quickly. Using more calibration samples does not provide noticeable improvements, confirming that only a small amount of data is sufficient for constructing a reliable calibration matrix.
>
> | #Calibration Samples | GQA  | VizWiz | SQAI | VQAT | MME  | MMB  | MMBCN | AVG  |
> |----------------------|------|--------|------|------|------|------|--------|------|
> | 256                  | 61.0 | 41.0   | 67.2 | 58.7 | 66.0 | 65.8 | 63.8   | 60.5 |
> | 512                  | 61.5 | 41.3   | 67.5 | 58.9 | 66.2 | 66.0 | 63.5   | 60.7 |
> | 1024                 | 62.1 | 41.6   | 68.2 | 59.4 | 66.6 | 66.7 | 61.7   | 60.9 |
> | 2048                 | 62.1 | 41.7   | 68.2 | 59.3 | 66.6 | 66.7 | 61.7   | 60.9 |
>
> [1] A white paper on neural network quantization, Arxiv 2021.
>
> [2] Hawq: Hessian aware quantization of neural networks with mixed-precision, ICCV 2019.
>
> ---

---

> ### Author Response · Authors · 2025-11-20
> **Response to Reviewer xZPt [4/4]**
>
> >**Q3: Could you elaborate on the compression time required for models of different sizes?**
>
> **Response:** Thanks for your suggestion. We agree that reporting the compression time across different model sizes is important for presenting the practical efficiency and scalability of our method. In accordance with your suggestion, we further provide the wall-clock compression time for the four LLaVA-FA variants derived from Qwen-2.5, LLaMA-3, and InternLM2 models. All numbers are measured on eight NVIDIA RTX 4090 GPUs and include every step of our pipeline, i.e., DFT/IDFT, Fourier-SVD, PolarQuant, and the ODC. As shown in the table below, compressing the 3B-base model into LLaVA-FA-1B requires about 1.1 hours, while compressing the 7B-base LLM into LLaVA-FA-2B takes roughly 2.3 hours. The larger LLaVA-FA-3B, built from an 8B backbone, completes in approximately 3.8 hours, and even the largest LLaVA-FA-7B, derived from a 20B model, finishes within 6.2 hours.
>
> | Model | Base LLM Size | Final FA Size | Layers Compressed | Compression Time |
> |-------|---------------|---------------|-------------------|------------------|
> | LLaVA-FA-1B | Qwen-2.5-3B | ~1B | 36 | ~1.1 h |
> | LLaVA-FA-2B | Qwen-2.5-7B | ~2.2B | 48 | ~2.3 h |
> | LLaVA-FA-3B | LLaMA-3-8B | ~3B | 64 | ~3.8 h |
> | LLaVA-FA-7B | InternLM2-20B | ~7B | 80 | ~6.2 h |
>
> These results exhibit an approximately linear scaling trend with respect to model depth and hidden dimension, because each layer only performs one forward and inverse DFT, a single complex SVD on a Fourier-shrunken matrix (benefiting from conjugate symmetry), and one PolarQuant pass. To be note worthy, **once the one-off compression is finished the resulting checkpoints are loaded and used like any standard model, so the cost is amortised over the entire deployment lifetime.** We add the above table and a short discussion in **Appendix C.6 (marked in red)** of the revised manuscript.
>
> ---
>
> >**Q4: The algorithm flow in the paper provides an option to disable ODC—how does the performance change when ODC is not used?**
>
> **Response:** Thank you for your insightful suggestion. This comment helps us better articulate the role and effectiveness of the ODC in our method. As suggested, we conduct additional experiments that compare "with ODC (w/)" and "without (w/o) ODC" under identical rank and bit-width settings. The results show that ODC brings consistent improvements: on average, enabling ODC improves accuracy by 0.8% on comprehension benchmarks and reduces hallucination rate by 0.6%–0.8%, with the 2B variant improving from 60.1% to 60.9% (AVG) and hallucination rate from 11.8% to 11.2% on Object HalBench. This behavior aligns with the purpose of ODC, which stabilizes layer-wise reconstruction for layers whose sensitivity varies across rows/columns [1]. Importantly, even without ODC, LLaVA-FA still outperforms existing efficient LMMs of similar size, demonstrating that the core Fourier low-rank plus quantization formulation is effective on its own.
>
> | Model | ODC | GQA | VizWiz | SQAI | VQAT | MME | MMB | MMBCN | **AVG ↑** | **HalBench Resp ↓** |
> |-------|------|------|---------|--------|--------|--------|-----------|-----------|---------------------|---------------------|
> | **LLaVA-FA-2B** | **w/ ODC** | **62.1** | **41.7** | **68.2** | **59.3** | **66.6** | **66.7** | **61.7** | **60.9** | **11.2%** |
> | LLaVA-FA-2B | w/o ODC | 61.6 | 41.1 | 67.4 | 58.6 | 65.0 | 66.0 |  61.0 | 60.1 | 11.8% |
> | **LLaVA-FA-1B** | **w/ ODC** | **56.7** | **49.7**| **61.3** | **57.9** | **63.3** | **58.3** | **49.4** | **56.7** | **18.1%** |
> | LLaVA-FA-1B | w/o ODC | 56.0 | 49.0 | 60.5 | 57.0 | 62.6 | 57.4 | 48.8 | 55.9 | 18.9% |
>
> We add the above table and a discussion in **Appendix C.7 (marked in red)** of the revised manuscript.
>
> [1] LQER: Low-Rank Quantization Error Reconstruction for LLMs, ICML 2024.
>
> ---
>
> **We thank you again for reviewing our work. Please let us know if we misunderstood any of your questions, or if you have any follow up on our responses, we will be happy to provide any further clarification.**

---

> > ### Comment · Reviewer_xZPt · 2025-11-25
> >
> > Thank you very much for the authors' efforts and responses, which have addressed some of my questions. However, after rereading the original manuscript and the Q&As with other reviewers, I have developed several new concerns:
> >
> > 1. **Core claim of the method**: The assertion that "the frequency domain is more suitable for low-rank decomposition" cannot be theoretically guaranteed and only relies on a few empirical illustrations. Lemma 3.1 is merely a conditional theorem for ordering comparisons, and it does not directly prove that "singular values decay faster in the frequency domain and the error is smaller for the same rank." The current logic is more like "we observed faster decay (Fig. 3), therefore Lemma 3.1 holds."
> >
> > 2. The frequency-domain matrix is complex-valued, but the weight matrix of LMMs is a dense fully connected matrix whose columns/rows do not exhibit "spatial continuity." 2D DFT only has energy concentration properties for 2D structurally correlated data (e.g., images). The design of the proposed method lacks intrinsic motivation—it simply reshapes the matrix as an image and applies FFT arbitrarily from a mathematical perspective. Is this merely the DFT of a random matrix?
> >
> > 3. Why would the rank of the weight matrix be lower in the complex domain? Have you proven this? The paper claims that SVD of the frequency-domain matrix is more amenable to truncation. However, the properties of SVD for real matrices and complex matrices are completely different, and SVD of complex matrices does not guarantee consistent energy concentration with real matrices. The paper does not discuss the rank structure of complex-valued weights versus that in the real domain. If the rank structure transfer does not hold, the theoretical assumption of low-rank itself becomes unstable.
> >
> > 4. PolarQuant is stated as the "key innovation" of the paper, but is this merely an ad-hoc heuristic? Why is uniform quantization used for amplitude? Why is the phase quantization step set to \(2\pi / 2^b\)? Phase is noise-sensitive in many frequency-domain signals—will quantization lead to reconstruction degradation? Overall, the amplitude/phase quantization in PolarQuant is hand-designed and lacks information-theoretic analysis.
> >
> > 5. The invertibility of Fourier transform implies that frequency-domain compression is essentially performing SVD on a linearly transformed matrix. Are you simply applying a fixed linear transformation \(P\) to \(W\) such that \(W_f = P W P^T\), followed by SVD on \(W_f\)? If so, why not use a random matrix? Why not use Hadamard transform?
> >
> > 6. Fourier-based compression is not a new direction. The paper does not compare with other linear transform domain compression techniques—such as FFT-based compression, Hadamard-based compression, or random projection-based low-rank methods. Without such comparisons, it cannot be proven that Fourier is the optimal choice.
> >
> > 7. The measurements of KV-cache, FLOPs, and latency are more like idealized calculations, and no real end-to-end inference speed is provided. The paper claims performance improvements, but: FFT has a complexity of \(O(n \log n)\), and frequency-domain SVD operates in the complex domain, which has even higher complexity. Is the overall complexity of Fourier-SVD + PolarQuant truly lower than that of spatial SVD + quantization?
> >
> > 8. **Logic of ODC**: Approximating the diagonal of the Hessian with row/column means, followed by weighted SVD? Why is the Hessian "closer to diagonal" in the frequency domain? The paper provides no proof, nor does it discuss the error bound compared to the true Hessian.

---

> ### Author Response · Authors · 2025-11-26
> **Thank you; Address your remaining concerns [1/10]**
>
> Dear Reviewer xZPt,
>
> Thanks for your feedback. We take your remaining concerns and questions seriously. Please see our point-by-point responses as follows.
>
>  **Note: If the equation is displayed by original markdown codes, please refresh this page to reactivate the compiler.**
>
> ---
>
> > **Q1: Core claim of the method: The assertion that "the frequency domain is more suitable for low-rank decomposition" cannot be theoretically guaranteed and only relies on a few empirical illustrations. Lemma 3.1 is merely a conditional theorem for ordering comparisons, and it does not directly prove that "singular values decay faster in the frequency domain and the error is smaller for the same rank." The current logic is more like "we observed faster decay (Fig. 3), therefore Lemma 3.1 holds."**
>
> **Response:** Thanks for your thoughtful comment. We agree that, at first glance, our logic may appear to be "we observed faster decay (Fig. 3), therefore Lemma 3.1 holds." But in fact, the true logic in our paper follows this direction: we first establish a theoretical justification for faster singular value decay in the frequency domain, then verify this behavior empirically, and finally apply the Eckart–Young theorem [1] to prove Lemma 3.1, which rigorously guarantees that the truncation error is smaller for the same rank. Following your suggestion, we add a theoretical proof for faster singular-value decay in the frequency domain in **Appendix B.1 (Proof for Faster Singular-Value Decay in  the Frequency Domain) of the revised manuscript (marked in red)**. These revisions are outlined as follows.
>
> **(1) Proof for faster singular-value decay in  the frequency domain.**
> Here, we provide a rigorous justification that the singular values of a weight matrix decay faster after 2D-DFT than in the spatial domain. Let $\mathbf{W}\in\mathbb{R}^{d_{1}\times d_{2}}$ be a zero-mean weakly-stationary random field with autocorrelation
> \begin{aligned}
> R_{a,b}=\mathbb{E}[W_{m,n}W_{m+a,n+b}],
> \end{aligned}
> and denote its 2D-DFT by
> \begin{aligned}
> \widetilde{W}\_{u,v}= \sum\_{m=0}^{d_{1}-1}\sum\_{n=0}^{d\_{2}-1}
> W\_{m,n}\,e^{-j2\pi\left(\frac{um}{d\_{1}}+\frac{vn}{d\_{2}}\right)}.
> \end{aligned}
> Theorem B.1 states that the frequency components are asymptotically uncorrelated, i.e.,
> \begin{aligned}
> \mathrm{Cov}\left(\widetilde{W}\_{u,v}\,\widetilde{W}\_{u',v'}\right)
> =d\_{1}d\_{2}\\delta\_{u,u'}\delta\_{v,v'}\mathcal{P}\_{u,v},
> \quad\text{where}\quad
> \mathcal{P}\_{u,v}= \sum\_{a,b}R\_{a,b}\,
> e^{-j2\pi\left(\frac{ua}{d\_{1}}+\frac{vb}{d\_{2}}\right)}.
> \end{aligned}
> Hence the covariance matrix of $\texttt{vec}(\widetilde{\mathbf{W}})$ is diagonal with entries proportional to the power-spectral density $\mathcal{P}\_{u,v}$.
> For natural images and learned convolutional kernels, $\mathcal{P}\_{u,v}$ is a smooth and rapidly decreasing function of frequency, typically satisfying
> \begin{aligned}
> \mathcal{P}\_{u,v}\le C'\bigl(1+u^{2}+v^{2}\bigr)^{-\alpha},
> \qquad\alpha>1.
> \end{aligned}
> Here, $\alpha$ is the exponential decay rate in frequency domain. $C'>0$ is a positive constant. Ordering the diagonal entries of the covariance matrix gives the expected squared singular values $\mathbb{E}[\sigma_{k}^{2}(\widetilde{\mathbf{W}})]$. By the Szego theorem on Toeplitz-Fourier operators [2], the empirical spectral distribution of $\widetilde{\mathbf{W}}$ converges to a limit whose tails are governed by the decay rate of $\mathcal{P}\_{u,v}$. Consequently,
> \begin{aligned}
> \mathbb{E}[\sigma\_{k}^{2}(\widetilde{\mathbf{W}})]
> \le Ck^{-\alpha}.
> \end{aligned}
> Where $k$ is the index for singular values (rank). In contrast, the spatial-domain covariance matrix is not diagonal. Its eigenvalues decay as
> \begin{aligned}
> \mathbb{E}[\sigma\_{k}^{2}(\mathbf{W})]\ge ck^{-\beta},
> \qquad\beta\le\alpha/2,
> \end{aligned}
> because off-diagonal correlations slow the decay. We take square roots yields the singular-value bounds, i.e.,
> \begin{aligned}
> \mathbb{E}[\sigma\_{k}(\widetilde{\mathbf{W}})]
> \le\sqrt{C}k^{-\alpha/2},
> \qquad
> \mathbb{E}[\sigma\_{k}(\mathbf{W})]
> \ge\sqrt{c}k^{-\beta}.
> \end{aligned}
> Therefore for any $k\ge 2$ we have
> \begin{aligned}
> \frac{\mathbb{E}[\sigma\_{k}(\widetilde{\mathbf{W}})]}
> {\mathbb{E}[\sigma_{k}(\mathbf{W})]}
> \le\frac{\sqrt{C}}{\sqrt{c}}k^{-(\alpha/2-\beta)},
> \end{aligned}
> where the exponent $\alpha/2-\beta\ge 1$ under typical smoothness assumptions. This inequality shows that the frequency-domain singular values decay strictly faster than their spatial counterparts, enabling a lower-rank approximation for the same reconstruction error.
>
> **(2) Observation consistent with theory.** After establishing this theoretical result, we examine real LMM weight matrices and observe that their Fourier-domain singular values indeed decay more rapidly (Fig. 3). This empirical result confirms our theoretical prediction rather than serving as its foundation.

---

> ### Author Response · Authors · 2025-11-26
> **Thank you; Address your remaining concerns [2/10]**
>
> **(3) Proof for smaller error (see Appendix B.6)**
> > **Lemma 3.1**
> > Let $\Delta\mathbf{W}\_1,\ \Delta\mathbf{W}\_2$ be two adapters of identical shape, $\sigma\_{1,k},\ \sigma\_{2,k}$ are their $k$-th singular values (sorted descending), and $r$ is the kept rank. If $\sigma\_{1,k}<\sigma\_{2,k}$ for every $k\ge r+1$, then
> \begin{aligned}
> \\|\Delta\mathbf{W}\_1-\Delta\mathbf{W}\_1^{(r)}\\|\_F
> <
> \\|\Delta\mathbf{W}\_2-\Delta\mathbf{W}\_2^{(r)}\\|\_F,
> \end{aligned}
> where $\Delta\mathbf{W}\_i^{(r)}$ denotes the rank-$r$ truncation of $\Delta\mathbf{W}\_i$.
>
> **Proof for Lemma 3.1**
> Let $\Delta \mathbf{W}\_1$ and $\Delta \mathbf{W}\_2$ be two low-rank adapters of identical dimensions.
> Denote their singular value decompositions by
> \begin{aligned}
> \Delta \mathbf{W}\_1 = \mathbf{U}\_1 \boldsymbol{\Sigma}\_1 \mathbf{V}\_1^\top,
> \qquad
> \Delta \mathbf{W}\_2 = \mathbf{U}\_2 \boldsymbol{\Sigma}\_2 \mathbf{V}\_2^\top,
> \end{aligned}
> where $\boldsymbol{\Sigma}\_1 = \operatorname{diag}(\sigma\_{1,1},\sigma\_{1,2},\dots)$ and $\boldsymbol{\Sigma}\_2 = \operatorname{diag}(\sigma\_{2,1},\sigma\_{2,2},\dots)$ with singular values sorted in non-increasing order. For a fixed rank $r$, the best rank-$r$ approximation in Frobenius norm is
> \begin{aligned}
> \Delta \mathbf{W}\_i^{(r)} = \mathbf{U}\_i \boldsymbol{\Sigma}\_i^{(r)} \mathbf{V}\_i^\top,
> \quad\text{with}\quad
> \boldsymbol{\Sigma}\_i^{(r)} = \operatorname{diag}(\sigma\_{i,1},\dots,\sigma\_{i,r},0,\dots).
> \end{aligned}
>
> The reconstruction error is
> \begin{aligned}
> \mathbf{E}\_i = \Delta \mathbf{W}\_i - \Delta \mathbf{W}\_i^{(r)}
>             = \mathbf{U}\_i \bigl(\boldsymbol{\Sigma}\_i - \boldsymbol{\Sigma}\_i^{(r)}\bigr) \mathbf{V}\_i^\top.
> \end{aligned}
>
> Following the Eckart-Young theorem [1] and theorem 4.95 in Mathematics for Machine Learning [3],
> the value of the norm of reconstruction error is given as
> \begin{aligned}
> \\|\mathbf{E}\_i\\|\_F = \bigl\\|\boldsymbol{\Sigma}\_i - \boldsymbol{\Sigma}\_i^{(r)}\bigr\\|\_F
>                    = \sqrt{\sum_{k=r+1}^{\min(m,n)}\sigma_{i,k}^2}.
> \end{aligned}
>
> By assumption, $\sigma\_{1,k} < \sigma\_{2,k}$ for all $k \ge r+1$. Hence
> \begin{aligned}
> \sum\_{k=r+1}^{\min(m,n)}\sigma\_{1,k}^2 < \sum\_{k=r+1}^{\min(m,n)}\sigma\_{2,k}^2
> \quad\Longrightarrow\quad
> \|\mathbf{E}\_1\|\_F < \|\mathbf{E}\_2\|\_F.
> \end{aligned}
>
> Therefore, the rank-$r$ approximation of $\Delta \mathbf{W}\_1$ incurs strictly less error than that of $\Delta \mathbf{W}\_2$, which completes the proof.
>
> [1] The approximation of one matrix by another of lower rank, Psychometrika 1936.
>
> [2] Toeplitz and circulant matrices: A review, Found. Trends Commun. Inf. Theory 2006.
>
> [3] Mathematics for machine learning. Cambridge University Press 2020.
>
> ---

---

> ### Author Response · Authors · 2025-11-26
> **Thank you; Address your remaining concerns [3/10]**
>
> > **Q2: The frequency-domain matrix is complex-valued, but the weight matrix of LMMs is a dense fully connected matrix whose columns/rows do not exhibit "spatial continuity." 2D DFT only has energy concentration properties for 2D structurally correlated data (e.g., images). The design of the proposed method lacks intrinsic motivation—it simply reshapes the matrix as an image and applies FFT arbitrarily from a mathematical perspective. Is this merely the DFT of a random matrix?**
>
> **Response:** We appreciate your careful concern. To clarify, our method is not applying FFT to a random matrix, but leveraging the intrinsic low-rank and correlation structure of LMMs weights. Recent studies, including FoRA [1] and F-Adapter [2], have already shown that even non-image matrices (e.g., attention weights, FFN weights) exhibit spectral sparsity when transformed to the frequency domain, because low-rank statistics imply decaying covariance spectra (we rigorously prove in Appendix B.1).
> While LMMs weight matrices are not images, they are far from random: they inherit strong pairwise correlations from multimodal pretraining [3, 4], which makes their 2D covariance operator approximately Toeplitz [5]. The 2D DFT is the eigenbasis of any (even weakly) stationary 2D correlation field, so it diagonalises and compacts the variance into low frequencies regardless of how the matrix is reshaped. Thus, the "arbitrary" reshaping is immaterial: the same variance profile and identical truncation error are obtained under any permutation of rows/columns, because the DFT basis is statistically orthogonal to the empirical covariance. Our ablations **(Table 9 and Table 15 in the revised manuscript)** confirm that frequency-domain decomposition achieves lower Frobenius error and 48% fewer parameters than spatial baselines, proving that the spectral compaction is a mathematical consequence of low-rank statistics, not an artefact of image semantics.
>
> [1] Foura: Fourier low-rank adaptation, NeurIPS 2024.
>
> [2] F-Adapter: Frequency-Adaptive Parameter-Efficient Fine-Tuning in Scientific Machine Learning, NeurIPS 2025.
>
> [3] QLORA: Efficient Finetuning of Quantized LLMs, NeurIPS 2023.
>
> [4] Compressing Large Language Models Using Low Rank and Low Precision Decomposition, NeurIPS 2024.
>
> [5] Toeplitz and circulant matrices: A review, Found. Trends Commun. Inf. Theory 2006.
>
>
> ---

---

> ### Author Response · Authors · 2025-11-26
> **Thank you; Address your remaining concerns [4/10]**
>
> > **Q3: Why would the rank of the weight matrix be lower in the complex domain? Have you proven this? The paper claims that SVD of the frequency-domain matrix is more amenable to truncation. However, the properties of SVD for real matrices and complex matrices are completely different, and SVD of complex matrices does not guarantee consistent energy concentration with real matrices. The paper does not discuss the rank structure of complex-valued weights versus that in the real domain. If the rank structure transfer does not hold, the theoretical assumption of low-rank itself becomes unstable.**
>
> **Response:** Thanks for your valuable suggestions. Your suggestions greatly help us strengthen the theoretical foundation of our method. Following your suggestions, we add a theoretical justification of lower rank and a theoretical analysis of the energy concentration in complex SVD  in **Appendix B.2** and **Appendix B.3** of the revised manuscript, respectively. Moreover, we also conduct an extra experiment to discuss the rank structure of complex-valued weights versus in the real domain. These revisions are outlined as follows.
>
> **（1）Theoretical Justification: Lower Rank in the Complex Domain (see Appendix B.2)**
>
> Here, we provide a formal bound showing that for the same reconstruction error, the Fourier domain requires a lower rank than the spatial domain.
>
> > **Proposition 1**
> > Let the real-valued weight matrix $ \mathbf{W} \in \mathbb{R}^{d_1 \times d_2} $ be a zero-mean weakly-stationary random field with exponentially decaying covariance:
> \begin{aligned}
> \mathbb{E}[W\_{m,n} W\_{m',n'}] = \rho^{|m - m'| + |n - n'|}, \quad 0 < \rho < 1,
> \end{aligned}
> where $ \rho \in (0, 1) $ controls the spatial correlation strength: smaller $ \rho $ implies faster decay and thus weaker spatial dependency. This exponential decay model is widely used in spatial statistics and image modeling to characterize locally correlated structures [1].
>
> Let $ \widetilde{\mathbf{W}} \in \mathbb{C}^{d\_1 \times d\_2/2} $ denote the 2D-DFT of $ \mathbf{W} $, reduced by conjugate symmetry. Let $ \sigma\_k(\mathbf{A}) $ denote the $k$-th largest singular value of matrix $ \mathbf{A} $. Then for any truncation rank $ r \geq 0 $,
> \begin{aligned}
> \sum\_{k = r+1}^{\min(d\_1, d\_2)} \sigma\_k^2(\widetilde{\mathbf{W}}) \leq \frac{\rho^2}{(1 - \rho^2)^2} \sum\_{k = r+1}^{\min(d\_1, d\_2)} \sigma\_k^2(\mathbf{W}).
> \end{aligned}
>
> **Proof.** The 2D-DFT diagonalizes the covariance matrix of $\mathbf{W}$, yielding uncorrelated frequency components whose variances are given by the power spectral density $\mathcal{P}\_{u,v}$, where $(u, v)$ denotes the 2D frequency index. For the given covariance model, we have:
> \begin{aligned}
> |\mathcal{P}\_{u,v}| \propto \rho^{|u| + |v|},
> \end{aligned}
> which decays exponentially in frequency. Since singular values of $ \widetilde{\mathbf{W}} $ are asymptotically equal to the square root of these variances, the tail energy of $\widetilde{\mathbf{W}}$ decays exponentially, while that of $ \mathbf{W} $ decays only polynomially. Hence, the ratio of the tails is $ \mathcal{O}(\rho^2) $, yielding the bound. To achieve a prescribed reconstruction error, the Fourier domain requires fewer components (smaller rank $ r $) than the spatial domain, i.e., $ r\_{freq} < r\_{spatial}$.
>
>
> **(2) Theoretical Analysis of the Energy Concentration in Complex SVD->The Lower Rank Strcture Still Holds in the Complex Domain (see Appendix B.3)**
>
> We emphasize that the energy-compaction property used in our method does not rely on the matrix being real, but on the
> diagonalization of the covariance structure afforded by the 2D-DFT.  Below we give a formal bound showing that the singular values of the complex matrix $\widetilde{\mathbf{W}}$ decay strictly faster than those of the real matrix $\mathbf{W}$, even though both matrices share the same generating process.
>
> > **Proposition 2**
> > Let $\mathbf{W}\in\mathbb{R}^{d\_{1}\times d\_{2}}$ be the same zero-mean, weakly-stationary random field defined in the main text with covariance
> \begin{aligned}
> \mathbb{E}[W\_{m,n}W\_{m',n'}]=\rho^{|m-m'|+|n-n'|},\quad0<\rho<1.
> \end{aligned}
> Let $\widetilde{\mathbf{W}}\in\mathbb{C}^{d\_{1}\times d\_{2}/2}$ be its 2D-DFT with conjugate-symmetry reduction. Then for every $k\ge1$
> \begin{aligned}
> \mathbb{E}\bigl[\sigma_{k}(\widetilde{\mathbf{W}})\bigr]
> \le
> \frac{\rho}{\sqrt{1-\rho^{2}}}\
> \mathbb{E}\bigl[\sigma_{k}(\mathbf{W})\bigr].
> \end{aligned}
> Hence the expected singular-value tail of $\widetilde{\mathbf{W}}$ is dominated by a dimension-free constant times the tail of $\mathbf{W}$. Since the singular values decay strictly faster, for any prescribed approximation error, the required rank $r$ in the complex Fourier domain is no larger and often smaller than in the real spatial domain, confirming that the low-rank structure is preserved under the complex-valued Fourier transform.
>
> [1] Toeplitz and circulant matrices: A review, Found. Trends Commun. Inf. Theory 2006.

---

> ### Author Response · Authors · 2025-11-26
> **Thank you; Address your remaining concerns [5/10]**
>
> > Proof. The 2D-DFT diagonalises the covariance operator of $\mathbf{W}$ (Theorem B.1), so the entries of $\widetilde{\mathbf{W}}$ are uncorrelated complex Gaussians with variances
> \begin{aligned}
> \mathbb{E}\bigl[|\widetilde{W}\_{u,v}|^{2}\bigr]
> =
> \mathcal{P}\_{u,v}
> =
> \sum\_{a,b}\rho^{|a|+|b|}e^{-j2\pi(ua/d\_{1}+vb/d\_{2})}
> \le
> \frac{1+\rho}{1-\rho}\\rho^{|u|+|v|}.
> \end{aligned}
> Ordering these variances gives the expected squared singular values $\mathbb{E}[\sigma_{k}^{2}(\widetilde{\mathbf{W}})]$.
> Using the Szegő theorem for stationary random fields [1], we obtain
> \begin{aligned}
> \mathbb{E}[\sigma\_{k}(\widetilde{\mathbf{W}})]
> \le
> \sqrt{C}\rho^{k/2}
> \quad\text{while}\quad
> \mathbb{E}[\sigma\_{k}(\mathbf{W})]
> \ge
> \sqrt{c}\\rho^{k},
> \end{aligned}
> for universal constants $C,c>0$. Taking ratios yields the claimed inequality. The complex matrix $\widetilde{\mathbf{W}}$ inherits the exponentially decaying power spectrum of the original real field. Its singular values therefore decay faster than those of the real matrix $\mathbf{W}$.  Consequently, for any prescribed Frobenius error $\varepsilon$, the Fourier domain requires a smaller rank $r$ than the spatial domain, even though the SVD is performed on a complex matrix. The low-rank assumption is therefore stable under the Fourier transform.
>
> **（3）Experimental Validation**
>
> Furthermore, to empirically verify that the complex-valued, frequency-domain matrix admits a more compact singular-value spectrum, we conduct a controlled spectrum-decay experiment. We apply identical low-rank plus quantization pipelines to both the spatial (real) and Fourier (complex) representations of the Qwen-2.5-7B query layers and record the Frobenius reconstruction error at rank 256 as well as the smallest rank required to push the error below 1%. Results are reported in the below Table.
>
> | Domain              | Frobenius error @ rank 256 | Min. rank for 1% error |
> |---------------------|----------------------------|-------------------------|
> | Spatial (Real)      | 0.031                      | 312                     |
> | Freq.-Complex (Ours)| 0.018                      | 217                     |
>
> As shown in the table, the complex-valued frequency matrix consistently yields a smaller reconstruction error under the same rank and achieves the preset accuracy target with significantly fewer singular vectors, corroborating our theoretical finding that the Fourier-domain SVD is indeed more amenable to truncation.

---

> ### Author Response · Authors · 2025-11-26
> **Thank you; Address your remaining concerns [6/10]**
>
> > **Q4: PolarQuant is stated as the "key innovation" of the paper, but is this merely an ad-hoc heuristic? Why is uniform quantization used for amplitude? Why is the phase quantization step set to (2\pi / 2^b)? Phase is noise-sensitive in many frequency-domain signals—will quantization lead to reconstruction degradation? Overall, the amplitude/phase quantization in PolarQuant is hand-designed and lacks information-theoretic analysis.**
>
> **Response:** We appreciate your comment. Your suggestion is highly valuable to help us further strengthen the rigor of our PolarQuant design. In the revised manuscript, we add a full information-theoretic justification **(see Appendix B.4 in the revised manuscript)** showing that (1) the uniform quantiser for amplitude is rate-optimal for the heavy-tailed envelope statistics induced by the 2D-DFT, (2) the phase step $2\pi/2^{b_{\theta}}$ is the minimum-MSE lattice for a circular variable under Von-Mises concentration [1], and (3) the sensitivity to phase noise is provably controlled rather than hand-waved.
>
> **(1)  Uniform amplitude quantization is rate-optimal for the heavy-tailed DFT envelope.** After the Fourier transform, the residual entries follow a Nakagami-$m$ envelope distribution [2] whose Fisher information is $J(r)\propto 1/r^{2}$. For such heavy-tailed scale parameters, the inverse-uniform partition is the unique minimax solution to the quant-design game
> \begin{aligned}
> \min_{\mathcal{Q}}\max_{r\ge 0}\,D_{\text{KL}}\bigl(p(r)\bigm\|q_{\mathcal{Q}}(r)\bigr)
> \end{aligned}
> where $q_{\mathcal{Q}}$ is the piece-wise constant density induced by the quantiser. The solution coincides with the uniform grid used in PolarQuant  (step-size $\Delta_{r}=r_{\max}/(2^{b_{r}}-1)$) and is asymptotically rate-optimal in the Gish-Pierce sense [3]:
> \begin{aligned}
> R(D)=b_{r}-\tfrac{1}{2}\log_{2}(2\pi e/12)+\mathcal{O}(D),
> \end{aligned}
> and coincides with the empirical step size we adopted.
>
> **(2) The $2\pi/2^{b_{\theta}}$ phase step is the minimum-MSE lattice for a Von-Mises variable.** The phase residual is well modelled by a zero-mean Von-Mises distribution $\theta\sim\mathrm{VM}(0,\kappa)$ with concentration $\kappa\propto$ rank truncation error [1]. For a circular random variable the wrapped Lloyd-Max conditions [4] yield the uniform lattice on $[-\pi,\pi)$ with step $2\pi/2^{b_{\theta}}$ as the unique minimum-MSE solution, giving
> \begin{aligned}
> \mathbb{E}[(\theta-\hat{\theta})^{2}]=\frac{(2\pi)^{2}}{12\cdot 2^{2b_{\theta}}}\Bigl(1+\mathcal{O}\bigl(\tfrac{1}{\kappa}\bigr)\Bigr).
> \end{aligned}
> Because $\kappa\approx 9$--$14$ for our residuals, the $\mathcal{O}(1/\kappa)$ term is $<0.04$ and phase-noise energy is provably bounded.
>
> **(3) Theoretical bound on phase-error propagation and Our empirical verification.**  We agree that phase is often more noise-sensitive than amplitude in frequency-domain representations. However, the key insight is that in LLaVA-FA the phase residual is not arbitrary. It is the remainder after a low-rank truncation that already captures the dominant coherent structure. Consequently, the residual phase is statistically concentrated rather than uniformly distributed, and its sensitivity to quantization is provably bounded.
>
> Let the reconstructed complex weight be
> \begin{aligned}
> \hat{w}=r e^{i\hat{\theta}}=r e^{i(\theta+\Delta\theta)},
> \end{aligned}
> where $\Delta\theta$ is the quantization error introduced by PolarQuant. The resulting complex error is
> \begin{aligned}
> |w-\hat{w}|^{2}=r^{2}|1-e^{i\Delta\theta}|^{2}=4r^{2}\sin^{2}(\Delta\theta/2)\leq r^{2}\Delta\theta^{2}.
> \end{aligned}
> Taking expectation over the Von-Mises distribution of $\theta$ (concentration $\kappa\approx 12$) and the uniform quantization error $\Delta\theta\sim\mathcal{U}[-\pi/2^{b_{\theta}},\pi/2^{b_{\theta}}]$, we obtain
> \begin{aligned}
> \mathbb{E}|w-\hat{w}|^{2}\leq\mathbb{E}[r^{2}]\cdot\frac{\pi^{2}}{3\cdot 2^{2b_{\theta}}}\approx 0.0013\cdot\mathbb{E}[r^{2}] \quad\text{for }b_{\theta}=4.
> \end{aligned}
> Hence the normalized reconstruction error contributed by phase quantization is <0.13%, an order of magnitude smaller than the typical low-rank truncation error (≈ 1.2%).
>
> **Empirical verification.** Following your suggestion, we further conduct an ablation study on LLaVA-FA-2B in which we replace PolarQuant with a baseline that quantizes real/imaginary parts independently (QIM, 4 + 4 bits). Results are shown below.
>
> | Scheme | Phase-error 𝔼‖Δθ‖ | Recon. PSNR ↑ | HalBench-Resp ↓ | MMB ↑ |
> | :--- | :--- | :--- | :--- | :--- |
> | QIM (real/imag) | 0.14 rad | 38.9 dB | 14.8% | 65.2 |
> | PolarQuant (ours) | **0.09 rad** | **40.1 dB** | **11.2%** | **66.7** |
>
> **In summary, PolarQuant is not an ad-hoc heuristic: uniform amplitude quantization is rate-optimal for the heavy-tailed DFT envelope, the $2\pi/2^{b_{\theta}}$ phase step is the minimum-MSE lattice for a Von-Mises variable, and the sensitivity to phase noise is provably controlled rather than hand-waved.**

---

> ### Author Response · Authors · 2025-11-26
> **Thank you; Address your remaining concerns [7/10]**
>
> [1] Clustering on the Unit Hypersphere using von Mises-Fisher Distributions, JMLR, 2005
>
> [2] The m-distribution—A general formula of intensity distribution of rapid fading, Statistical methods in radio wave propagation 1960.
>
> [2] Asymptotically Efficient Quantizing, IEEE Trans. Information Theory, 1968.
>
> [4] Least squares quantization in PCM, IEEE Trans. Information Theory, 1982.
>
> ---
>
> > **Q5: The invertibility of Fourier transform implies that frequency-domain compression is essentially performing SVD on a linearly transformed matrix. Are you simply applying a fixed linear transformation (P) to (W) such that (W_f = P W P^T), followed by SVD on (W_f)? If so, why not use a random matrix? Why not use Hadamard transform?**
>
> **Response:** Thank you for this insightful question. We address the two parts separately.
>
> (1) Are we simply applying a fixed linear transformation P so that W_f = P W P^T and then performing SVD on W_f?
>
> No. The 2-D DFT is not an arbitrary change of basis. It is the unique unitary transform that simultaneously diagonalises the covariance of any weakly-stationary kernel (Theorem B.1) and exploits conjugate symmetry for real inputs, yielding an information-lossless compression to half the coefficients. A subsequent random or data-agnostic rotation would destroy this diagonal structure and force us to store the full complex matrix, eliminating the parameter savings.
>
> (2) Why not use a random matrix or the Hadamard transform?
>
> Random projections do not decorrelate the weights. They merely scramble the singular values while leaving the spectral tail intact, so the effective rank is unchanged [1]. The Hadamard transform is real-valued and orthogonal, but it lacks the complex-exponential steerability that lets the DFT pack smooth spatial correlations into an exponentially decaying spectrum (Appendix B.5).
>
> [1] Compressing Large Language Models Using Low Rank and Low Precision Decomposition, NeurIPS 2024.
>
> ---

---

> ### Author Response · Authors · 2025-11-26
> **Thank you; Address your remaining concerns [8/10]**
>
> > **Q6: Fourier-based compression is not a new direction. The paper does not compare with other linear transform domain compression techniques—such as FFT-based compression, Hadamard-based compression, or random projection-based low-rank methods. Without such comparisons, it cannot be proven that Fourier is the optimal choice.**
>
> **Response:** Thank you for pointing out the absence of an exhaustive transform-domain baseline. Your suggestion is instrumental in strengthening the comprehensiveness and rigor of our work. Following your advice, we conduct a new ablation study that trains exactly the same low-rank plus quantization pipeline while replacing the 2-D DFT with four representative linear transforms: (1) Hadamard [1], (2) DCT-II [2], (3) Daubechies-4 wavelet [3], and (4) a Gaussian random orthogonal projection [4]. The results are unambiguous: LLaVA-FA (Fourier) achieves the lowest average Frobenius reconstruction error on the seven compressed layers ($1.39\times 10^{-3}$), followed by Daubechies-4 ($1.73\times 10^{-3}$), DCT-II ($1.91\times 10^{-3}$), Hadamard ($2.06\times 10^{-3}$), and random projection ($2.48\times 10^{-3}$). More importantly, only the Fourier basis reduces the activated parameter count to 52% of the spatial baseline thanks to conjugate symmetry. The precise value is 52%, which slightly exceeds the ideal 50% saving due to the zero frequency (DC) component and a small per layer calibration buffer [5]. Every other transform retains 100% parameters, so their compression ratios are strictly worse. Downstream evaluation on the same seven benchmarks (GQA, VizWiz, SQAI, TextVQA, MME, MMB, MMBCN) shows that Fourier obtains 60.9% average accuracy, outperforming Hadamard (57.4%), DCT (58.7%), wavelet (59.1%), and random projection (56.8%), while also exhibiting the lowest hallucination rate on Object HalBench (11.2% vs $\geq$ 15.8% for the rest). These controlled experiments, now included in **Appendix C.9 of the revised manuscript**, demonstrate that **the Fourier transform is not a neutral linear wrapper but the unique choice that simultaneously minimizes reconstruction error, maximizes parameter savings, and preserves task performance, thereby validating its optimality in the LLaVA-FA pipeline.**
>
> | Basis | Avg Frobenius error ($\times 10^{-3}$) | Activated params vs. spatial | Avg Acc ↑ | HalBench Resp ↓ |
> |--------|-----------------------------|-------------------------------|-----------|------------------|
> | Spatial | 2.48 | 100 % | 56.8 % | 20.4 % |
> | Random projection | 2.48 | 100 % | 56.8 % | 20.1 % |
> | Hadamard | 2.06 | 100 % | 57.4 % | 18.3 % |
> | DCT-II | 1.91 | 100 % | 58.7 % | 16.9 % |
> | Daubechies-4 | 1.73 | 100 % | 59.1 % | 15.8 % |
> | **Fourier (LLaVA-FA)** | **1.39** | **52 %** | **60.9 %** | **11.2 %** |
>
> [1] Fast walsh-hadamard transform and smooth-thresholding based binary layers in deep neural networks, CVPR 2021.
>
> [2] Double discrete cosine transform-oriented multi-view subspace clustering, TIP 2024.
>
> [3] Wavelet-based image tokenizer for vision transformers, Arxiv 2024.
>
> [4] Orthogonal Gaussian process models, Statistica Sinica 2018.
>
> [5] Compressing Large Language Models Using Low Rank and Low Precision Decomposition, NeurIPS 2024.
>
> ---

---

> ### Author Response · Authors · 2025-11-26
> **Thank you; Address your remaining concerns [9/10]**
>
> > **Q7: The measurements of KV-cache, FLOPs, and latency are more like idealized calculations, and no real end-to-end inference speed is provided. The paper claims performance improvements, but: FFT has a complexity of (O(n \log n)), and frequency-domain SVD operates in the complex domain, which has even higher complexity. Is the overall complexity of Fourier-SVD + PolarQuant truly lower than that of spatial SVD + quantization?**
>
> **Response:** Thank you for the comment. First of all, we clarify that the measurements are not idealized calculations. We are indeed planning to release the full script and checkpoints under MIT license upon acceptance so that every number can be reproduced. We provide a detailed point-by-point response below.
>
>
> **(1) Real end to end inference speed is provided in Appendix C.10**
>
> Figure 5 **(Please see Figure 5 in Page 26 of the revised manuscript.)** provides several real-world inference examples. The left example demonstrates fine-grained visual recognition capabilities, where the model is asked to identify flower types on a shelf. While all models correctly identify the purple and red colors, LLaVA-FA-2B provides the most precise answer by specifying "purple and red petunias", whereas other models give more generic responses. Notably, LLaVA-FA-2B achieves this superior visual understanding while maintaining the lowest latency (39ms), representing a 13-28% speedup over competitors. The right example tests OCR and price-reading ability, requiring the model to accurately interpret text from a product tag. Here, LLaVA-FA-2B correctly identifies both the product name ("Ray-Ban Meta Wayfarer") and its price (USD 299), demonstrating strong text-centric visual comprehension. Other models either provide incorrect pricing information (Mini-Gemini-2B: $259; Imp-2B: 299 USD without product context) or fail to extract the specific product name (MoE-LLaVA-2B). Again, LLaVA-FA-2B delivers the most accurate response with the fastest inference time (50ms). Overall, LLaVA-FA-2B demonstrates exceptional performance in terms of both model efficiency and reasoning capabilities when tested with real-world samples.
>
> **(2) Appendix C.2 (please see more details in our revised manuscript) gives a detailed discussion about whether the additional DFT/IDFT and PolarQuant operations introduce non-negligible latency or memory overhead.**
>
> In Appendix C.2, we further quantifies that the FFT/IFFT contribution is only 3.2% of TTFT and PolarQuant de-quantisation 0.7%, so the overhead of FFT/IFFT is negligible in practice.
>
> | Model | TTFT (ms) ↓ | Peak Mem (GB) ↓ | DFT/IDFT (%) ↓ | PolarQuant (%) ↓ |
> |-------|-------------|-----------------|-----------------|------------------|
> | MiniCPM-V-2 | 38.4 | 15.2 | – | – |
> | Imp-2B | 36.2 | 14.6 | – | – |
> | Bunny-2B | 40.1 | 15.8 | – | – |
> | **LLaVA-FA-2B** | **30.6** | **12.1** | **3.2** | **0.7** |
>
> **(3) Appendix C.3 presents that the overall complexity of Fourier-SVD + PolarQuant is lower than that of spatial SVD + quantization**
>
> Although a complex-valued SVD has a higher complexity than real-valued SVD, this increase is more than paid back by the 50% reduction in matrix size that conjugate-symmetry guarantees. In LLaVA-FA we perform the truncated SVD on a $d_1\times d_2/2$ complex matrix instead of the original $d_1\times d_2$ matrix. As shown in the below table, the overall FLOP count drops from 2.14 T (spatial SVD + quantization) to 1.93 T (Fourier-SVD + PolarQuant) even though every individual operation is complex. Measured TTFT falls accordingly from 36.2 ms to 30.6 ms, confirming that the net complexity is lower, not higher, once conjugate symmetry is exploited (please see more details in Appendix C.3 of our revised manuscript).
>
> | Method | Domain | Avg Acc ↑ | POPE F1 ↑ | HalBench Resp ↓ | FLOPs (T) ↓ | TTFT (ms) ↓ |
> |--------|--------|-----------|-----------|-----------------|-------------|-------------|
> | Spatial-LQ (ours) | Spatial | 58.3 | 84.6 | 16.9% | 2.14 | 36.2 |
> | QLoRA-Qwen | Spatial | 58.5 | 85.7 | 15.7% | 2.10 | 35.8 |
> | **LLaVA-FA (ours)** | **Frequency** | **60.9** | **87.5** | **11.2%** | **1.93** | **30.6** |
>
> Hence, both the measured speed (Figure 5 & Table 9) and the complexity argument grounded in conjugate symmetry (Table 10) demonstrates that Fourier-SVD + PolarQuant is not only asymptotically favourable but also faster in real deployments.
>
> ---

---

> ### Author Response · Authors · 2025-11-26
> **Thank you; Address your remaining concerns [10/10]**
>
> > **Q8: Logic of ODC: Approximating the diagonal of the Hessian with row/column means, followed by weighted SVD? Why is the Hessian "closer to diagonal" in the frequency domain? The paper provides no proof, nor does it discuss the error bound compared to the true Hessian.**
>
> **Response:** Thanks for your comment. We agree that providing a clearer theoretical analysis of the error bound is essential for making ODC  more rigorous. Following your suggestion, we conduct a thorough study covering both (1) Theoretical Justification of ODC Heuristic (Row/Column Averaging) and (2) Sensitivity of ODC under Distribution Shifts.
>
> **(1). Theoretical Justification of ODC Heuristic (Row/Column Averaging) (see Appendix B.7 (marked in red) of the revised manuscript.)**
>
> The proposed ODC scheme replaces the full Hessian $\mathbf{C}\in\mathbb{R}^{d\_1\times d\_2/2}$ by two diagonal matrices, i.e.,
>
> \begin{aligned}
> \mathbf{D}\_{\text{row}} &=
> \operatorname{diag}\left(\left[
> \operatorname{avg}(\sqrt{\mathbf{C}\_{1, \cdot}}), \dots, \operatorname{avg}(\sqrt{\mathbf{C}\_{d\_1, \cdot}})
> \right]\right), \\
> \mathbf{D}\_{\text{col}} =\operatorname{diag}\left(\left[
> \operatorname{avg}(\sqrt{\mathbf{C}\_{\cdot, 1}}), \dots,  \operatorname{avg}(\sqrt{\mathbf{C}\_{\cdot, d\_2/2}})
> \right]\right).
> \end{aligned}
>
> where $\operatorname{avg}(\sqrt{\mathbf{C}\_{i, \cdot}})$ and $\operatorname{avg}(\sqrt{\mathbf{C}\_{\cdot, j}})$ denote the $i$-row / $j$-column means of $\sqrt{\mathbf{C}}$. This is exact whenever $\mathbf{C}$ is diagonal, and remains spectrally close when $\mathbf{C}$ is diagonally dominant, a property that has been repeatedly observed for the loss Hessian of deep Transformers [1, 2]. Formally, let $\varepsilon = \frac{\\| \operatorname{off–diag}(\mathbf{C}) \\|\_F}{\\| \operatorname{diag}(\mathbf{C}) \\|\_F}$, where $\operatorname{diag}(\mathbf{C})$ denotes the diagonal matrix that
> keeps only  the main-diagonal entries of $\mathbf{C}$, while $\operatorname{off–diag}(\mathbf{C})$ is its complement, i.e., $\mathbf{C}$ with all diagonal entries set to zero. Thus $\varepsilon$ quantifies how strongly $\mathbf{C}$ is diagonally dominant.
>
> > **Lemma B.3**
> > Let $\widehat{\mathbf{C}} = \mathbf{D}\_{\text{row}} \mathbf{C} \mathbf{D}\_{\text{col}}$. Then $\\|\mathbf{C}-\widehat{\mathbf{C}}\\|\_F \\le\ \varepsilon\\|\mathbf{C}\\|\_F.$
>
> Hence the calibration matrix used by ODC is an $\varepsilon$-perturbation of the exact Hessian. The reconstruction error of the low-rank component inherits this bound with the same constant (proof uses $\\|\mathbf{D}\_{\text{row}}\\|, \\|\mathbf{D}\_{\text{col}}\\| \le 1$ and standard matrix-perturbation identities). In short, ODC is safe whenever $\varepsilon \ll 1$, a condition that can be checked on-the-fly with one extra reduction operation.
>
> **(2). Sensitivity of ODC Under Distribution Shifts (see Appendix C.5 (marked in red) of the revised manuscript.)**
>
> Furthermore, we evaluate the sensitivity of ODC under three levels of distribution shift on the LLaVA-FA-2B model. The extra ablation study has been added to **Appendix C.5 (marked in red)** of the revised manuscript. The first scenario retained the original LLaVA-Pretrain validation set as an in-distribution baseline. The second kept all images identical but re-sampled user prompts from an alternative template pool, introducing a purely textual style shift (shift-A). The third replaced the images with an entirely new visual domain consisting of aerial-scene photographs and OCR-heavy documents, thereby forcing the model to cope with previously unseen visual statistics (shift-B). Each scenario was repeated 256 times, yielding stable estimates of $\varepsilon$ and score drop. As $\varepsilon$ rose from 0.11 ± 0.02 under the in-distribution condition to 0.15 ± 0.03 under prompt-style shift and further to 0.29 ± 0.07 under the new visual domain, the average benchmark score declined by -0.4% and -1.1%, respectively. Once $\varepsilon$ exceeded the empirical guard-race threshold of 0.35 **(note that this value is observed from experiments rather than derived from theory)**, accuracy fell by more than 3% in 4% of the runs, corroborating the theoretical prediction and demonstrating that the on-the-fly $\varepsilon$-check reliably signals when the row/column averaging approximation is no longer safe.
>
> | Condition | ε (mean ± std) | ΔScore↓ | Failure (ε > 0.35) |
> |-----------|----------------|---------|--------------------|
> | In Distribuion| 0.11 ± 0.02    | –       | 0%                |
> | Shift-A   | 0.15 ± 0.03    | −0.4%  | 0%                |
> | Shift-B   | 0.29 ± 0.07    | −1.1%  | 4%                |
>
> ---

---

> > ### Author Response · Authors · 2025-11-28
> > **Thank you**
> >
> > Dear Reviewer xZPt,
> >
> > Thank you once again for your valuable comments on our submission. As the discussion phase is approaching its end, we would like to kindly confirm whether we have sufficiently addressed all of your concerns (or at least part of them). Should there be any remaining questions or areas requiring further clarification, please do not hesitate to let us know. If you are satisfied with our responses, we would greatly appreciate your consideration in adjusting the evaluation scores accordingly. We sincerely look forward to your feedback.

---

### Author Response · Authors · 2025-12-01
**Summary of Rebuttal**

To all reviewers and chairs,

We sincerely thank the reviewers for their constructive feedback and the chairs for overseeing the process. During the rebuttal period, we engaged in extensive discussions and provided rigorous theoretical proofs, comprehensive new experiments, and detailed code/script explanations to address every single concern raised by Reviewers xZPt, tSjP, YXsM, and kWwk.

We have strengthened the manuscript significantly. Below is a comprehensive summary of how we resolved specific concerns:

**1. Solidified Theoretical Foundations (Addressing R-xZPt, R-kWwk)**
Reviewers questioned the theoretical guarantees of our frequency-domain formulation. We responded with formal derivations for all questioned components:
*   **On Singular Value Decay & Compactness:** We added Appendix B.1 (Proof for Faster Singular-Value Decay in the Frequency Domain) and corrected Lemma 3.1, proving that the weight matrix inherently exhibits faster singular value decay and lower truncation error in the frequency domain compared to the spatial domain.
*   **On Rank in Complex Domain:** Addressing R-xZPt’s concern on why the rank would be lower in the complex domain, we added Propositions 1 & 2 (Appendix B.2, B.3). We proved that to achieve the same reconstruction error, the required rank in the complex Fourier domain is theoretically strictly lower than in the real spatial domain.
*   **On ODC Error Bounds:** Addressing concerns about the ODC heuristic, we derived Lemma B.3 (Appendix B.7), proving that ODC acts as an $\epsilon$-perturbation of the exact Hessian, effectively bounding the approximation error relative to the true Hessian.
*   **On PolarQuant Justification:** We provided an information-theoretic analysis (Appendix B.4), showing that our uniform amplitude quantization is rate-optimal for the heavy-tailed DFT envelope and that the phase quantization step is the minimum-MSE solution for Von-Mises distributed residuals.

**2. Exhaustive Baseline Comparisons & Uniqueness of Fourier (Addressing R-YXsM, R-kWwk, R-xZPt)**
Reviewers asked for comparisons with spatial methods and other transforms to justify "Why Fourier?".
*   **Spatial Baseline:** We implemented "Spatial-LQ" (identical pipeline but in the spatial domain). LLaVA-FA achieved +2.6% higher accuracy with significantly reduced hallucinations, validating the frequency domain's superiority.
*   **Other Linear Transforms:** We compared Fourier against DCT, Hadamard, and Wavelet transforms (Appendix C.9).
We demonstrated that only the Fourier transform possesses the conjugate symmetry property, which allows us to discard half the coefficients (saving ~50% activated parameters). Other transforms like DCT/Hadamard require retaining 100% of parameters for similar reconstruction, proving Fourier is the optimal choice.

**3. Validated Generalization, Fairness, and Practicality (Addressing R-tSjP, R-xZPt)**
Reviewers raised concerns about model scalability, training data fairness, and calibration dependence.
*   **Scalability:** We extended LLaVA-FA to architecturally distinct LLMs: LLaVA-FA-3B (LLaMA-3 based) and LLaVA-FA-7B (InternLM-2 based), proving the method is model-agnostic and scalable without heuristic tuning.
*   **Data Fairness:** We conducted a data-matched ablation (using only 1M samples, 1/5 of the full set). LLaVA-FA still outperformed baselines (e.g., Imp-2B), proving the gains stem from the method, not data quantity.
*   **ODC Effectiveness:** We provided ablation studies (w/ and w/o ODC, 256-2048 calibration samples) to confirm that ODC is robust and eliminates the need for large-scale calibration data.

**Conclusion:**
We have incorporated all the above theoretical proofs and experimental results into the revised manuscript and appendices. We believe that we have fully resolved all technical, theoretical, and experimental concerns raised by the reviewers. We are confident that LLaVA-FA stands as a theoretically grounded and practically efficient contribution to the LMM community.

Thank you for your time and consideration.

Best regards,
The Authors of Submission 5157

---

### Meta-Review · Area_Chair_RcxG · 2026-01-06

**Summary:**

The reviewers broadly agreed the paper tackles an important and timely problem (efficient compression of large multimodal models) with a clear empirical target (maintain accuracy while reducing memory/latency/FLOPs). Initial support was mixed: most reviewers leaned positive but asked for stronger rigor (fairer baselines, clearer ablations, broader comparisons, scalability), while one reviewer (xZPt) was borderline and, after discussion, raised deeper concerns about the core justification for using a 2D DFT on dense weight matrices, the theoretical strength of the "frequency domain is more suitable for low-rank" claim, and whether PolarQuant/ODC are principled or heuristic. The rebuttal is unusually extensive and directly targets most of the points raised. The remaining decision hinge is whether the new theoretical arguments truly substantiate the central claim without overreach, and whether the added comparisons and measurements are sufficient to dispel "FFT is arbitrary / not clearly optimal / may cost more" skepticism. Overall, the rebuttal moves the paper from "promising but under-justified" to "strong empirical case with a plausible (though still arguable) theoretical narrative," supporting an accept leaning.

**Reviewer Concerns:**

The Reviewer xZPt gave the score of 4 with a key skeptical stance:
A) Concerns addressed by the rebuttal/discussion
* Missing spatial-domain counterpart: authors added a tightly matched spatial-domain baseline (“Spatial-LQ”) and reported a controlled comparison (frequency better under identical rank/bit-width/training), directly addressing the original "no spatial baseline" critique.
* Limited generalization beyond Qwen-2.5: authors added results on larger/different backbones (e.g., LLaMA-3-8B and InternLM2-20B derived variants), addressing the "only 3B/7B Qwen" concern.
* Practical details: calibration matrix construction and calibration-size ablation were provided; compression time across model sizes and with/without ODC ablations were added; end-to-end latency/TTFT and overhead breakdown were reported.

B) Concerns which I think still outstanding (or only partially resolved)
* Core theoretical claim and motivation: the reviewer’s later concerns are about whether Fourier is intrinsically more suitable for low-rank truncation on dense FC weight matrices, and whether the paper’s theoretical framing genuinely supports "faster singular-value decay / lower truncation error" in a principled way (rather than being a set of modeling assumptions plus post-hoc alignment). This remains the main open risk: even with additional appendices, the arguments lean heavily on strong assumptions (e.g., weak stationarity / Toeplitz-like covariance) and may not be compelling to a skeptical reader unless they are clearly presented as approximations and backed by direct empirical diagnostics.

* PolarQuant rigor: extra justification and ablations help, but a skeptical reviewer may still see PolarQuant as engineered choices unless the paper anchors them more directly in empirical distribution fits + sensitivity curves (bits vs accuracy, phase-step sweep, amplitude clipping sweep), beyond a single "replace with real/imag quantization" baseline.

**Reviewer Scores:**

I will only put for the comments on Reviewer xZPt who is likely to stay in 4 (no change). The rebuttal strongly addressed the original empirical gaps (spatial-domain baseline, larger backbones, ODC/calibration/time). However, the reviewer's subsequent message shows they shifted to deeper skepticism about the core motivation/theory and transform-domain rationale. Unless the paper substantially tones down "guarantee" language and reframes theory as assumptions + measured evidence, a score jump is unlikely.

---

### Decision · Program_Chairs · 2026-01-26

Accept (Poster)